YITP-SB-2023-14

# Majorana chain and Ising model – (non-invertible) translations, anomalies, and emanant symmetries

Nathan Seiberg[1] and Shu-Heng Shao[2]

[1]*School of Natural Sciences, Institute for Advanced Study*
[2]*C. N. Yang Institute for Theoretical Physics, Stony Brook University*

**Abstract**

We study the symmetries of closed Majorana chains in 1+1d, including the translation, fermion parity, spatial parity, and time-reversal symmetries. The algebra of the symmetry operators is realized projectively on the Hilbert space, signaling anomalies on the lattice, and constraining the long-distance behavior. In the special case of the free Hamiltonian (and small deformations thereof), the continuum limit is the 1+1d free Majorana CFT. Its continuum chiral fermion parity $(-1)^{F_L}$ emanates from the lattice translation symmetry. We find a lattice precursor of its mod 8 't Hooft anomaly. Using a Jordan-Wigner transformation, we sum over the spin structures of the lattice model (a procedure known as the GSO projection), while carefully tracking the global symmetries. In the resulting bosonic model of Ising spins, the Majorana translation operator leads to a non-invertible lattice translation symmetry at the critical point. The non-invertible Kramers-Wannier duality operator of the continuum Ising CFT emanates from this non-invertible lattice translation of the transverse-field Ising model.

# 1  Introduction

Symmetries are a powerful tool in the analysis of physical systems. They help organize lattice models and continuum quantum field theories and constrain their physical observables. A refinement of the notion of symmetries is their 't Hooft anomalies [1]. One of the applications of these anomalies is a relation between the microscopic problem and its possible macroscopic behavior [1]. This is particularly important when the microscopic problem is complicated and then its 't Hooft anomalies constrain its possible outcomes.

The modern view of 't Hooft anomaly in the context of continuum quantum field theory is the statement that while the system has a symmetry group $G$, it cannot be coupled to classical background gauge fields for $G$ in a gauge invariant way. See e.g., [2] for a recent review.

An independent line of investigation, starting with lattice models, also uses global symmetries and their realization to constrain the long-distance behavior of the system. A characteristic example of such a constraint is the famous Lieb-Schultz-Mattis (LSM) theorem [3] and its generalizations [4–31]. Various authors [17, 14, 16, 20, 19, 29, 32] have suggested that these results should be phrased as 't Hooft anomalies. In particular, [32] has presented a framework to couple the lattice system to background gauge fields and to probe for anomalies as a failure of their gauge symmetry. Here, we will follow this approach.

It is known that the description of symmetries and anomalies in fermionic systems is more subtle than in bosonic systems [33–44]. Our goal here is to present the analog of the analysis in [32] for fermionic systems. See the earlier discussion in [15, 45, 27, 46].

The models we will be discussing here have been studied by many researchers from various perspectives. In order to make the paper accessible to people in the different communities, our presentation will be self-contained, including reviews of many known results. Also, some of the free, and therefore exactly solvable, models that we will analyze are well-known for a long time. We will use them mostly in order to demonstrate the more general techniques based on the symmetries and their anomalies.

## 1.1 Majorana chain

Concretely, we will focus on a 1+1-dimensional system of Majorana fermions. (In [47], we will extend the discussion of this model in various directions and will provide more details. And in [48], we will study a similar system of Dirac fermions. See also [49, 50] for recent discussions of the translation symmetry of Dirac fermions coupled to gauge fields on the lattice.) Many authors have studied the anomalies of the continuum 1+1-dimensional theory of Majorana fermions. See e.g., the recent discussions in [51–53, 37, 38, 42, 43, 54]. And many authors have studied the lattice Majorana chain [55, 51, 56, 57, 15, 58]. In fact, the relation between this lattice model and the LSM theorem was discussed in [15]. Here we will follow the approach of [32] to bridge the gap between the lattice analysis and the continuum perspective of anomalies in these systems.

Our lattice consists of $L$ sites, labeled by $\ell$ with periodic boundary conditions, i.e.,

$$\ell \sim \ell + L\,. \tag{1.1}$$

On every site, we have a real fermion $\chi_\ell$ satisfying

$$\{\chi_\ell, \chi_{\ell'}\} = 2\delta_{\ell,\ell'}\,. \tag{1.2}$$

The anticommutation relations (1.2) mean that the Hilbert space $\mathcal{H}$ is a representation of this Clifford algebra. We will take it to be one copy of that representation. It is well known that the properties of these representations depend sensitively on $L$. For example, for even $L = 2N$, $\mathcal{H}$ is a sum of the two different spinor representations of $Spin(2N)$. They differ by the eigenvalue of

$$\mathsf{G} = \chi_1\chi_2\cdots\chi_L \qquad , \qquad L = 2N\,. \tag{1.3}$$

For odd $L = 2N + 1$, $\mathcal{H}$ is the unique spinor representation of $Spin(2N + 1)$ and

$$\mathcal{C} = \chi_1\chi_2\cdots\chi_L \qquad , \qquad L = 2N + 1 \tag{1.4}$$

is central. It acts as a c-number.[1]

---

[1]The quantum mechanical system of $2N+1$ real fermions is notoriously confusing. One approach, which we will follow here, is to let the Hilbert space be in a single $2^N$-dimensional spinor representation of $Spin(2N+1)$.

An alternative approach is to double this Hilbert space, $\mathcal{H} = \mathcal{H}^+ \oplus \mathcal{H}^-$. This means that we should add to the theory another decoupled fermionic operator $\chi_{L+1}$ that cannot be constructed out of the other $L = 2N+1$ fermions $\chi_\ell$. $\chi_{L+1}$ maps $\mathcal{H}^+ \leftrightarrow \mathcal{H}^-$. Now, the total Hilbert space $\mathcal{H}$ is $\mathbb{Z}_2$-graded and $\mathcal{C}$ is an operator rather than a c-number. In a way, this added fermion is like 't Hooft's spectator fermion [1], which cancels an anomaly. It is also related to the Majorana mode at the end of a higher dimensional SPT phase [55, 51, 56]. (This approach of adding a decoupled fermion has appeared also in [59].)

Neither of these approaches is compatible with the path integral presentation of this theory. The latter is well defined, but it leads to a partition function with a factor of $\sqrt{2}$, which does not admit a standard Hilbert space interpretation. See e.g., [60, 42, 54] for recent discussions.

It is amusing to compare the situation in the first two approaches to the case of a system with a spontaneously broken $\mathbb{Z}_2$ symmetry generated by the symmetry operator $\zeta$ satisfying $\zeta^2 = 1$. In finite volume, the symmetry is unbroken, $\zeta$ exists, and the Hilbert space is decomposed as $\mathcal{H} = \mathcal{H}^+ \oplus \mathcal{H}^-$ with the order parameter $\mathcal{C}$ having

We consider a Hamiltonian, which is a sum of local terms preserving $\mathbb{Z}_L$ lattice translation

$$T: \qquad \chi_\ell \to \chi_{\ell+1} \tag{1.5}$$

and fermion parity

$$\mathsf{G}: \qquad \chi_\ell \to -\chi_\ell . \tag{1.6}$$

This transformation will be related to the fermions parity transformation of the continuum theory, which is often denoted as $(-1)^F$. The reason we denote the lattice expression as $\mathsf{G}$ rather than $(-1)^F$ will become clear below. (For odd $L$, there is no operator $\mathsf{G}$ that implements the automorphism (1.6); it is an outer-automorphism. Yet, we can impose that the Hamiltonian is invariant under (1.6).)

A good example to keep in mind is the free Hamiltonian[2]

$$H = \frac{i}{2} \sum_{\ell=1}^{L} \chi_{\ell+1}\chi_\ell = \frac{i}{2} \sum_{\ell=1}^{L-1} \chi_{\ell+1}\chi_\ell + \frac{i}{2}\chi_1\chi_L . \tag{1.7}$$

But we emphasize that for most of our discussion, the particular form of the Hamiltonians is not essential. For example, as in [57, 58], we can add to it terms of the form $\sum_{\ell=1}^{L} \chi_\ell\chi_{\ell+1}\chi_{\ell+2}\chi_{\ell+3}$ without affecting its symmetries and our analysis.[3]

In addition to the lattice translation $T$ (1.5) and fermion parity $\mathsf{G}$ (1.6), the systems can also have parity $\mathsf{P}$ and time-reversal symmetry $\mathcal{T}$, which we will discuss in detail. (More precisely, these are symmetries of the system with even $L$, while the odd $L$ system does not have the symmetries $\mathsf{G}$, $\mathsf{P}$, and $\mathcal{T}$.)

We will also be interested in the Hamiltonian with a defect associated with fermion parity. For the Hamiltonian (1.7), the system with the defect is described by the Hamiltonian

$$H_\mathsf{G} = \frac{i}{2} \sum_{\ell=1}^{L-1} \chi_{\ell+1}\chi_\ell - \frac{i}{2}\chi_1\chi_L , \tag{1.8}$$

where the defect is on the link $(L, 1)$. This defect is topological; it can be moved by conjugating $H_\mathsf{G}$ by a unitary transformation. For example, conjugating it by the (fermionic) unitary operator $\chi_1$ moves the defect to the link $(1, 2)$. As in [32], in the presence of the defect, the symmetry operators are modified and their algebra is different.

---

eigenvalues $\pm 1$ in the two subspaces. (This is similar to the situation in the second approach.) In the infinite volume limit, the symmetry is spontaneously broken and the Hilbert space is split into two superselection sectors $\mathcal{H}^+$ and $\mathcal{H}^-$ with no local operator relating them. In this case, the order parameter $\mathcal{C}$ becomes a c-number in each sector and the symmetry operator $\zeta$ does not exist. (This is similar to the situation in the first approach.)

Finally, we should comment that this discussion of a system with an odd number of fermions involves another subtlety that arises when we consider a tensor product of our system with another system. The third approach that follows from the path integral assumes that we can take simple tensor products and leads to the peculiar factor of $\sqrt{2}$. For this reason, below, when we will consider tensor products of our systems we will have to be quite careful.

[2]In contrast, the Majorana chains in [55,51,56] do not generally have this lattice translation symmetry by one site.

[3]In the continuum limit, this term flows to an irrelevant deformation of the Ising CFT and therefore it does not change its extreme IR behavior near the critical point. Specifically, this operator flows to a product of the left-moving and the right-moving components of the stress tensor. To leading order, this deformation is the interesting $T\bar{T}$ deformation of [61].

## 1.2   Local Hilbert spaces and lattice translations

So far, this seems very similar to the starting point of the discussion in [32]. However, fermionic systems exhibit new subtleties.

Locality is central in continuum field theory and in lattice systems. However, in fermionic systems, fermions at separated points do not commute, but they anticommute. As is well understood, this fact is consistent with locality, but can raise questions once we consider symmetry operators and defects. We will show that this is particularly important for the analysis of the anomalies.

Often, the total Hilbert space is a tensor product of local Hilbert spaces

$$\mathcal{H} = \bigotimes_{j=1}^{N} \mathcal{H}_j \tag{1.9}$$

and we can examine how the various symmetry operators act on the local factors $\mathcal{H}_j$. Furthermore, with periodic (or twisted) boundary conditions, we can expect a translation operator that acts as

$$\mathcal{H}_j \to \mathcal{H}_{j+1} \qquad , \qquad j \sim j + N \,. \tag{1.10}$$

Indeed, the Hilbert space of the Majorana chain has such a structure. For even $L = 2N$, we can associate $\chi_{2j-1}$ and $\chi_{2j}$ with $\mathcal{H}_j$. And for odd $L = 2N + 1$, $\chi_L$ is also associated with $\mathcal{H}_N$. Furthermore, for even $L$, the fermion parity operator $\mathsf{G}$ can be written as a product of local factors each acting linearly on $\mathcal{H}_j$. However, since $\mathcal{H}_j$ is associated with two fermions at different sites, $\ell = 2j - 1$ and $\ell = 2j$, the translation generator $T$ does not act simply on $\mathcal{H}_j$. One might have expected that $T^2$ acts as in (1.10). But as we will discuss in Section 5, even this is not quite true. Hence, the decomposition (1.9) of the Hilbert space of the Majorana chain does not expose its $\mathbb{Z}_L$ translation symmetry.[4]

Because of these subtleties, we need to adjust the procedure of [32] such that it does not rely on the tensor product structure (1.9). We will first discuss the symmetry operators by specifying how they act on the fundamental fields $\chi_\ell$. This will lead us to concrete expressions for them in

---

[4]It is often the case that an internal symmetry operator $\mathbf{S}$ is given by a product of local operators

$$\mathbf{S} = \mathbf{s}_1 \mathbf{s}_2 \cdots \mathbf{s}_N \,, \tag{1.11}$$

where each factor $\mathbf{s}_j$ acts only on $\mathcal{H}_j$ of (1.9). The bosonic Heisenberg chain has such a decomposition, but the local factors $\mathbf{s}_j$ act projectively on $\mathcal{H}_j$. This fact is at the root of the LSM-anomaly. In that case, we could have taken $\mathcal{H}_j$ to include two sites, as we do in the Majorana chain. Then, the local symmetry operators $\mathbf{s}_j$ act linearly on $\mathcal{H}_j$. However, then, the translation operator does not have a simple action on $\mathcal{H}_j$; only $T^2$ acts simply, as in (1.10). Therefore, in this case, depending on how we define $\mathcal{H}_j$, the subtleties are either with $\mathbf{s}_j$ or with the action of $T$, leading to a mixed anomaly between the internal symmetry and translation [32]. For this reason we will refer to an internal symmetry as acting "on-site" only when its factors $\mathbf{s}_j$ act linearly on the local Hilbert space $\mathcal{H}_j$ and the translation operator $T$ maps $\mathcal{H}_j \to \mathcal{H}_{j+1}$.

terms of $\chi_\ell$. For example, for even $L = 2N$, we will find

$$\mathsf{G} = \chi_1 \chi_2 \cdots \chi_{2N} \,,$$

$$T = \frac{1}{2^{\frac{2N-1}{2}}} \chi_1 (1 + \chi_1 \chi_2)(1 + \chi_2 \chi_3) \cdots (1 + \chi_{2N-1} \chi_{2N}) \,, \quad (1.12)$$

$$T_\mathsf{G} = \frac{1}{2^{\frac{2N-1}{2}}} (1 - \chi_1 \chi_2)(1 - \chi_2 \chi_3) \cdots (1 - \chi_{2N-1} \chi_{2N}) \,,$$

where $T_\mathsf{G}$ is the translation symmetry operator of the problem with the $\mathsf{G}$-defect, e.g., the one described by the Hamiltonian (1.8). And for odd $L = 2N + 1$, where we do not have $\mathsf{G}$,

$$T = \frac{1}{2^N} (1 - \chi_1 \chi_2)(1 - \chi_2 \chi_3) \cdots (1 - \chi_{2N} \chi_{2N+1}) \,. \quad (1.13)$$

In the continuum limit, these three cases, even $L$ without or with the defect, and odd $L$ correspond to imposing different boundary conditions on the left- and right-moving fermions. More specifically, they correspond to the RR (periodic for the left- and right-movers), NSNS (anti-periodic for the left- and right-movers), and NSR or RNS boundary conditions, respectively.[5]

## 1.3 Anomalies on the lattice

In writing the expressions (1.12) and (1.13), we made an arbitrary phase choice. Since these symmetry operators act on the fields by conjugation, this arbitrary phase choice does not affect their action on the operators. However, it does affect the algebra they satisfy. Equivalently, the Hilbert space could be in a projective representation of the symmetry group. In order to understand this projective representation and the corresponding anomaly, we need to have better control of these phases.

When the Hilbert space factorizes as in (1.9), and the translation operator acts as $\mathcal{H}_j \to \mathcal{H}_{j+1}$, it has a natural phase. Similarly, internal symmetries can have a local action on $\mathcal{H}_j$ as in (1.11) and then their phase redefinitions are restricted. However, this is not the case in the Majorana chain and therefore it is less clear how to set the phases of the symmetry operators.

Instead, we will postulate that the allowed phase redefinitions of the symmetry operators in (1.12) and (1.13) are restricted. In particular, let us assume that the symmetry operator $\mathcal{S}$ can be written as a product of local factors

$$\mathcal{S} = \hat{\mathbf{s}}_1 \hat{\mathbf{s}}_2 \cdots \hat{\mathbf{s}}_L \,. \quad (1.14)$$

Unlike (1.11), here the local factors are labeled by the same index as the fermions $\ell = 1, \cdots, L$. The reason for that is that we do not assume that the Hilbert space factorizes as (1.9). By local factors we mean that $\hat{\mathbf{s}}_\ell$ is constructed out of the fermions near $\ell$ and that for sufficiently separated points the local factors $\hat{\mathbf{s}}_\ell$ commute

$$\hat{\mathbf{s}}_\ell \hat{\mathbf{s}}_{\ell'} = \hat{\mathbf{s}}_{\ell'} \hat{\mathbf{s}}_\ell \qquad , \qquad |\ell - \ell'| \bmod L > \ell_0 \,. \quad (1.15)$$

---

[5]Here we follow the string theory terminology: R stands for Ramond boundary condition, where the fermion is periodic, while NS stands for Neveu-Schwarz boundary condition, where the fermion is anti-periodic.

This condition is satisfied for all the symmetry operators in (1.12) and (1.13) except for G. (Below we will present other equivalent expressions for $T$ and $T_G$ that do not satisfy (1.15), but since their expressions in (1.12) and (1.13) do satisfy (1.15), the discussion below applies to them.) When the condition (1.15) is satisfied, we postulate that the allowed phase redefinitions are linear functions of $L$

$$\mathcal{S} \to e^{i(\alpha L + \beta)}\mathcal{S} \tag{1.16}$$

with the constants $\alpha$ and $\beta$ independent of $L$. Different operators, including $T$ for even and for odd $L$, can have different values of $\alpha$ and $\beta$. This restriction on the phase redefinitions can be thought of as local renormalization of the operators. We are not going to impose this restriction on the phase redefinition of the parity operator P, because it does not act locally on the chain.

After using the freedom to make such phase redefinitions, we can see whether the symmetry algebra has anomalous phases. Following 't Hooft, these anomalous phases should be present also in the long distance theory and thus they restrict its possible phases.

However, this immediately leads to the following question. The anomalies in [32] and here involve lattice translation. Since there are no anomalies involving continuum translation, how can the microscopic anomaly be realized in the macroscopic theory? The answer to this question is that the lattice translation symmetry leads to a new internal symmetry in the long-distance theory. As emphasized in [32], such a new symmetry should not be viewed as an emergent (equivalently, accidental) symmetry. Instead, it was named an *emanant symmetry* because it emanates from the underlying exact lattice translation symmetry. This emanant internal symmetry can have anomalies in the continuum theory. These anomalies should match the anomalies of the underlying lattice model.

As we will see, the same is true in the Majorana chain. For the specific choice of the Hamiltonians (1.7) and (1.8), the lattice translation leads to an emanant chiral fermion parity $(-1)^{F_L}$. More precisely, after a phase redefinition (1.16), the lattice translation operator $T_{\text{lattice}}$ is related to the chiral fermion parity operator $(-1)^{F_L}$ of the continuum massless Majorana CFT as[6]

$$T_{\text{lattice}} = (-1)^{F_L} e^{\frac{2\pi i P}{L}} . \tag{1.17}$$

which is an exact relation for the low-lying states on the lattice. Here $P$ is the continuum momentum operator.[7]

As in [32], the details of the emanant symmetry depend not only on the symmetry of the system, but also on the continuum limit. In the specific case of the Hamiltonian (1.7) or small deformations of it the emanant $(-1)^{F_L}$ symmetry can have additional anomalies. The continuum version of these anomalies have a known $\mathbb{Z}_8$ classification [62, 52, 63, 33], which is related to the critical spacetime dimension of the superstring theory [64, 65]. Here we will find a precursor of this anomaly for a lattice with finite $L$.

---

[6]Here $T_{\text{lattice}}$ collectively stands for $T_{\text{RR}}, T_{\text{NSNS}}, T_{\text{odd}}$ (defined in Section 3.3 and Section 4) for even $L$ without or with the defect, and odd $L$, respectively.

[7]Similarly, the lattice translation symmetry $GT$ leads to the right-moving chiral fermion parity $(-1)^{F_R}$. The two lattice translation symmetries $T$ and $GT$ are related by parity or time reversal, as do the two continuum symmetries that emanate from them.

## 1.4 Non-invertible lattice translations

Our discussion of the fermionic Majorana chain leads to interesting results for the Ising model. Unlike a bosonic theory, a fermionic theory depends on a choice of spin structure. This fact leads to essential differences between the symmetries and anomalies of bosonic and fermionic systems. Given a fermionic theory, depending on its anomalies, we can construct a closely related bosonic system by summing over these spin structures (a procedure known as the GSO projection [66] in the string theory literature).

In our case, the bosonic theory constructed out of the Majorana system, is the Ising model. In Section 5, we will review the relation between them on the lattice. And in Appendix A.3, we will discuss it in the continuum.

An interesting byproduct of the GSO projection on the lattice is that the Majorana lattice translation operator $T$ leads to a novel non-invertible symmetry D (defined in (6.7)) in the resulting bosonic lattice transverse-field Ising model.[8] This non-invertible symmetry D mixes with the lattice translation $T_{\text{Ising}}$ of the Ising model.

$$
\begin{aligned}
\mathsf{D}^2 &= \frac{1}{2}(1+\eta)T_{\text{Ising}}\,, \\
T_{\text{Ising}}^N &= 1\,,
\end{aligned}
\tag{1.18}
$$

where $\eta$ is the $\mathbb{Z}_2$ spin-flip symmetry and $N$ is the number of Ising spins. The Kramers-Wannier duality symmetry $\mathcal{D}$ of the continuum Ising CFT [76–79] emanates from this non-invertible symmetry D on the lattice; that is, it is a *non-invertible emanant symmetry*. We present this non-invertible symmetry on the lattice both in terms of a defect in Section 5.4, and of a conserved operator[9] in Section 6. See, for example, [80–84] for recent reviews on non-invertible symmetries.

## 1.5 Outline

The rest of the paper is organized as follows. Section 2 reviews the symmetries and their algebras in the NSNS, RR, NSR, and RNS Hilbert spaces of the continuum Majorana CFT. In Section 3, we analyze the closed Majorana chain. We study the translation, fermion parity, spatial parity, and time-reversal symmetries. In Section 3.3, we discuss the allowed phase redefinition of these symmetry operators, and find that the algebras are sometimes realized projectively on the Hilbert space, signaling an anomaly. In Section 4, we focus on the free fermion Hamiltonian, and compare the symmetries and their algebras with those in the free Majorana CFT in the continuum. In particular, we give the precise relation between the lattice translation operator and the chiral fermion parity, and find a precursor for the mod 8 anomaly on the lattice.

---

[8]Various authors [67–74] have discussed similar lattice operators, which roughly involve "translation by half the lattice spacing." (See also [75].) Our operator D is not exactly the same as theirs and it also obeys a different algebra.

[9]By definition, an invertible or a non-invertible symmetry operator $\mathbf{S}$ commutes with the Hamiltonian $H$. Therefore, using the Heisenberg equation of motion $i\frac{d}{dt}\mathbf{S} = [\mathbf{S}, H]$, it is independent of time, i.e., conserved. Therefore, we will use the phrases "conserved operator" and "symmetry operator" interchangeably.

In Section 5, we perform the Jordan-Wigner transformation of the Majorana chain, and discuss the (lack of) local Hilbert spaces. For even $L$, the lattice bosonization leads to the Ising model without or with a $\mathbb{Z}_2$ defect, while for odd $L$, we find the Ising model with a Kramers-Wannier duality defect. In Section 6, we discuss a non-invertible translation symmetry of the critical transverse-field Ising model.

Appendix A reviews standard facts about the continuum Majorana CFT and bosonization in 1+1d.

## 2 Majorana fermion CFT in 1+1d

Consider a single, non-chiral, massless, free Majorana fermion field in the continuum, with its left- and right-moving components denoted by $\chi_L, \chi_R$. The global symmetry includes the $\mathbb{Z}_2 \times \mathbb{Z}_2^f$ generated by $(-1)^{F_L}$ and $(-1)^F$, which act on the fields as

$$
\begin{aligned}
(-1)^F : & \quad \chi_L(t,x) \to -\chi_L(t,x), \quad & \chi_R(t,x) \to -\chi_R(t,x), \\
(-1)^{F_L} : & \quad \chi_L(t,x) \to -\chi_L(t,x), \quad & \chi_R(t,x) \to \chi_R(t,x).
\end{aligned}
\tag{2.1}
$$

Of course, we can also define $(-1)^{F_R} = (-1)^{F_L}(-1)^F$, which acts only on the right-movers. Below we will examine whether the operators generating these transformations actually exist in the theory and how they act on the various states.

There is a spatial parity symmetry $\mathcal{P}$ which acts as

$$
\mathcal{P} : \quad \chi_L(t,x) \to \chi_R(t,-x), \quad : \quad \chi_R(t,x) \to \chi_L(t,-x).
\tag{2.2}
$$

These symmetries generate a $D_4$ global symmetry, the dihedral group of order 8.

We will also discuss an anti-unitary time-reversal transformation $\mathcal{T}$[10]

$$
\mathcal{T} : \quad \chi_L(t,x) \to -\chi_R(-t,x), \quad \chi_R(t,x) \to -\chi_L(-t,x).
\tag{2.3}
$$

Finally, we will also discuss the spatial momentum operator $P$, which is related to the conformal weights as $P = h_L - h_R$.

The Lagrangian and Hamiltonian

$$
\begin{aligned}
L &= i \int dx \big( \chi_L(\partial_t - \partial_x)\chi_L + \chi_R(\partial_t + \partial_x)\chi_R \big) \\
H_{\text{Maj}} &= i \int dx \big( \chi_L \partial_x \chi_L - \chi_R \partial_x \chi_R \big)
\end{aligned}
\tag{2.4}
$$

are invariant under all these symmetries.

---

[10] An anti-unitary transformation $\mathcal{T}$ acts on a linear combination of operators $\mathcal{O}_1$ and $\mathcal{O}_2$ as $\mathcal{T}(c_1\mathcal{O}_1 + c_2\mathcal{O}_2)\mathcal{T}^{-1} = \bar{c}_1 \mathcal{T}\mathcal{O}_1\mathcal{T}^{-1} + \bar{c}_2 \mathcal{T}\mathcal{O}_2\mathcal{T}^{-1}$, where $\bar{c}_1$ and $\bar{c}_2$ are the complex conjugates of $c_1$ and $c_2$, respectively. The overall minus signs on the right-hand side are chosen so that it matches with the time-reversal symmetry on the lattice in Section 3.2. The signs can be flipped by redefining $\mathcal{T} \to (-1)^F \mathcal{T}$.

We can also add to the Lagrangian density a mass term

$$i\chi_L\chi_R \,, \tag{2.5}$$

with a real coefficient. It violates $\mathcal{P}$, $(-1)^{F_L}$, and $(-1)^{F_R}$, but preserves the symmetries $(-1)^F$, $\mathcal{P}' = (-1)^{F_L}\mathcal{P}$, and $\mathcal{T}$.

We quantize the CFT on a spatial circle parameterized by the coordinate $x$ with $x \sim x + 2\pi$. As common in the string theory literature, we refer to the anti-periodic boundary as the Neveu-Schwarz (NS) boundary condition, while the periodic boundary condition as the Ramond (R) boundary condition. We can choose independent boundary conditions for the left $\chi_L$ and the right fermions $\chi_R$, leading to four different quantizations of the Lagrangian (2.4). We refer to them as the NSNS, RR, NSR, and RNS theories and Hilbert spaces.[11] (More precisely, the NSR and RNS theories are equivalent because they are related by conjugation by either parity $\mathcal{P}$ or time-reversal $\mathcal{T}$.)

On a circle, the fermions obey the boundary conditions

$$\chi_L(t, x + 2\pi) = e^{2\pi i \nu_L}\chi_L(t, x) \,, \quad \chi_R(t, x + 2\pi) = e^{2\pi i \nu_R}\chi_R(t, x) \,, \tag{2.6}$$

where $\nu = 0$ (or $\nu = \frac{1}{2}$) corresponds to the R (or NS) boundary condition. We write the fermion fields in momentum modes

$$\chi_L(t, x) = \frac{1}{\sqrt{4\pi}} \sum_{r \in \mathbb{Z} + \nu_L} \chi_{L,r} \, e^{-ir(t+x)} \,,$$
$$\chi_R(t, x) = \frac{1}{\sqrt{4\pi}} \sum_{r \in \mathbb{Z} + \nu_R} \chi_{R,r} \, e^{-ir(t-x)} \,. \tag{2.7}$$

Canonical quantization leads to the standard anticommutation relation

$$\{\chi_{L,r}, \chi_{L,r'}\} = \delta_{r,-r'} \,, \quad \{\chi_{R,r}, \chi_{R,r'}\} = \delta_{r,-r'} \,. \tag{2.8}$$

When $\nu_L = \nu_R \bmod 1$, we have the parity and time-reversal symmetries, which act on the momentum modes as $\mathcal{P}\chi_{L,r}\mathcal{P}^{-1} = \chi_{R,r}$, $\mathcal{P}\chi_{R,r}\mathcal{P}^{-1} = \chi_{L,r}$ and $\mathcal{T}\chi_{L,r}\mathcal{T}^{-1} = -\chi_{R,r}$, $\mathcal{T}\chi_{R,r}\mathcal{T}^{-1} = -\chi_{L,r}$.

The Hamiltonian in momentum space is

$$H_{\text{Maj}} = \sum_{r=1-\nu_L}^{\infty} r \, \chi_{L,-r}\chi_{L,r} + \sum_{r=1-\nu_R}^{\infty} r \, \chi_{R,-r}\chi_{R,r} + \text{const} \,. \tag{2.9}$$

The ground state(s) in each Hilbert space is defined so that they are annihilated by all the positive momentum modes $r > 0$.

Below we analyze the algebras of the symmetry operators of the free Majorana CFT subject to different boundary conditions.

---

[11]In string theory, one performs a sum over spin structures. Correspondingly, the total Hilbert space of the worldsheet theory has several sectors. Each sector is a subspace of the Hilbert space of the fermionic theory with different boundary conditions [85]. (We will perform a similar sum over spin structures on the lattice in Section 5.3.) Here, on the other hand, we study a fermionic field theory with fixed spin structure. For this reason, we refer to these quantizations as the RR, NSNS, NSR, and RNS "theories" or "Hilbert spaces," rather than "sectors."

## 2.1 NSNS Hilbert space

In the NSNS theory, the fermions obey the boundary condition (2.6) with $\nu_L = \nu_R = \frac{1}{2}$. The NSNS boundary condition is natural from the continuum CFT point of view because of the operator-state correspondence. In particular, the identity operator corresponds to the non-degenerate ground state $|\Omega\rangle_{NSNS}$ in the NSNS Hilbert space, which is annihilated by all the raising operators:

$$\chi_{L,r}|\Omega\rangle_{NSNS} = \chi_{R,r}|\Omega\rangle_{NSNS} = 0 \,, \quad r = \frac{1}{2}, \frac{3}{2}, \cdots \,. \tag{2.10}$$

We can normalize the symmetry operators in the NSNS theory as

$$
\begin{aligned}
(-1)^F|\Omega\rangle_{NSNS} &= |\Omega\rangle_{NSNS} \,, \quad (-1)^{F_L}|\Omega\rangle_{NSNS} = |\Omega\rangle_{NSNS} \,, \\
\mathcal{P}|\Omega\rangle_{NSNS} &= |\Omega\rangle_{NSNS} \,, \quad \mathcal{T}|\Omega\rangle_{NSNS} = |\Omega\rangle_{NSNS} \,, \quad P|\Omega\rangle_{NSNS} = 0 \,.
\end{aligned}
\tag{2.11}
$$

It follows that, in the NSNS Hilbert space, we have the following algebra

$$
\begin{aligned}
\text{NSNS}: \quad & \left((-1)^F\right)^2 = 1 \,, \quad \left((-1)^{F_L}\right)^2 = 1 \,, \quad \mathcal{P}^2 = 1 \,, \quad \mathcal{T}^2 = 1 \,, \\
& (-1)^F (-1)^{F_L} = (-1)^{F_L} (-1)^F \,, \\
& (-1)^F \mathcal{P} = \mathcal{P} (-1)^F \,, \\
& \mathcal{P} (-1)^{F_L} = (-1)^F (-1)^{F_L} \mathcal{P} \,, \\
& \mathcal{T}(-1)^F = (-1)^F \mathcal{T} \,, \quad (\mathcal{TP})^2 = 1 \,, \quad \mathcal{T}(-1)^{F_L} = (-1)^{F_L}(-1)^F \mathcal{T} \,.
\end{aligned}
\tag{2.12}
$$

The unitary symmetries realize a $D_4 = \mathbb{Z}_2^L \times \mathbb{Z}_2^R \rtimes \mathbb{Z}_2^{\mathcal{P}}$. One way to see that is to replace the generator $(-1)^F$ by $(-1)^{F_R} = (-1)^{F_L}(-1)^F$ and then the relations (2.12) become

$$
\begin{aligned}
\text{NSNS}: \quad & \left((-1)^{F_R}\right)^2 = 1 \,, \quad \left((-1)^{F_L}\right)^2 = 1 \,, \quad \mathcal{P}^2 = 1 \,, \\
& (-1)^{F_R} (-1)^{F_L} = (-1)^{F_L} (-1)^{F_R} \,, \\
& \mathcal{P} (-1)^{F_L} = (-1)^{F_R} \mathcal{P} \,, \\
& \mathcal{T}(-1)^{F_R} = (-1)^{F_L} \mathcal{T} \,, \quad (\mathcal{TP})^2 = 1 \,.
\end{aligned}
\tag{2.13}
$$

Alternatively, we can use the generators $r = (-1)^{F_L}\mathcal{P}$ and $s = (-1)^{F_R}$ and note that they satisfy $s^2 = r^4 = (sr)^2 = 1$. We see that the symmetry is realized linearly in the NSNS Hilbert space.

The momentum operator $P$ obeys[12]

$$
\begin{aligned}
\text{NSNS}: \quad & e^{2\pi i P} = (-1)^F \,, \\
& (-1)^{F_L} P(-1)^{F_L} = P \,, \quad (-1)^{F_R} P(-1)^{F_R} = P \,, \quad \mathcal{P}P\mathcal{P}^{-1} = -P \,, \quad \mathcal{T}P\mathcal{T}^{-1} = -P \,.
\end{aligned}
\tag{2.14}
$$

Finally, the operators $\mathcal{T}' = \mathcal{T}(-1)^{F_L}$ and $\mathcal{P}' = \mathcal{P}(-1)^{F_L}$ satisfy

$$(\mathcal{P}')^2 = (-1)^F \quad , \quad (\mathcal{T}')^2 = (-1)^F \,. \tag{2.15}$$

---

[12] Note our notation: $\mathcal{P}$ is parity, while $P$ is the momentum.

## 2.2 RR Hilbert space and the projective algebra

In the RR theory, the fermions obey the boundary condition (2.6) with $\nu_L = \nu_R = 0$. As a result, we have a pair of Majorana zero modes $\chi_{L,0}, \chi_{R,0}$ obeying

$$\{\chi_{L,0}, \chi_{L,0}\} = 1 , \quad \{\chi_{R,0}, \chi_{R,0}\} = 1 , \quad \{\chi_{L,0}, \chi_{R,0}\} = 0 . \tag{2.16}$$

We can choose a basis for the two RR ground states so that $\chi_{L,0}, \chi_{R,0}$ are realized as $\frac{1}{\sqrt{2}}\sigma^x, \frac{1}{\sqrt{2}}\sigma^z$, respectively.[13] It follows that $(-1)^F, (-1)^{F_L}, \mathcal{P}, \mathcal{T}$ can be taken to be

RR :

$$(-1)^F = \sigma^y , \quad (-1)^{F_L} = \sigma^z , \quad \mathcal{P} = \frac{1}{\sqrt{2}} \begin{pmatrix} 1 & 1 \\ 1 & -1 \end{pmatrix} , \quad \mathcal{T} = \frac{1}{\sqrt{2}} \begin{pmatrix} 1 & -1 \\ -1 & -1 \end{pmatrix} \mathcal{K} , \tag{2.18}$$

where $\mathcal{K}$ is the complex conjugation. For this choice of the normalization, we have the following algebra in the RR Hilbert space

RR :

$$\begin{aligned}
&\left((-1)^F\right)^2 = 1 , \quad ((-1)^{F_L})^2 = 1 , \quad \mathcal{P}^2 = 1 , \quad \mathcal{T}^2 = 1 , \\
&(-1)^F (-1)^{F_L} = -(-1)^{F_L} (-1)^F , \\
&(-1)^F \mathcal{P} = -\mathcal{P} (-1)^F , \\
&\mathcal{P} (-1)^{F_L} = -i(-1)^F (-1)^{F_L} \mathcal{P} , \\
&\mathcal{T}(-1)^F = (-1)^F \mathcal{T} , \quad (\mathcal{T}\mathcal{P})^2 = -1 , \quad \mathcal{T}(-1)^{F_L} = -i(-1)^{F_L}(-1)^F \mathcal{T} .
\end{aligned} \tag{2.19}$$

The algebra in the RR theory realizes that of the NSNS theory in (2.12) projectively. In particular, the unitary operators lead to a central extension of $D_4 = \mathbb{Z}_2^L \times \mathbb{Z}_2^R \rtimes \mathbb{Z}_2^{\mathcal{P}}$.[14]

As in (2.15), we can also consider $\mathcal{T}' = \mathcal{T}(-1)^{F_L}$ and $\mathcal{P}' = \mathcal{P}(-1)^{F_L}$. Now they satisfy

$$(\mathcal{P}')^2 = -i(-1)^F \qquad , \qquad (\mathcal{T}')^2 = i(-1)^F . \tag{2.20}$$

The minus sign in

$$(-1)^F(-1)^{F_L} = -(-1)^{F_L}(-1)^F \tag{2.21}$$

signals an anomaly of the $\mathbb{Z}_2 \times \mathbb{Z}_2^f$ symmetry. In a general 1+1d fermionic QFT, it is known that the 't Hooft anomaly of a $\mathbb{Z}_2 \times \mathbb{Z}_2^f$ symmetry is classified by the spin cobordism group $\mathrm{Hom}(\mathrm{Tors}\,\Omega_3^{\mathrm{Spin}}(B\mathbb{Z}_2), U(1)) = \mathbb{Z}_8$ [62, 52, 63, 33, 64]. For a single Majorana fermion, the chiral

---

[13]Our conventions are

$$\sigma^x = \begin{pmatrix} 0 & 1 \\ 1 & 0 \end{pmatrix} \qquad , \qquad \sigma^y = \begin{pmatrix} 0 & -i \\ i & 0 \end{pmatrix} \qquad , \qquad \sigma^z = \begin{pmatrix} 1 & 0 \\ 0 & -1 \end{pmatrix} . \tag{2.17}$$

[14]We can redefine $(-1)^F \to i(-1)^F$ so that all the phases in the algebra are $\pm 1$. This shows that the extension is by $\mathbb{Z}_2$. However, we choose not to do this so that $(-1)^F$ remains an order 2 operator, i.e., $\left((-1)^F\right)^2 = 1$.

fermion parity $(-1)^{F_L}$ realizes the $\nu = 1 \in \mathbb{Z}_8$ anomaly. It was argued in [42] (see also [36, 43]) that the minus sign in (2.21) is a consequence of the anomaly when $\nu \in \mathbb{Z}_8$ is odd.

Including the spatial parity symmetry $\mathcal{P}$ we have a more general anomaly. The algebra (2.19) realizes $D_4$ projectively, which includes

$$
\begin{aligned}
(-1)^F \mathcal{P} &= -\mathcal{P}(-1)^F \,, \\
\mathcal{P}\,(-1)^{F_L} &= -i(-1)^F\,(-1)^{F_L}\,\mathcal{P}\,.
\end{aligned}
\tag{2.22}
$$

Note that in terms of $\mathcal{P}' = \mathcal{P}(-1)^{F_L}$, the first of these simplifies as

$$
(-1)^F \mathcal{P}' = \mathcal{P}'(-1)^F \,.
\tag{2.23}
$$

showing that there is no anomaly involving $(-1)^F$ and $\mathcal{P}'$.

It is known that this $D_4$ also has a mod 8 anomaly, classified by $\mathrm{Hom}(\mathrm{Tors}\,\Omega_3^{\mathrm{DPin}}(pt), U(1)) = \mathbb{Z}_8$, where DPin is the double Pin group that contains both $\mathrm{Pin}^{\pm}$ [64, 65]. Indeed, $\mathcal{P}$ and $(-1)^{F_L}\mathcal{P}$ respectively can be used to define a $\mathrm{Pin}^+$ and a $\mathrm{Pin}^-$ structure. Similarly, the symmetry generated by $(-1)^F, (-1)^{F_L}, \mathcal{T}$ has the same mod 8 anomaly. If we just focus on the symmetry generated by $(-1)^F$ and $\mathcal{P}$, this symmetry group has a mod 2 anomaly classified by $\mathrm{Hom}(\mathrm{Tors}\,\Omega_3^{\mathrm{Pin}^+}(pt), U(1)) = \mathbb{Z}_2$ [33]. This anomaly is detected by the minus sign in the algebra between $(-1)^F$ and $\mathcal{P}$ in the first line in (2.22). In contrast, there is no anomaly in the symmetry generated by $(-1)^F$ and $\mathcal{T}$, since $\mathrm{Hom}(\mathrm{Tors}\,\Omega_3^{\mathrm{Pin}^-}(pt), U(1))$ is trivial. Indeed, $(-1)^F$ commutes with $\mathcal{T}$.[15]

All this is consistent with the fact that the mass term (2.5) preserves $(-1)^F, \mathcal{P}' = \mathcal{P}(-1)^{F_L}$, and $\mathcal{T}$ and therefore these symmetries are anomaly free.

We can also rewrite the algebra by replacing the generator $(-1)^F$ by $(-1)^{F_R} = i(-1)^{F_L}(-1)^F$.[16] Then, in the normalizations in (2.18), we have $(-1)^{F_R} = \sigma^x$. And the relations (2.19) become

$$
\begin{aligned}
\mathrm{RR}: \quad & \left((-1)^{F_R}\right)^2 = 1\,, \quad \left((-1)^{F_L}\right)^2 = 1\,, \quad \mathcal{P}^2 = 1\,, \\
& (-1)^{F_R}\,(-1)^{F_L} = -(-1)^{F_L}\,(-1)^{F_R}\,, \\
& \mathcal{P}\,(-1)^{F_L} = (-1)^{F_R}\,\mathcal{P}\,, \\
& \mathcal{T}(-1)^{F_R} = -(-1)^{F_L}\mathcal{T}\,, \quad (\mathcal{T}\mathcal{P})^2 = -1\,.
\end{aligned}
\tag{2.24}
$$

It is then clear that the extension is by $\mathbb{Z}_2$.

The momentum operator $P$ obeys

$$
\begin{aligned}
\mathrm{RR}: \quad & e^{2\pi i P} = 1\,, \\
& (-1)^{F_L} P (-1)^{F_L} = P\,, \quad (-1)^{F_R} P (-1)^{F_R} = P\,, \quad \mathcal{P} P \mathcal{P}^{-1} = -P \quad \mathcal{T} P \mathcal{T}^{-1} = -P\,.
\end{aligned}
\tag{2.25}
$$

---

[15]We thank S. Seifnashri for discussions on this point.
[16]The inverse relation is $(-1)^F = i(-1)^{F_R}(-1)^{F_L} = -i(-1)^{F_L}(-1)^{F_R}$.

## 2.3 NSR Hilbert space and the mod 8 anomaly

Let us discuss the NSR Hilbert space. The discussion in the RNS Hilbert space is similar. In the NSR theory, the fermions obey the boundary conditions in (2.6) with $\nu_L = \frac{1}{2}, \nu_R = 0$. Parity $\mathcal{P}$ is not a symmetry in either NSR or RNS; rather, it exchanges the two Hilbert spaces. The NSR Hilbert space can be obtained from the NSNS theory by a $(-1)^{F_R}$ twist, or from the RR theory by a $(-1)^{F_L}$ twist.

There is a single fermion zero mode $\chi_{R,0}$ obeying $\{\chi_{R,0}, \chi_{R,0}\} = 1$. We can canonically quantize the theory by choosing a ground state $|\Omega\rangle_{NSR}$ that obeys

$$\begin{aligned}
\chi_{R,0}|\Omega\rangle_{NSR} &= |\Omega\rangle_{NSR}\,, \\
\chi_{R,\tilde{r}}|\Omega\rangle_{NSR} &= 0\,, \quad \tilde{r} = 1, 2, \cdots, \\
\chi_{L,r}|\Omega\rangle_{NSR} &= 0\,, \quad r = \frac{1}{2}, \frac{3}{2}, \cdots.
\end{aligned} \tag{2.26}$$

Because of the odd number of fermion zero modes, the NSR theory does not have the operators $(-1)^F$ or $(-1)^{F_R}$. While they are automorphisms of the operator algebra and are symmetries of the Lagrangian, they do not act in the Hilbert space. See footnote 1.

However, the internal symmetry operator $(-1)^{F_L}$ and the momentum operator $P$ do exist. They act on the NSR ground state as

$$(-1)^{F_L}|\Omega\rangle_{NSR} = |\Omega\rangle_{NSR}\,, \quad P|\Omega\rangle_{NSR} = -\frac{1}{16}|\Omega\rangle_{NSR}\,. \tag{2.27}$$

It follows that they obey the algebra:

$$\text{NSR}: \quad \left((-1)^{F_L}\right)^2 = 1\,, \quad e^{2\pi i P} = (-1)^{F_L} e^{-\frac{2\pi i}{16}}\,. \tag{2.28}$$

The second relation indicates an 't Hooft anomaly of the chiral fermion parity $(-1)^{F_L}$, as we will discuss soon.

Alternatively, one can quantize the NSR theory on a two-dimensional ground space so that $\chi_{R,0}$ is realized as $\frac{1}{\sqrt{2}}\sigma^z$ so that the fermion parity $(-1)^F$ is a symmetry. However, as we discuss in Appendix A, neither of these options match a path integral description. Again, see footnote 1.

The anomaly of $\mathbb{Z}_2 \times \mathbb{Z}_2^f$ labeled by $\nu \in \text{Hom}(\text{Tors}\,\Omega_3^{\text{Spin}}(B\mathbb{Z}_2), U(1)) = \mathbb{Z}_8$ can be detected from the eigenvalue of the momentum operator $P$ in the NSR Hilbert space. More specifically, the eigenvalue of $P$ in the $\mathbb{Z}_2$-twisted Hilbert space (from the NSNS Hilbert space) takes values in [42, 43]

$$P \in \frac{\nu}{16} + \frac{\mathbb{Z}}{2}\,. \tag{2.29}$$

This is the fermionic version of the spin selection rule in [68, 79, 86–88]. In our case, the NSR Hilbert space is the $(-1)^{F_L}$-twisted Hilbert space. The anomaly of the chiral fermion parity $(-1)^{F_L}$ of a free Majorana CFT corresponds to $\nu = 1$, consistent with (2.28) and (2.29).

In the RNS theory, the fermions obey the boundary condition (2.6) with $\nu_L = 0, \nu_R = \frac{1}{2}$. The RNS theory can be quantized in a similar way by choosing a ground state such that $\chi_{L,0}|\Omega\rangle_{RNS} = $

$|\Omega\rangle_{\text{RNS}}$. It is important that $(-1)^F, (-1)^{F_{\text{L}}}$ are not symmetries in the RNS Hilbert space, but $(-1)^{F_{\text{R}}}$ is. We have the following algebra in the RNS theory:

$$\text{RNS}: \quad \left((-1)^{F_{\text{R}}}\right)^2 = 1, \quad e^{2\pi i P} = (-1)^{F_{\text{R}}} e^{\frac{2\pi i}{16}}. \tag{2.30}$$

The $(-1)^{F_{\text{R}}}$ symmetry realizes the $\nu = -1 \in \mathbb{Z}_8$ anomaly, consistent with (2.29) and (2.30).

# 3  Majorana chain

Consider $L$ Majorana fermions $\chi_\ell$ obeying

$$\{\chi_\ell, \chi_{\ell'}\} = 2\delta_{\ell,\ell'}. \tag{3.1}$$

A concrete Hamiltonian to keep in mind is the nearest neighbor Hamiltonian

$$H = \frac{i}{2} \sum_{\ell=1}^{L} \chi_{\ell+1}\chi_\ell \quad , \quad \chi_{\ell+L} = \chi_\ell. \tag{3.2}$$

We can also add the four Fermi term $\sum_\ell \chi_\ell \chi_{\ell+1} \chi_{\ell+2} \chi_{\ell+3}$, which preserves the same symmetries. (See footnote 3.) However, we emphasize that for most of our discussion the particular form of the Hamiltonian will not matter.

The Hilbert space of the problem is a realization of the Clifford algebra (3.1). We take it to be a single irreducible representation of this algebra. Its dimension is $2^{\frac{L}{2}}$ for $L$ even and it is $2^{\frac{L-1}{2}}$ for $L$ odd.

There is a well known relation between the Clifford algebra and the orthogonal groups. Consider the transformation

$$\chi_\ell \to \sum_{\ell'} R_{\ell,\ell'} \chi_{\ell'}. \tag{3.3}$$

In order to preserve (3.1), $R_{\ell,\ell'}$ should be an $O(L)$ matrix. When this transformation is an inner automorphism, there is an operator $\mathcal{O}$ such that

$$\mathcal{O}\chi_\ell \mathcal{O}^{-1} = \sum_{\ell'} R(\mathcal{O})_{\ell,\ell'} \chi_{\ell'}. \tag{3.4}$$

For even $L$, we can find such an $\mathcal{O}$ for every $O(L)$ matrix $R_{\ell,\ell'}$. The hermitian operators $\frac{i}{2}\chi_\ell \chi_{\ell'}$ generate the $SO(L)$ subgroup and the remaining "reflection" transformation can be taking to be generated by $\chi_1$. For odd $L$, only the $SO(L)$ subgroup corresponds to an inner automorphism. There is no operator constructed out of $\chi_\ell$ that implements a transformation $R_{\ell,\ell'}$ with $\det R_{\ell,\ell'} = -1$.

Another significant fact is that the operators realizing the $O(L)$ (or $SO(L)$) transformations realize these groups projectively. The Hilbert space is in a spinor representation of $Spin(L)$. More precisely, for even $L$ we have the two spinor representations each with dimension $2^{\frac{L-2}{2}}$. And for odd $L$ we have the single spinor representation with dimension $2^{\frac{L-1}{2}}$.

So far the fermions $\chi_\ell$ were unrelated to each other. In the physical problem, they are arranged on a closed chain with $L$ sites and there is a Majorana fermion on each site with a clear notion of locality. This restricts the global symmetry of the problem to be smaller than the $O(L)$ (or $SO(L)$) automorphism. In particular, we will be interested in the transformations $T$ and $\mathsf{G}$ corresponding to translation and fermion number, respectively. We will also be interested in the parity transformation $\mathcal{P}$ and the time-reversal transformation $\mathcal{T}$, which is anti-unitary.

## 3.1 Translation and fermion parity symmetries

### 3.1.1 Even $L$

We would like to realize these operators on the Hilbert space. Let us start with even $L = 2N$ and focus on the translation $T$ and the fermion parity $\mathsf{G}$, which act on the fermion operators as

$$
\begin{aligned}
T : & \quad \chi_\ell \to T\chi_\ell T^{-1} = \sum_{\ell'} R(T)_{\ell,\ell'} \chi_{\ell'} = \chi_{\ell+1} \\
\mathsf{G} : & \quad \chi_\ell \to \mathsf{G}\chi_\ell \mathsf{G}^{-1} = \sum_{\ell'} R(\mathsf{G})_{\ell,\ell'} \chi_{\ell'} = -\chi_\ell ,
\end{aligned}
\tag{3.5}
$$

where $\ell \sim \ell + L$.

The algebra satisfied by these operators is

$$
R(\mathsf{G})^2 = 1 , \quad R(T)^L = 1 , \quad R(\mathsf{G})\, R(T) = R(T)\, R(\mathsf{G}) .
\tag{3.6}
$$

We would like to see how these operators are realized on the Hilbert space. Clearly,

$$
\begin{aligned}
\det R(T)_{\ell,\ell'} &= -1 , \\
\det R(\mathsf{G})_{\ell,\ell'} &= +1 .
\end{aligned}
\tag{3.7}
$$

This means that $\mathsf{G}$ corresponds to an $SO(L)$ transformation and it is realized as a product of an even number of fermions $\chi_\ell$, while $T$ involves the reflection in $O(L)$ and therefore it is realized as a product of an odd number of fermions. As we will see, the operators $\mathsf{G}$, $T$ realize the relations corresponding to (3.6) projectively. But since the $O(L)$ action on the operators becomes a $Pin(L)$ action on the states, the projective phases are only $\pm 1$.

Let us find an explicit expressions for these symmetry transformations in terms of the fermion fields.[17] Using the way $\mathsf{G}$ and $T$ act on the fermions (3.5) and imposing that they are unitary transformations determines them only up to an overall phase. First, we will find particular expressions for $\mathsf{G}$ and $T$ by assuming there is no additional phase in these expressions in terms of $\chi_\ell$. Second, in Section 3.3, we will see how we can redefine them by phases, and compare them to the continuum operators.

The $\mathbb{Z}_2$ generator $\mathsf{G}$ can be written in terms of the fermion fields as

$$
\mathsf{G} = \chi_1 \chi_2 \cdots \chi_{2N} .
\tag{3.8}
$$

---

[17]We thank M. Cheng, S. Ryu, and S. Seifnashri for discussions on the local expressions for the translation operator.

As written, it satisfies

$$\mathsf{G}^2 = (-1)^N \, . \tag{3.9}$$

We could have redefined $\mathsf{G}$ by a factor of $i^N$ such that it squares to one, as one would expect of a $\mathbb{Z}_2$ generator. As discussed above, we prefer not to do it at this stage. In Sections 3.3 and 4.2, we will rescale it to compare with the continuum parity operator $(-1)^F$.

We would like to write $\mathsf{G}$ as a product of local factors. One option is to view (3.8) as such a product, but then, the local factors $\chi_\ell$ do not act on the fermions as the total unitary operator $\mathsf{G}$, because $\chi_\ell \chi_{\ell'} \chi_\ell^{-1} = -\chi_{\ell'}$ if $\ell \neq \ell'$. This can be fixed by using a Klein transformation. We write $\mathsf{G}$ as

$$\mathsf{G} = \mathsf{g}_1 \mathsf{g}_2 \cdots \mathsf{g}_{2N} \, , \tag{3.10}$$

where $\mathsf{g}_\ell$ is defined as

$$\mathsf{g}_\ell = \mathsf{G}\chi_\ell = (-1)^\ell \chi_1 \cdots \chi_{\ell-1} \chi_{\ell+1} \cdots \chi_{2N} \, . \tag{3.11}$$

It acts on the fermions as

$$\mathsf{g}_\ell \, \chi_{\ell'} \, \mathsf{g}_\ell^{-1} = \begin{cases} -\chi_\ell \, , & \text{if } \ell = \ell' \, , \\ \chi_{\ell'} \, , & \text{if } \ell \neq \ell' \, , \end{cases} \tag{3.12}$$

and obeys

$$\{\mathsf{g}_\ell, \mathsf{g}_{\ell'}\} = 2(-1)^{N+1}\delta_{\ell,\ell'} \, . \tag{3.13}$$

Note that $\mathsf{G}$ is a boson, as it is constructed out of an even number of fermions, but the factors $\chi_\ell$ or $\mathsf{g}_\ell$ are fermions. (Because of that, they do not satisfy the locality condition (1.15).)

In the context of symmetries, it is common to use the phrase "on-site" symmetry action. Often, it refers to the action on the local Hilbert spaces $\mathcal{H}_j$. We will discuss the action on the local Hilbert spaces in Section 5 and in particular, we will see around (5.7) that $\mathsf{G}$ can be written as a product of local factors $\chi_{2j-1}\chi_{2j}$ acting linearly in $\mathcal{H}_j$. (Note that this is unlike the discussion around (1.14).)

Here, we would like to comment on another notion of "on-site" action, which refers to how the symmetry operator acts on the local fields at different sites. Unlike the discussion in Section 5, here the word "site" refers to the site of the lattice, labeled by $\ell = 1, \cdots, L$. Even though $\mathsf{G}$ can be written as a product of factors $\chi_\ell$ or $\mathsf{g}_\ell$ as in (3.8) and (3.10), its action on the fields is not the standard on-site action for the following reasons. In both expressions, the factors $\chi_\ell$ or $\mathsf{g}_\ell$ at different sites anticommute with each other. In (3.8), the factor $\chi_\ell$ acts on $\chi_{\ell'}$ with $\ell \neq \ell'$ nontrivially, i.e., $\chi_\ell \chi_{\ell'} \chi_\ell^{-1} = -\chi_{\ell'}$ . In (3.10), while $\mathsf{g}_\ell$ acts locally on the fermion fields, i.e., it commutes with $\chi_\ell$ with $\ell \neq \ell'$, it is not a local operator; in fact, it is the product of all the fermions except for the one at site $\ell$. In any case, we cannot write $\mathsf{G}$ as a product of local operators that implement the symmetry transformation on local fields, i.e., they commute with operators at different sites.

Before we give the explicit expression for $T$, we can readily derive a relation between $T$ and $\mathsf{G}$ using (3.10) [57, 15]:

$$T\mathsf{G}T^{-1} = \chi_2\chi_3 \cdots \chi_{2N}\chi_1 = -\mathsf{G} \, . \tag{3.14}$$

Equivalently, $\mathsf{G}T = -T\mathsf{G}$ stating that $T$ is constructed out of an odd number of fermions. This is consistent with the fact that $\det R(T)_{\ell,\ell'} = -1$.

Next, we write the translation operator $T$ in terms of the fermions. Again, the action (3.5) does not determine the overall phase normalization of $T$, and below we will make a particular choice. In Section 3.3, we will rescale it to $T_{RR}$ and compare with the continuum operators.

We can think of the translation as shifting the fermions in steps. First, a rotation between $L$ and $L-1$, then a rotation between $L-1$ and $L-2$, etc. Finally we need another factor to arrange the signs. Explicitly, we write the translation operator as[18]

$$T = \frac{1}{2^{\frac{L-1}{2}}} \chi_1 (1 + \chi_1\chi_2)(1 + \chi_2\chi_3) \cdots (1 + \chi_{L-1}\chi_L) \,. \tag{3.16}$$

It is important that $T$ is written as a product of local factors,

$$t_\ell = \frac{1}{\sqrt{2}}(1 + \chi_\ell\chi_{\ell+1}) \,, \quad t_\ell^8 = 1 \,. \tag{3.17}$$

$t_\ell$ is a $\frac{\pi}{2}$ rotation in the plane labeled by $(\ell, \ell+1)$, which is indeed an order 8 rotation. One finds[19] that [90]

$$T^L = e^{\frac{i\pi L(L-2)}{8}} = (-1)^{\frac{N(N-1)}{2}} = \begin{cases} -1, & \text{if } L = 4, 6 \bmod 8 \\ +1, & \text{if } L = 0, 2 \bmod 8 \end{cases} \,. \tag{3.18}$$

In Section 3.3, we will discuss the allowed redefinition of the operator $T$ that are compatible with locality.

Similarly, we define[20]

$$\hat{T} = \frac{1}{2^{\frac{L-1}{2}}} \chi_1 (1 - \chi_1\chi_2)(1 - \chi_2\chi_3) \cdots (1 - \chi_{L-1}\chi_L) = (-1)^N T\mathsf{G} \,, \tag{3.19}$$

which is also a product of local factors $t_\ell^{-1} = -t_\ell^3 = \frac{1}{\sqrt{2}}(1 - \chi_\ell\chi_{\ell+1})$, or $\chi_\ell t_\ell^{-1} = \frac{1}{\sqrt{2}}(\chi_\ell - \chi_{\ell+1})$. The significance of $\hat{T}$ will be clear below.

---

[18]The translation operator for even $L$ can be equivalently written as

$$\begin{aligned} T &= \frac{1}{2^{\frac{L-1}{2}}} \chi_2\chi_3 \cdots \chi_L (1 - \chi_1\chi_2)(1 - \chi_2\chi_3) \cdots (1 - \chi_{L-1}\chi_L) \\ &= \frac{1}{2^{\frac{L-1}{2}}} (\chi_1 + \chi_2)(\chi_2 + \chi_3) \cdots (\chi_{L-1} + \chi_L) \,. \end{aligned} \tag{3.15}$$

These alternative expressions are convenient for different purposes. (Note that in these forms, $T$ does not satisfy the locality condition (1.15).)

[19]Many of the calculations in this paper involve long and painful manipulations using fermions. The following procedure simplifies them. It is often easy to find the answer up to an overall phase. Since we manipulate real fermions, that phase is a sign. The reason the calculations are tedious is that they involve a large number of such transpositions of fermions that can lead to some signs. The number of such transpositions grows as a power of $L$, say $L^3$. Then, the overall sign is $(-1)^{a_3 L^3 + a_2 L^2 + a_1 L + a_0}$ with some constants $a_I$. These constants can be determined by performing the explicit calculation for some small values of $L$, thus deriving the answer for all $L$. Some of these calculations are done with the help of the Mathematica package of [89].

[20]We normalize $\hat{T}$ so that $\hat{T} = \mathcal{T}T\mathcal{T}^{-1}$ where $\mathcal{T}$ is the time-reversal symmetry defined in Section 3.2.1.

To summarize, we find the following algebra for even $L = 2N$:

$$\mathsf{G}^2 = (-1)^N, \quad T^L = (-1)^{\frac{N(N-1)}{2}}, \quad \mathsf{G}T = -T\mathsf{G}. \tag{3.20}$$

They realize the relations (3.6) projectively. Note that all the projective phases are $\pm 1$. This is consistent with our statement above about lifting the $O(L)$ transformations to $Pin(L)$.

### 3.1.2 Even $L$ with a fermion parity defect

Let us introduce a $\mathsf{G}$ defect. A concrete Hamiltonian to keep in mind is

$$H_\mathsf{G} = \frac{i}{2} \sum_{\ell=1}^{L-1} \chi_{\ell+1}\chi_\ell - \frac{i}{2}\chi_1\chi_L, \tag{3.21}$$

where the defect is in the link connecting $(L, 1)$. We use the subscript $\mathsf{G}$ for the Hamiltonian and symmetry operators in the system with a $\mathsf{G}$ defect. However, we emphasize that for most of our discussion the particular form of the Hamiltonian will not matter. Note that the defect can be moved to other links, e.g., to the link $(1, 2)$, by conjugating $H_\mathsf{G}$ by a local $\mathsf{G}$ transformation, $\chi_1$ or $\mathsf{g}_1$.

Let us determine the symmetry operators of the theory with the defect. We use the same fermion parity operator $\mathsf{G}$ as in (3.10), because it commutes with $H_\mathsf{G}$.[21] On the other hand, instead of (3.5), the translation operator now acts on the fermion fields as

$$T_\mathsf{G}: \quad \chi_\ell \to T_\mathsf{G}\chi_\ell T_\mathsf{G}^{-1} = \sum_{\ell'} R(T_\mathsf{G})_{\ell,\ell'}\chi_{\ell'} = \begin{cases} \chi_{\ell+1} & \ell = 1, 2, \cdots, L-1 \\ -\chi_1 & \ell = L \end{cases} \tag{3.22}$$

The algebra satisfied by these operators is

$$R(\mathsf{G})^2 = 1, \quad R(T_\mathsf{G})^L = R(\mathsf{G}), \quad R(\mathsf{G})\, R(T_\mathsf{G}) = R(T_\mathsf{G})\, R(\mathsf{G}). \tag{3.23}$$

In contrast to the case without the defect (3.7), now we have

$$\begin{aligned} \det R(T_\mathsf{G})_{\ell,\ell'} &= +1, \\ \det R(\mathsf{G})_{\ell,\ell'} &= +1. \end{aligned} \tag{3.24}$$

This means that the twisted translation operator is an $SO(L)$ transformation and is constructed out of an even number of fermions, i.e., it is bosonic.

Let us write $T_\mathsf{G}$ in terms of the fermion fields. Again, (3.22) does not determine its phase normalization, and we will make an arbitrary choice below. Later, in Section 3.3, we will rescale it to $T_{\text{NSNS}}$ and compare it to the continuum operators. Its action in (3.22) means that we should

---

[21] We do not write $\mathsf{G}_\mathsf{G}$ because it is the same as $\mathsf{G}$.

multiply $T$ by an operator that maps $\chi_1 \to -\chi_1$ and leaves the other fermions unchanged, i.e., we should multiply it by $\mathsf{g}_1 = -\chi_2\chi_3\cdots\chi_L$. Therefore, we take [91][22]

$$T_{\mathsf{G}} = (-1)^N \mathsf{g}_1 T = \frac{1}{2^{\frac{L-1}{2}}}(1 - \chi_1\chi_2)(1 - \chi_2\chi_3)\cdots(1 - \chi_{L-1}\chi_L)\,. \tag{3.26}$$

Using $T\mathsf{g}_\ell T^{-1} = -\mathsf{g}_{\ell+1}$ and $T^L = (-1)^{\frac{N(N-1)}{2}}$, we have

$$T_{\mathsf{G}}^L = (-1)^{\frac{N(N+1)}{2}}\chi_1\cdots\chi_L = (-1)^{\frac{N(N+1)}{2}}\mathsf{G}\,. \tag{3.27}$$

We summarize that these operators in the system with even $L = 2N$ and with a defect satisfy

$$\mathsf{G}^2 = (-1)^N\,, \quad T_{\mathsf{G}}^L = (-1)^{\frac{N(N+1)}{2}}\mathsf{G}\,, \quad \mathsf{G}T_{\mathsf{G}} = T_{\mathsf{G}}\mathsf{G}\,. \tag{3.28}$$

They realize the relations (3.23) projectively. As for the problem without the defect, all the projective phases are $\pm 1$. Later, we will see how we can remove these phases.

### 3.1.3 Odd $L$ and a translation symmetry defect

Next, we repeat this discussion for odd $L = 2N + 1$. As in footnote 1, unlike the case of even $L$, now the operator

$$\mathcal{C} = \chi_1\chi_2\cdots\chi_L \tag{3.29}$$

commutes with all the operators in the theory. Since it is central, it can be taken to be a c-number. Since

$$\mathcal{C}^2 = (-1)^N, \tag{3.30}$$

the value of the c-number for fixed $N$ is $\pm i^N$ and the Hilbert space is characterized by that sign. Recall that for even $L$, an operator of the form (3.29) generated the $\mathsf{G}$ symmetry. Instead, now it is a c-number and as we will soon discuss, the theory does not have such a symmetry. For this reason, we distinguish $\mathcal{C}$ in (3.29) from the symmetry transformation $\mathsf{G}$.[23]

Again, we keep in mind two possible Hamiltonians $H$ in (3.2) and the one with a defect $H_{\mathsf{G}}$ in (3.21) with $L = 2N+1$. As above, we are only interested in the symmetries of these Hamiltonians rather than in their particular form.

---

[22]Alternatively, the translation operator for even $L$ with a defect can be written as

$$T_{\mathsf{G}} = \frac{(-1)^N}{2^{\frac{L-1}{2}}}(\chi_1 - \chi_2)(\chi_2 - \chi_3)\cdots(\chi_{L-1} - \chi_L)\chi_L\,. \tag{3.25}$$

(Note that in this forms, $T_{\mathsf{G}}$ does not satisfy the locality condition (1.15).)

[23]The discussion in footnote 1 was focused on a quantum mechanical system and no locality in space was important. Now, that we have a "space coordinate" $\ell$, we should address the question of locality. Should local operators commute or anticommute? Even though our system does not have a fermion parity symmetry generated by $\mathsf{G}$, $\mathsf{G}$ is an outer-automorphism of the symmetry algebra. Therefore, we can still assign a fermion parity value $\pm 1$ to every operator. Operators with fermion parity $+1$ are bosons and operators with fermion parity $-1$ are fermions. This labeling determines whether local operators commute or anticommute when they are separated. Note that in this case of odd $L$, $\mathcal{C}$ is a fermion, but it is a c-number. This does not lead to problems with locality because $\mathcal{C}$ is not a local operator.

We start by discussing the Hamiltonian $H$ in (3.2). We can still take $T$ to act on the fermion field as in (3.5). Unlike the case of even $L$, now $\det R(T)_{\ell,\ell'} = 1$, so this is an $SO(L)$ transformation. One might attempt to use the same expression for the translation operator (3.16) for odd $L = 2N + 1$. However, the $L$-th power of this translation operator is proportional to the central element $\mathcal{C}$. Instead, we define the translation operator for odd $L$ as[24]

$$T = \frac{1}{2^{\frac{L-1}{2}}}(1 - \chi_1\chi_2)(1 - \chi_2\chi_3)\cdots(1 - \chi_{L-1}\chi_L)\,. \tag{3.32}$$

Its $L$-th power is

$$T^L = e^{\frac{i\pi(L^2-1)}{8}} = (-1)^{\frac{N(N+1)}{2}} = \begin{cases} +1\,, & \text{if } L = 1, 7 \text{ mod } 8 \\ -1\,, & \text{if } L = 3, 5 \text{ mod } 8 \end{cases}\,, \tag{3.33}$$

which does not depend on $\mathcal{C}$. We will discuss possible phase redefinitions of $T$ that are compatible with locality in Section 3.3.

Next, we consider G. We would like it to act as in (3.5). Indeed, this action is a symmetry of the Hamiltonian (3.2). However, since $\det R(\mathsf{G})_{\ell,\ell'} = -1$, it cannot be realized on the Hilbert space. The odd $L$ problem does not have the $\mathbb{Z}_2$ symmetry generated by G. The operator algebra has an automorphism $\chi_\ell \to -\chi_\ell$ and this is a symmetry of the Hamiltonian. But this automorphism is an outer-automorphism.

### *Translation symmetry defect*

The odd $L = 2N+1$ theory that we have been discussing is closely related to the even $L = 2N$ theory. Their Hilbert spaces are the same and $\chi_\ell$ with $\ell = 1, 2, \cdots, 2N$ satisfy the same algebra. The additional fermion of the $L = 2N+1$ theory $\chi_{2N+1}$ can be constructed out of the $2N$ fermions as follows. We pick a c-number $\mathcal{C} = \pm i^N$ and use $\mathsf{G}(2N) = \chi_1\chi_2\cdots\chi_{2N}$ to define

$$\chi_{2N+1} = (-1)^N \mathcal{C} \mathsf{G}(2N)\,. \tag{3.34}$$

It is easy to check that it satisfies the correct anticommutation relations.

Let us express the specific Hamiltonian (3.2) of the $L = 2N+1$ theory in terms of the degrees of freedom of the $L = 2N$ theory. The Hamiltonians are related

$$H(2N + 1) = H(2N) + \frac{i}{2}\left[-\chi_1\chi_{2N} + (-1)^N \mathcal{C}\, \mathsf{G}(2N)\,(\chi_{2N} - \chi_1)\right]\,. \tag{3.35}$$

---

[24]This translation operator for odd $L = 2N + 1$ can be written equivalently as

$$\begin{aligned} T &= \frac{(-1)^N}{2^{\frac{L-1}{2}}}(\chi_1 - \chi_2)(\chi_2 - \chi_3)\cdots(\chi_{L-2} - \chi_{L-1})(\chi_{L-1} - \chi_L) \\ &= \frac{(-1)^N}{2^{\frac{L-1}{2}}}\mathcal{C}\chi_1(1 + \chi_1\chi_2)(1 + \chi_2\chi_3)\cdots(1 + \chi_{L-1}\chi_L)\,. \end{aligned} \tag{3.31}$$

(Note that the first expression does not satisfy the locality condition (1.15).) The second expression makes it manifest that it differs from (3.16) for even $L$ only by a c-number. See more discussions on the relation between the even and odd $L$ cases below.

Looking at (3.35), it is clear that it does not respect the translation symmetry and the internal symmetry of the $2N$ theory generated by $T(2N)$ and $\mathsf{G}(2N)$ respectively. However, it respects a new translation symmetry generated by

$$T(2N+1) = \frac{(-1)^N}{\sqrt{2}}\mathcal{C}T(2N)(1+\chi_{2N}\chi_{2N+1})\,. \tag{3.36}$$

This symmetry acts on the fermions as

$$\begin{aligned} T(2N+1)\chi_\ell T(2N+1)^{-1} &= \chi_{\ell+1} &\quad, &\qquad \ell = 1,2,\cdots,2N-1 \\ T(2N+1)\chi_{2N} T(2N+1)^{-1} &= (-1)^N\mathcal{C}\mathsf{G}(2N) = \chi_{2N+1} \\ T(2N+1)\chi_{2N+1} T(2N+1)^{-1} &= \chi_1\,, \end{aligned} \tag{3.37}$$

and satisfies $T(2N+1)^{2N+1} = (-1)^{\frac{N(N+1)}{2}}$. Hence, it is the correct translation operator of the $L = 2N+1$ theory.

As always in this paper, our discussion is more general than the specific Hamiltonian (3.35) and can be repeated for any Hamiltonian with the same symmetries, leading to the translation operator (3.36).

We would like to interpret the change in the Hamiltonian (3.35) as a defect. Written in terms of the $2N$ fermions, as in (3.35), this modification of the Hamiltonian does not look local. However, since $\mathsf{G}(2N)$ has simple anticommutation relations, $\{\chi_\ell, \mathsf{G}(2N)\} = 0$, the operator $\mathsf{G}$ and therefore also $\chi_{2N+1}$ as defined in (3.34) can be thought of as local operators and then the right-hand side in (3.35) looks local.

We suggest that the defect is associated with the translation symmetry.[25] To motivate this suggestion, note that since $T(2N)\mathsf{G}(2N) = -\mathsf{G}(2N)T(2N)$, a $T(2N)$-defect breaks the $\mathsf{G}(2N)$ symmetry. This is indeed the case in (3.35). Also, the translation symmetry is always violated in the presence of a defect and instead we have a new translation symmetry. In our case, it is $T(2N+1)$. This new translation symmetry satisfies another relation

$$T(2N+1)^{2N} = (-1)^{\frac{N(N+1)}{2}}T(2N+1)^{-1}\,. \tag{3.38}$$

Following our general picture, we see on the right-hand side the symmetry generator associated with the defect – the translation generator. Unlike defects associated with an internal symmetry, where the symmetry we use in constructing the defect is not modified by the defect, here it is not the original symmetry generator $T(2N)$, but the deformed one $T(2N+1)$. Finally, we also have a projective phase $(-1)^{\frac{N(N+1)}{2}}$.

As we end this discussion of odd $L = 2N+1$ as a translation defect in the even $L = 2N$ theory, we would like to add another comment. If instead of analyzing $H$ in (3.35), we had analyzed $H_{\mathsf{G}}$, we would have found that the twisted odd $L = 2N+1$ theory can be viewed as a $\hat{T} = (-1)^N T\mathsf{G}$ defect in the even $L = 2N$ twisted theory. This fact will be important below.

---

[25]We thank S. Seifnashri for useful discussions of translation defects in other systems.

## 3.2 Parity and time-reversal symmetries

Next, we discuss the parity and time-reversal transformations.

### 3.2.1 Even $L$

For even $L = 2N$, we define the parity $\mathsf{P}$ and the time-reversal $\mathcal{T}$ transformations as

$$\mathsf{P}: \qquad \chi_\ell \to \mathsf{P}\chi_\ell\mathsf{P}^{-1} = \sum_{\ell'} R(\mathsf{P})_{\ell,\ell'}\chi_{\ell'} = (-1)^\ell \chi_{-\ell}$$

$$\mathcal{T}: \qquad \chi_\ell \to \mathcal{T}\chi_\ell\mathcal{T}^{-1} = \sum_{\ell'} R(\mathcal{T})_{\ell,\ell'}\chi_{\ell'} = (-1)^{\ell+1}\chi_\ell \qquad (3.39)$$

$$\ell \sim \ell + L\,.$$

The expressions for $R(\mathsf{P})$ and $R(\mathcal{T})$ are motivated by the fact that they are symmetries of the Hamiltonian (3.2).[26] Clearly, we can redefine $\mathsf{P}$ or $\mathcal{T}$ by combining them with $\mathsf{G}$.

For even $L$, we have

$$\begin{aligned} \det R(\mathsf{P})_{\ell,\ell'} &= -1 \\ \det R(\mathcal{T})_{\ell,\ell'} &= (-1)^N\,. \end{aligned} \qquad (3.40)$$

The first line means that $\mathsf{P}$ involves the reflection in $O(L)$ and therefore it is realized as a product of an odd number of fermions. Since $\mathcal{T}$ is anti-unitary, its determinant does not tell us directly its relation to $Pin(L)$.

All together, the Hamiltonian (3.2) for even $L$ enjoys the translation $T$, fermion parity $\mathsf{G}$, parity $\mathsf{P}$, and time-reversal symmetry $\mathcal{T}$. The inclusion of the four Fermi term $\sum_\ell \chi_\ell\chi_{\ell+1}\chi_{\ell+2}\chi_{\ell+3}$ also preserves all these symmetries. (See footnote 3.) We can add to the Hamiltonian (3.2) a real mass term (compare with (2.5))

$$\frac{im}{2}\sum_{\ell=1}^L (-1)^\ell \chi_{\ell+1}\chi_\ell\,. \qquad (3.41)$$

It violates the symmetries $\mathsf{P}$ and $T$, but preserves the symmetries $\mathsf{G}$, $T\mathsf{P}$, $T^2$, and $\mathcal{T}$.

These symmetry operators form the following algebra when acting on the fermion operators

---

[26]Note that in the problem of $L$ decoupled real fermions in quantum mechanics, it is more natural to let $\mathcal{T}$ map $\chi_\ell \to \chi_\ell$ (or $\chi_\ell \to -\chi_\ell$), such that it commutes with the global $SO(L)$ symmetry. This is indeed the $\mathcal{T}$ transformation that was analyzed in [60, 42, 54]. In our case, the transformation is different because the Hamiltonian (3.2) couples the different $\chi_\ell$'s. We can also consider another parity transformation, which is not a symmetry of the Hamiltonian (3.2), $\mathsf{P}_0 : \chi_\ell \to \chi_{-\ell}$.

$$R(\mathsf{G})^2 = 1\,, \quad R(T)^L = 1\,, \quad R(\mathsf{P})^2 = 1\,, \quad R(\mathcal{T})^2 = 1\,,$$
$$R(\mathsf{G})\,R(T) = R(T)\,R(\mathsf{G})\,,$$
$$R(\mathsf{G})\,R(\mathsf{P}) = R(\mathsf{P})R(\mathsf{G})\,,$$
$$R(\mathsf{P})\,R(T) = R(\mathsf{G})\,R(T)^{-1}\,R(\mathsf{P})\,, \qquad\qquad (3.42)$$
$$R(\mathsf{G})\,R(\mathcal{T}) = R(\mathcal{T})\,R(\mathsf{G})\,,$$
$$R(\mathsf{P})\,R(\mathcal{T}) = R(\mathsf{P})R(\mathcal{T})\,,$$
$$R(\mathcal{T})\,R(T) = R(\mathsf{G})\,R(T)\,R(\mathcal{T})\,.$$

Note that because of the factor of $(-1)^\ell$ in the action of $R(\mathsf{P})$, the algebra of $R(T),\ R(\mathsf{P})$, and $R(\mathsf{G})$ is not merely $D_L$ acting on the indices $\ell$, but it is a $\mathbb{Z}_2$ extension of it by $R(\mathsf{G})$. The addition of the anti-unitary transformation $R(\mathcal{T})$ also extends the algebra in a nontrivial way, as can be seen in the last relation. Again, this algebra will be realized projectively on the Hilbert space.

Before we give an expression for $\mathsf{P}$ in terms of the fields, we can straightforwardly compute the conjugation of $\mathsf{G}$ and $T$ by $\mathsf{P}$ using $\mathsf{P}\chi_\ell\mathsf{P}^{-1} = (-1)^\ell\chi_{-\ell}$. We have

$$\mathsf{P}\mathsf{G}\mathsf{P}^{-1} = (-1)^N\chi_{2N-1}\chi_{2N-2}\cdots\chi_1\chi_{2N} = -\mathsf{G}\,,$$
$$\mathsf{P}T\mathsf{P}^{-1} = T^{-1}\chi_1\cdots\chi_L = -\mathsf{G}T^{-1}\,. \qquad\qquad (3.43)$$

As we said above, for even $L$, the parity operator is an inner automorphism and can be expressed in terms of the fermion fields:

$$\mathsf{P} = \begin{cases} \dfrac{1}{2^{\frac{N-1}{2}}}\chi_0(\chi_1 - \chi_{-1})(\chi_2 + \chi_{-2})\cdots(\chi_{N-1} - \chi_{-N+1})\chi_N\,, & \text{even } N \\[2mm] \dfrac{1}{2^{\frac{N-1}{2}}}\chi_0(\chi_1 - \chi_{-1})(\chi_2 + \chi_{-2})\cdots(\chi_{N-1} + \chi_{-N+1})\,, & \text{odd } N \end{cases} \qquad (3.44)$$

As a check, $\mathsf{P}$ has an odd number of fermions, which is consistent with the fact that it is not an $SO(L)$ transformation. It satisfies

$$\mathsf{P}^2 = (-1)^{\frac{N(N-1)}{2}} = e^{\frac{i\pi L(L-2)}{8}}\,. \qquad\qquad (3.45)$$

Using the time-reversal transformation (3.39), we have

$$\mathcal{T}\mathsf{G}\mathcal{T}^{-1} = (-1)^N\chi_1\chi_2\cdots\chi_{2N} = (-1)^N\mathsf{G}$$
$$\mathcal{T}\mathsf{P}\mathcal{T}^{-1} = \begin{cases} (-1)^{\frac{N}{2}+1}\mathsf{P}\,, & \text{even } N \\ (-1)^{\frac{N+1}{2}}\mathsf{P}\,, & \text{odd } N \end{cases} = -\mathsf{P}^{-1} \qquad (3.46)$$
$$\mathcal{T}T\mathcal{T}^{-1} = (-1)^N T\mathsf{G}$$

Since $\mathcal{T}$ is anti-unitary, when we write it in terms of the fundamental fields, we must include a factor of the complex conjugation operator $\mathcal{K}$. Hence, the expression for $\mathcal{T}$ in terms of the

fermions depends on a choice of basis. It is easy to see that there is a basis (which we will use in Section 5) such that $\mathcal{T}$ is just the complex conjugation $\mathcal{K}$ and therefore

$$\mathcal{T}^2 = 1 \,. \tag{3.47}$$

To summarize, the algebra of $T, \mathsf{G}, \mathsf{P}, \mathcal{T}$ on the Hilbert space of even $L$ is

$$
\begin{aligned}
&\mathsf{G}^2 = (-1)^N \,, \quad T^L = (-1)^{\frac{N(N-1)}{2}} \,, \quad \mathsf{P}^2 = (-1)^{\frac{N(N-1)}{2}} \,, \quad \mathcal{T}^2 = 1 \,, \\
&\mathsf{G}T = -T\mathsf{G} \,, \quad \mathsf{G}\mathsf{P} = -\mathsf{P}\mathsf{G} \,, \quad \mathsf{P}T\mathsf{P}^{-1} = -\mathsf{G}T^{-1} \\
&\mathcal{T}\mathsf{G} = (-1)^N \mathsf{G}\mathcal{T} \,, \quad (\mathcal{T}\mathsf{P})^2 = -1 \,, \quad \mathcal{T}T\mathcal{T}^{-1} = (-1)^N T\mathsf{G} \,,
\end{aligned}
\tag{3.48}
$$

which realizes (3.42) projectively.[27] Again all the projective phases are $\pm 1$, consistent with the lifting from $O(L)$ to $Pin(L)$.

### 3.2.2 Even $L$ with a fermion parity defect

We now move on to discuss the parity and time-reversal transformations in the presence of a $\mathsf{G}$ defect. A particular Hamiltonian is the one in (3.21).

The time-reversal transformation $\mathcal{T}$ acts the same as in the theory without the defect. In contrast, the original parity operator $\mathsf{P}$ does not leave the defect invariant. Rather, the following transformation is a symmetry of (3.21):

$$
\mathsf{P}_\mathsf{G} : \qquad \chi_\ell \to \mathsf{P}_\mathsf{G} \chi_\ell \mathsf{P}_\mathsf{G}^{-1} = \sum_{\ell'} R(\mathsf{P}_\mathsf{G})_{\ell,\ell'} \chi_{\ell'} = \begin{cases} (-1)^\ell \chi_{-\ell} & \ell = 1, 2, \cdots, L-1 \\ -\chi_L & \ell = L \end{cases}
\tag{3.49}
$$

which has

$$\det R(\mathsf{P}_\mathsf{G})_{\ell,\ell'} = +1 \,. \tag{3.50}$$

The fermion parity $\mathsf{G}$, time-reversal symmetry $\mathcal{T}$, and the new translation $T_\mathsf{G}$ in (3.22) and parity $\mathsf{P}_\mathsf{G}$ in (3.49) together form the following algebra when acting on the operators:

$$
\begin{aligned}
&R(\mathsf{G})^2 = 1 \,, \quad R(T_\mathsf{G})^L = R(\mathsf{G}) \,, \quad R(\mathsf{P}_\mathsf{G})^2 = 1 \,, \quad R(\mathcal{T})^2 = 1 \,, \\
&R(\mathsf{G})\,R(T_\mathsf{G}) = R(T_\mathsf{G})\,R(\mathsf{G}) \,, \\
&R(\mathsf{G})\,R(\mathsf{P}_\mathsf{G}) = R(\mathsf{P}_\mathsf{G})R(\mathsf{G}) \,, \\
&R(\mathsf{P}_\mathsf{G})\,R(T_\mathsf{G}) = R(\mathsf{G})\,R(T_\mathsf{G})^{-1}\,R(\mathsf{P}_\mathsf{G}) \,, \\
&R(\mathsf{G})\,R(\mathcal{T}) = R(\mathcal{T})\,R(\mathsf{G}) \,, \\
&R(\mathsf{P}_\mathsf{G})\,R(\mathcal{T}) = R(\mathsf{P}_\mathsf{G})R(\mathcal{T}) \,, \\
&R(\mathcal{T})\,R(T_\mathsf{G}) = R(\mathsf{G})\,R(T_\mathsf{G})\,R(\mathcal{T}) \,.
\end{aligned}
\tag{3.51}
$$

More explicitly, the new parity operator can be constructed out of $\mathsf{P}$ and $\mathsf{g}_L$, i.e.,

$$\mathsf{P}_\mathsf{G} = \mathsf{g}_L \mathsf{P} \,. \tag{3.52}$$

---

[27] Note that in terms of $\hat{T} = (-1)^N T\mathsf{G}$, we can write $\mathsf{P}T\mathsf{P}^{-1} = -\hat{T}^{-1}$ and $\mathcal{T}T\mathcal{T}^{-1} = \hat{T}$.

Finally, for the time-reversal transformation, we can still take $\mathcal{T} = \mathcal{K}$ without a modification.

On the Hilbert space with even $L$, these operators satisfy

$$
\begin{aligned}
&\mathsf{G}^2 = (-1)^N, \quad T_{\mathsf{G}}^L = (-1)^{\frac{N(N+1)}{2}} \mathsf{G}, \quad \mathsf{P}_{\mathsf{G}}^2 = (-1)^{\frac{N(N+1)}{2}}, \quad \mathcal{T}^2 = 1, \\
&\mathsf{G} T_{\mathsf{G}} = T_{\mathsf{G}} \mathsf{G}, \quad \mathsf{G} \mathsf{P}_{\mathsf{G}} = \mathsf{P}_{\mathsf{G}} \mathsf{G}, \quad \mathsf{P}_{\mathsf{G}} T_{\mathsf{G}} \mathsf{P}_{\mathsf{G}}^{-1} = (-1)^N \mathsf{G} T_{\mathsf{G}}^{-1}, \\
&\mathcal{T} \mathsf{G} = (-1)^N \mathsf{G} \mathcal{T}, \quad \mathcal{T} \mathsf{P}_{\mathsf{G}} \mathcal{T}^{-1} = \mathsf{P}_{\mathsf{G}}^{-1}, \quad \mathcal{T} T_{\mathsf{G}} \mathcal{T}^{-1} = T_{\mathsf{G}} \mathsf{G}.
\end{aligned} \tag{3.53}
$$

They realize the relations (3.51) projectively. As for the problem without the defect, all the projective phases are $\pm 1$. Actually, as we will see in (3.77), all these phases can be absorbed in redefinitions of the operators.

Finally, as in the discussion around (3.41), we can add to the Hamiltonian (3.2) a real mass term

$$
\frac{im}{2} \left( \sum_{\ell=1}^{L-1} (-1)^\ell \chi_{\ell+1} \chi_\ell - \chi_1 \chi_L \right). \tag{3.54}
$$

It violates the symmetries $\mathsf{P}_{\mathsf{G}}$ and $T_{\mathsf{G}}$, but preserves the symmetries $\mathsf{G}$, $T_{\mathsf{G}} \mathsf{P}_{\mathsf{G}}$, $T_{\mathsf{G}}^2$, and $\mathcal{T}$.

### 3.2.3 Odd $L$

Finally, we discuss the parity and time-reversal transformations for odd $L = 2N + 1$. Even though they are not symmetries of the particular Hamiltonian (3.2), they are still useful as we will see below.

The parity transformation of the even $L$ system

$$
\mathsf{P}: \qquad \chi_\ell \to \sum_{\ell'} R(\mathsf{P})_{\ell,\ell'} \chi_{\ell'} = (-1)^\ell \chi_{-\ell} = (-1)^\ell \chi_{L-\ell} \qquad, \qquad \ell = 0, \cdots, L-1 \tag{3.55}
$$

does not preserve the periodicity. Therefore, we will take the same expression for $\ell = 0, \cdots, L-1$ and then extend this action periodically for odd $L$. With this definition, $\det R(\mathsf{P})_{\ell,\ell'} = +1$ and hence it can be realized on the Hilbert space. Note however that $R(\mathsf{P})^2$ acts as $-1$ on $\chi_\ell$ for $\ell = 1, \cdots L-1$ and as $+1$ on $\chi_0 = \chi_L$ and hence with this definition $\mathsf{P}^2$ is not central and $\mathsf{P}$ cannot be viewed as a standard parity transformation.

Another significant fact is that this $\mathsf{P}$ transformation does not preserve the Hamiltonian and hence it is not a symmetry. Consider for example the untwisted Hamiltonian $H$ in (3.2). It is mapped under $\mathsf{P}$ as[28]

$$
\mathsf{P} H \mathsf{P}^{-1} = \frac{i}{2} \sum_{\ell=0}^{L-2} \chi_{-\ell} \chi_{-\ell-1} - \frac{i}{2} \chi_1 \chi_0 = \frac{i}{2} \sum_{\ell=1}^{L-1} \chi_{\ell+1} \chi_\ell - \frac{i}{2} \chi_1 \chi_L = H_{\mathsf{G}}. \tag{3.57}
$$

---

[28] It is important to first write the untwisted Hamiltonian as

$$
H = \frac{i}{2} \sum_{\ell=0}^{L-2} \chi_{\ell+1} \chi_\ell + \frac{i}{2} \chi_0 \chi_{L-1} \tag{3.56}
$$

so that all the fermion fields are in the range $\ell = 0, 1, \cdots, L-1$, and then apply (3.55).

We see that it maps the Hamiltonian $H$ of (3.2) to the same Hamiltonian with a G defect at the link $(L, 1)$ (equivalently, $(0, 1)$) $H_\mathsf{G}$ in (3.21) with odd $L$. By redefining the fermion fields in $H_\mathsf{G}$ as

$$\chi_\ell \to (-1)^\ell \chi_\ell = \hat{\chi}_\ell, \quad \ell = 1, 2, \cdots, L, \tag{3.58}$$

we see that the twisted Hamiltonian is equivalent to the untwisted Hamiltonian with the overall minus sign flipped, i.e.,

$$H_\mathsf{G}(\chi_\ell) = -H(\hat{\chi}_\ell) = -\frac{i}{2} \sum_{\ell=1}^{L} \hat{\chi}_{\ell+1} \hat{\chi}_\ell. \tag{3.59}$$

More precisely, we can think of the defect as associated with $\mathcal{C} = \chi_1 \chi_2 \cdots \chi_L$. As we said, this is not an operator but a c-number. Yet, it can lead to topological defects. To see that, first note that the defect in $H_\mathsf{G}$ in (3.57) is topological. The defect can be moved, e.g., to the link $(1, 2)$, by conjugating $H_\mathsf{G}$ by $\chi_1$. And moving it all the way around is obtained by conjugating the Hamiltonian by the c-number $\mathcal{C} = \chi_1 \chi_2 \cdots \chi_L$.

Recall that for even $L$, the system has a parity symmetry and we could introduce a G defect. Now we see that for odd $L$, there are no parity or G symmetries, and the system with the G defect is actually conjugate to the system without the defect, i.e., $H_\mathsf{G} = \mathsf{P} H \mathsf{P}^{-1}$.

Let us turn to the time-reversal transformation $\mathcal{T}$. It acts on the fermion operators as

$$\mathcal{T}: \quad \chi_\ell \to \mathcal{T} \chi_\ell \mathcal{T}^{-1} = \sum_{\ell'} R(\mathcal{T})_{\ell,\ell'} \chi_{\ell'} = (-1)^{\ell+1} \chi_\ell \quad, \quad \ell = 1, 2, \cdots, L \tag{3.60}$$

and it is extended periodically. Note that we use a different range of $\ell$ compared to the parity transformation (3.55).

Similar to the case of parity, this anti-unitary transformation is not a symmetry of the Hamiltonian. Instead,[29]

$$\mathcal{T} H \mathcal{T}^{-1} = \frac{i}{2} \sum_{\ell=1}^{L-1} \chi_{\ell+1} \chi_\ell - \frac{i}{2} \chi_1 \chi_L = H_\mathsf{G}. \tag{3.62}$$

We find that the conjugation of the untwisted Hamiltonian by the time-reversal transformation gives the twisted one.

We define the conjugated translation operator as

$$\hat{T} = \mathcal{T} T \mathcal{T}^{-1}. \tag{3.63}$$

It does not commute with $H$, but it is a symmetry of $H_\mathsf{G}$ of (3.21) for odd $L$. It is given by

$$\hat{T} = \frac{1}{2^{\frac{L-1}{2}}} (1 + \chi_1 \chi_2)(1 + \chi_2 \chi_3) \cdots (1 + \chi_{L-1} \chi_L). \tag{3.64}$$

---

[29]It is important to first write the Hamiltonian as

$$H = \frac{i}{2} \sum_{\ell=1}^{L-1} \chi_{\ell+1} \chi_\ell + \frac{i}{2} \chi_1 \chi_L \tag{3.61}$$

so that all the fermions are in the range $\ell = 1, 2, \cdots, L$, and then apply (3.60).

We also have $\hat{T} = \mathsf{P}T^{-1}\mathsf{P}^{-1}$. It acts on the fermion fields as

$$\hat{T}\chi_\ell\hat{T}^{-1} = \begin{cases} -\chi_{\ell+1}, & \ell = 1, 2, \cdots, L-1, \\ \chi_1, & \ell = L, \end{cases} \tag{3.65}$$

$$\hat{T}\hat{\chi}_\ell\hat{T}^{-1} = \hat{\chi}_{\ell+1}.$$

We see that the conjugated translation operator $\hat{T}$ acts as the ordinary translation on the new fermion fields $\hat{\chi}_\ell$ in (3.59).

In Section 3.1.3, we argued that the odd $L = 2N + 1$ theory can be viewed as the even $L = 2N$ theory with a translation defect. $H$ is obtained using a $T$ defect and $H_\mathsf{G}$ is obtained using a $\hat{T}$ defect. These observations are consistent with the results of this Section where $H$ and $H_\mathsf{G}$ are related by parity, and so do $T$ and $\hat{T}$.

Finally, we note that the Hamiltonian (3.2) is invariant under an anti-unitary transformation, which we denote by $\mathcal{RT}$. (The unitary operator $\mathcal{R}$ will be given below soon.) It acts on the fermion fields as

$$\mathcal{RT}: \quad \chi_\ell \to -\chi_{L-\ell}, \quad \ell = 1, 2, \cdots, L, \tag{3.66}$$

with $j \sim j + L$. Explicitly, it is a composition of the above transformations, $\mathcal{RT} = T^{-1}\mathsf{P}T^{-1}\mathcal{T}$. This is the lattice version of the standard CPT symmetry. $\mathcal{RT}$ acts on the translation operator (3.32) as

$$\mathcal{RT}\,T\,(\mathcal{RT})^{-1} = \frac{1}{2^{\frac{L-1}{2}}}(1 - \chi_{L-1}\chi_{L-2})(1 - \chi_{L-2}\chi_{L-3})\cdots(1 - \chi_2\chi_1)(1 - \chi_1\chi_L) = T^{-1}. \tag{3.67}$$

## 3.3 Anomalies and local counterterms

We have discussed the algebras of various symmetry operators on a Majorana chain. While the symmetry operators are motivated by a particular Hamiltonian, such as (3.2) and (3.21), the algebras they obey are independent of the choice of the Hamiltonian. More specifically, for even $L$, the algebras of the translation $T$, fermion parity $\mathsf{G}$, parity $\mathsf{P}$, and time-reversal symmetry $\mathcal{T}$ obey the algebra in (3.48). When there is a $\mathsf{G}$ defect, the algebra becomes (3.53). For odd $L$, the only symmetry is the translation operator $T$, which obeys $T^L = (-1)^{\frac{N(N+1)}{2}}$.

These algebras contain some projective phases that depend on the number of lattice sites. Can we redefine our symmetry operators to simplify, or even remove, these projective signs? Can they be interpreted as the 't Hooft anomalies of these symmetries?

In a generic quantum mechanical problem where this no notion of space, one can rescale an operator by an arbitrary phase. It is clear that some of the projective signs cannot be removed by any phases redefinition. These include the minus signs in $\mathsf{G}T = -T\mathsf{G}$, $\mathsf{G}\mathsf{P} = -\mathsf{P}\mathsf{G}$ in (3.48) for the case of even $L$ without a defect, which are examples of anomalies on the lattice as we will discuss soon.

However, by allowing such a general phase redefinition, we are only able to probe anomalies of the quantum mechanical problem. To see the precursor of some more subtle anomalies of the 1+1d system, we need to impose more restrictions on the allowed phase redefinitions.

For operators $S$ that can be written as a product of local factors, we expect the phase redefinitions to depend on $L$ in a controlled manner. (See the discussion around (1.14).) Such operators include the translation operators $T$ in (3.16) and (3.32), and its twisted version $T_G$ in (3.26) when there is a defect. For these operators, it is natural to postulate that the allowed phase redefinition should be restricted to be of the form

$$S \to e^{i\alpha L + i\beta} S \,. \tag{3.68}$$

This corresponds to redefining the corresponding line operator/defect by a local counterterm. For instance, $\alpha$ corresponds to multiplying the local unitary operator by a phase. In case of a continuous symmetry, it arises from adding a real constant to the time component of the current. In fact, for even $L$ with or without a defect, the restriction due to (3.68) will not affect the conclusion. It will be important only for odd $L$ below. For other operators, e.g., the fermion parity $G$ in (3.8), the parity operators $P$ of (3.44), and $P_G$ of (3.52), there is no such restriction and the phase can have more complicated $L$ dependence.

### *Even $L$*

Following our rule of the phase redefinition, for even $L$ without a defect, we define a new translation operator $T_{RR}$ and a new fermion parity operator $(-1)^F$ on the lattice as[30]

$$T_{RR} = e^{\frac{2\pi i(N-1)}{8}} T = \frac{e^{\frac{2\pi i(N-1)}{8}}}{2^{\frac{2N-1}{2}}} \chi_1 (1 + \chi_1 \chi_2)(1 + \chi_2 \chi_3) \cdots (1 + \chi_{2N-1} \chi_{2N}) \,, \tag{3.69}$$
$$(-1)^F = i^N G = i^N \chi_1 \chi_2 \cdots \chi_{2N} \,.$$

The algebra (3.20) becomes

$$\left((-1)^F\right)^2 = 1 \,, \quad T_{RR}^L = 1 \,, \quad (-1)^F T_{RR} = -T_{RR} (-1)^F \,. \tag{3.70}$$

For the parity operator $P$, since it is not a product of operators with local support (see (3.44)), we are allowed to rescale it by an arbitrary phase. We define a new parity operator $\mathcal{P}$ on the lattice as

$$\mathcal{P} = e^{\frac{2\pi i N(N-1)}{8}} P \,. \tag{3.71}$$

---

[30]Clearly, we could have redefined $(-1)^F \to -(-1)^F$ without affecting any of the conclusions below. The arbitrary choice here is such that this expression in terms of the fermions, satisfies (4.26) in the $G$ twisted theory for the specific Hamiltonian (3.21). We denote this new, rescaled fermion parity operator as $(-1)^F$ because, for the specific choice of the Hamiltonian (3.2), it corresponds to the continuum fermion parity operator in the Majorana CFT. Similarly, we denote the rescaled translation operator as $T_{RR}$ because the fermions are periodic in the untwisted problem, which corresponds to the RR boundary condition in the continuum. See Section 4.1 for more discussions.

We leave the time-reversal symmetry operator $\mathcal{T}$ as is. Written in terms of the rescaled operators, the algebra (3.48) for even $L = 2N$ without a defect now becomes

$$
\begin{aligned}
&\left((-1)^F\right)^2 = 1\,, \quad T_{\mathrm{RR}}^L = 1\,, \quad \mathcal{P}^2 = 1\,, \quad \mathcal{T}^2 = 1\,, \\
&(-1)^F\, T_{\mathrm{RR}} = -T_{\mathrm{RR}}\,(-1)^F\,, \\
&(-1)^F\, \mathcal{P} = -\mathcal{P}\,(-1)^F\,, \\
&\mathcal{P}\, T_{\mathrm{RR}} = -i(-1)^F T_{\mathrm{RR}}^{-1}\, \mathcal{P} \\
&\mathcal{T}\,(-1)^F = (-1)^F\,\mathcal{T}\,, \quad (\mathcal{T}\mathcal{P})^2 = -1\,, \quad \mathcal{T}\, T_{\mathrm{RR}} = -iT_{\mathrm{RR}}\,(-1)^F\,\mathcal{T}
\end{aligned}
\tag{3.72}
$$

The advantage of the rescaled operators is that now all the projective phases are independent of $N$.[31]

The fact that we cannot completely remove all the projective phases in (3.72) signals an anomaly on the lattice. In particular we have

$$
\begin{aligned}
&(-1)^F T_{\mathrm{RR}} = -T_{\mathrm{RR}}(-1)^F\,, \\
&(-1)^F \mathcal{P} = -\mathcal{P}(-1)^F\,, \\
&\mathcal{P}\, T_{\mathrm{RR}} = -i(-1)^F T_{\mathrm{RR}}^{-1}\, \mathcal{P}\,.
\end{aligned}
\tag{3.73}
$$

The minus sign in the first line was observed in [57, 15]. Such an algebra is incompatible with a gapped phase with a non-degenerate ground state. It is interpreted as an anomaly. Here we further find other anomalies involving the spatial parity operator $\mathcal{P}$. In Section 4.1, we will focus on a specific free Hamiltonian and compare the algebra and anomalies in (3.72) with those in the continuum CFT discussed in Section 2.2.

### *Even L with a defect*

Next, we move on to the algebra (3.53) of even $L$ with a fermion parity defect. We redefine the fermion parity $\mathsf{G}$ as in (3.69), and rescale the twisted translation by the following local phase[32]

$$
T_{\mathrm{NSNS}} = e^{-\frac{2\pi i N}{8}} T_{\mathsf{G}} = \frac{e^{-\frac{2\pi i N}{8}}}{2^{\frac{2N-1}{2}}} (1 - \chi_1\chi_2)(1 - \chi_2\chi_3)\cdots(1 - \chi_{2N-1}\chi_{2N})\,.
\tag{3.74}
$$

The algebra (3.28) becomes

$$
\left((-1)^F\right)^2 = 1\,, \quad T_{\mathrm{NSNS}}^L = (-1)^F\,, \quad (-1)^F\, T_{\mathrm{NSNS}} = T_{\mathrm{NSNS}}\,(-1)^F\,.
\tag{3.75}
$$

For the twisted parity $\mathsf{P}_{\mathsf{G}}$, we define a new parity operator $\mathcal{P}_{\mathsf{G}}$ as[33]

$$
\mathcal{P}_{\mathsf{G}} = (-1)^{\frac{N(N-1)(N-2)}{2}} e^{\frac{2\pi i N(N+1)}{8}} \mathsf{P}_{\mathsf{G}}\,.
\tag{3.76}
$$

---

[31]Similar to the comment in the continuum in footnote 14, we can also redefine $(-1)^F \to i(-1)^F$ so that all the phases are $\pm 1$ and independent of $N$. Again we choose not to do it so that $(-1)^F$ is order 2.

[32]We denote this rescaled operator as $T_{\mathrm{NSNS}}$ because the fermions in the twisted problem with a defect correspond to the NSNS boundary condition in the continuum. See Section 4.2 for an example.

[33]The first factor on the right-hand side is chosen so that $\mathcal{P}_{\mathsf{G}}$ acts with $+1$ eigenvalue on the ground state $|\Omega\rangle_{\mathrm{NSNS}}$ of the particular Hamiltonian we will study in Section 4.2.

so that $\mathcal{P}_\mathsf{G}^2 = 1$. Note that since $\mathcal{P}_\mathsf{G}$ is not a product of local operators, we do not impose the rule (3.68) on the phase redefinition.

For even $L = 2N$ with a defect, the algebra (3.53) involving the parity and time-reversal symmetries now becomes

$$
\begin{aligned}
&\left((-1)^F\right)^2 = 1\,, \quad T_{\mathrm{NSNS}}^L = (-1)^F\,, \quad \mathcal{P}_\mathsf{G}^2 = 1\,, \quad \mathcal{T}^2 = 1\,, \\
&(-1)^F\,T_{\mathrm{NSNS}} = T_{\mathrm{NSNS}}\,(-1)^F\,, \\
&(-1)^F\,\mathcal{P}_\mathsf{G} = \mathcal{P}_\mathsf{G}\,(-1)^F\,, \\
&\mathcal{P}_\mathsf{G}\,T_{\mathrm{NSNS}} = (-1)^F\,T_{\mathrm{NSNS}}^{-1}\,\mathcal{P}_\mathsf{G}\,, \\
&\mathcal{T}\,(-1)^F = (-1)^F\,\mathcal{T}\,, \quad (\mathcal{T}\mathcal{P}_\mathsf{G})^2 = 1\,, \quad \mathcal{T}\,T_{\mathrm{NSNS}} = T_{\mathrm{NSNS}}\,(-1)^F\,\mathcal{T}\,.
\end{aligned}
\tag{3.77}
$$

We see that we are able to remove all the projective phases for even $L$ with a defect, and there is no anomaly in the system with a defect.

Odd L

For odd $L = 2N + 1$, the translation symmetry $T$ obeys $T^L = (-1)^{\frac{N(N+1)}{2}}$ (see (3.33)). Since $T$ is a product of local operators (3.32), we ask if there is a local phase redefinition (3.68) such that its $L$-th power is 1 for all odd $L$? Intriguingly, the answer is negative.

Following (3.68), we define[34]

$$
T_{\mathrm{odd}} = e^{i\pi(xN+y)} \frac{e^{\frac{-2\pi i(2N+1)}{16}}}{2^N}(1 - \chi_1\chi_2)(1 - \chi_2\chi_3)\cdots(1 - \chi_{2N}\chi_{2N+1})\,,
\tag{3.78}
$$

with $x, y \in \mathbb{R}$. Here we have pulled out the factor $e^{\frac{-2\pi i(2N+1)}{16}}$ for later convenience. Using (3.33), we compute

$$
T_{\mathrm{odd}}^L = e^{i\pi(xN+y)(2N+1)}e^{-\frac{2\pi i}{16}}\,.
\tag{3.79}
$$

We choose the coefficients $x, y$ so that the right-hand side is independent of $N$. This is achieved with $x = -\frac{n_1}{2}, y = \frac{3n_1}{4} + n_2$ with some general integers $n_1, n_2$. As a result, the local phase redefinition (3.68) can only simplify the $L$-th power of the translation to the following form

$$
T_{\mathrm{odd}}^L = \exp\left(\frac{2\pi i n}{16}\right)\,, \quad n \in 2\mathbb{Z} + 1\,.
\tag{3.80}
$$

Since $n = 6n_1 + 8n_2 - 1$, we can change the odd integer $n$ using the phase redefinition. We see that the only invariant fact here is that $n$ is odd and in particular it cannot vanish.

We interpret the phase in (3.80) as a more subtle anomaly than the anomaly in (3.73). In contrast to the latter anomaly, the one in (3.80) can be removed by a phase redefinition if we

---

[34]In the continuum, depending on the choice of the Hamiltonian, the lattice model with odd $L$ can be mapped to either the NSR or the RNS theory. For this reason, we denote the rescaled translation operator as $T_{\mathrm{odd}}$, rather than $T_{\mathrm{NSR}}$ or $T_{\mathrm{RNS}}$. See Section 4.3 for more discussions on the comparison to the continuum.

give up on the locality in the one-dimensional space. In other words, (3.80) presents an anomaly intrinsic to a 1+1d quantum system, not just of a quantum mechanical system with no notion of locality. As is manifest in (3.33), this anomaly is order 2. Indeed, if we stack two copies of the Majorana chain on top of each other, then there is a choice of the local phase redefinition such that $T^L_{\text{odd}} = 1$.

Importantly, the anomaly in (3.80) is universal for any translationally invariant Hamiltonian for the Majorana chain of odd number of sites. See Section 4.3 for more discussions on the particular case of the free Hamiltonian $H$ (3.2) and relations to the continuum anomalies.

## 3.4    Relations between the partition functions

In this section, we will follow [32] and show how the projective algebras discussed above can be probed using the Euclidean partition function of the system. The projective phases and the related anomalies will manifest themselves in minus signs relating partition functions that should naively be the same. Therefore, the corresponding partition functions should vanish.

We focus on the case of even $L = 2N$. Define the partition functions of the untwisted problem as[35]

$$Z(\beta, L, m, n, m_{\mathcal{P}}) = \text{Tr}[\, e^{-\beta H} \left((-1)^F\right)^m T^n_{\text{RR}} \, \mathcal{P}^{m_{\mathcal{P}}}]. \tag{3.81}$$

From (3.72), we see that $m, m_{\mathcal{P}}$ are defined modulo 2, while $n$ is defined modulo $2N$. Here (and similarly below for the twisted problem), we use the rescaled fermion parity $(-1)^F$, translation $T_{\text{RR}}$, and parity operators $\mathcal{P}$ introduced in Section 3.3. Inserting $\left((-1)^F\right)^2 = 1, T_{\text{RR}} T^{-1}_{\text{RR}} = 1, \mathcal{P}^2 = 1$ into the trace, we have

$$Z(\beta, L, m, n, m_{\mathcal{P}}) = (-1)^{n+m_{\mathcal{P}}} Z(\beta, L, m, n, m_{\mathcal{P}}),$$

$$Z(\beta, L, m, n, m_{\mathcal{P}}) = \begin{cases} (-1)^m Z(\beta, L, m, n, m_{\mathcal{P}} = 0), & \text{if } m_{\mathcal{P}} = 0, \\ i(-1)^{m+n} Z(\beta, L, m + 1, n - 2, m_{\mathcal{P}} = 1), & \text{if } m_{\mathcal{P}} = 1, \end{cases} \tag{3.82}$$

$$Z(\beta, L, m, n, m_{\mathcal{P}}) = e^{i\pi \frac{n(n-2)}{2} + i\pi m} Z(\beta, L, m + n, -n, m_{\mathcal{P}}).$$

Let us discuss the consequences of these relations. When $m_{\mathcal{P}} = 0$, the first two equalities of (3.82) imply that the only nonzero partition functions are those with $m = 0$ and even $n$:

$$Z(\beta, L, m, n, m_{\mathcal{P}} = 0) = 0, \quad \text{if } m \text{ or } n \text{ is odd}. \tag{3.83}$$

With $m, n$ both even, the third line of (3.82) then implies

$$Z(\beta, L, m = 0, n, m_{\mathcal{P}} = 0) = Z(\beta, L, m = 0, -n, m_{\mathcal{P}} = 0). \tag{3.84}$$

When $m_{\mathcal{P}} = 1$, the first line of (3.82) implies that $n$ has to be odd for the partition function to be nonzero.

$$Z(\beta, L, m, n, m_{\mathcal{P}} = 1) = 0, \quad \text{if } n \text{ is even}. \tag{3.85}$$

---

[35]Here and in Appendix A, we slightly abuse the notation and use $m = 0, 1$ as the exponent of the fermion parity operator. This is not to be confused with the Majorana mass $m$ in (3.41). Similarly, here $n$ is not to be confused with the odd integer in (3.80).

With $n$ odd, then the third equality of (3.82) is implied by the second, which reads

$$Z(\beta, L, m, n, m_{\mathcal{P}} = 1) = -i(-1)^m Z(\beta, L, m+1, n-2, m_{\mathcal{P}} = 1).$$ (3.86)

By applying this relation repeatedly, we can bring $n$ to $n = \pm 1$.

Next, we define the G-twisted partition functions

$$Z_{\mathsf{G}}(\beta, L, m, n, m_{\mathcal{P}}) = \text{Tr}\big[ e^{-\beta H_{\mathsf{G}}} \left((-1)^F\right)^m T_{\text{NSNS}}^n \mathcal{P}_{\mathsf{G}}^{m_{\mathcal{P}}} \big].$$ (3.87)

From (3.77), we see that $m, m_{\mathcal{P}}$ are defined modulo 2 while $n$ is defined modulo $2N$. Inserting $\left((-1)^F\right)^2 = 1$ does not yield any nontrivial relation. The relations from inserting $T_{\text{NSNS}} T_{\text{NSNS}}^{-1} = 1, \mathcal{P}_{\mathsf{G}}^2 = 1$ are generated by

$$
\begin{aligned}
Z_{\mathsf{G}}(\beta, L, m, n, m_{\mathcal{P}} = 0) &= Z_{\mathsf{G}}(\beta, L, m+n, -n, m_{\mathcal{P}} = 0), \\
Z_{\mathsf{G}}(\beta, L, m, n, m_{\mathcal{P}} = 1) &= Z_{\mathsf{G}}(\beta, L, m+1, n-2, m_{\mathcal{P}} = 1).
\end{aligned}
$$ (3.88)

Using the second relation, we can always set $n$ to be 1 when $m_{\mathcal{P}} = 1$.

The above relations hold true for any lattice Hamiltonian enjoying these symmetries. In Appendix A, we compare these lattice relations with those in the continuum for the specific case of a free Majorana fermion.

As we see, many of these partition functions vanish. This means that the Hilbert space, including the ground state, must have certain degeneracies, leading to cancellations in the partition function. One way to think about these degeneracies is that they are associated with projective representations of the symmetry algebra. More generally, the vanishing partition functions show that the spectrum of the system cannot be trivially gapped.

# 4   Emanant chiral fermion parity from the free fermion Hamiltonian

In this Section, we focus on a specific Hamiltonian (1.7), the free fermion Hamiltonian with the nearest neighbor interaction. We also study the Hamiltonian (1.8) twisted by the fermion parity operator. We compare the lattice analysis with the continuum Majorana CFT. In particular, we find that the chiral fermion parity $(-1)^{F_L}$ in the continuum emanates from the lattice translation symmetry.

## 4.1   Even $L$ and the RR theory

Let the number of sites be even $L = 2N$ without a defect, i.e.,

$$H = \frac{i}{2} \sum_{\ell=1}^{2N} \chi_{\ell+1} \chi_\ell,$$ (4.1)

$$\chi_\ell = \chi_{\ell+2N}.$$

We will show that it flows to the continuum CFT of a free, massless Majorana fermion. More specifically, it flows to the RR theory, where the fermion field is periodic around the spatial circle as in (2.6) with $\nu_L = \nu_R = 0$.

We use the momentum modes

$$\chi_\ell = \frac{1}{\sqrt{N}} \sum_k \exp\left(\pi i \frac{\ell k}{N}\right) d_k \qquad , \qquad d_{k+2N} = d_k . \tag{4.2}$$

The hermicity of $\chi_\ell$ implies

$$\begin{aligned} d_k &= d_{-k}^\dagger \\ d_0 &= d_0^\dagger , \quad d_N = d_N^\dagger . \end{aligned} \tag{4.3}$$

The momentum modes obey the anticommutation relation:

$$\{d_k, d_{k'}\} = \delta_{k,-k'} . \tag{4.4}$$

The Hamiltonian (4.1) in momentum space is

$$H = i \sum_k e^{i\pi \frac{k}{N}} d_k d_{-k} = 2 \sum_{k=1}^{N-1} \sin\left(\frac{\pi k}{N}\right) d_k^\dagger d_k + \text{const.} \tag{4.5}$$

The two hermitian zero modes $d_0, d_N$ generate a two-dimensional space of ground states. See Figure 1 for the spectrum.

In the large $L$ limit, let us focus on the low-lying modes created by the $d_k^\dagger$'s. They come in two groups, one near $k = 0$ and the other near $k = N$. The Hamiltonian for these low-lying modes is

$$H \sim \frac{2\pi}{N} \sum_{k=1}^{L_0} k \, d_k^\dagger d_k + \frac{2\pi}{N} \sum_{k=1}^{L_0} k \, d_{N-k}^\dagger d_{N-k} + \text{const.} \tag{4.6}$$
$$1 \ll L_0 \ll L .$$

We find that the low-lying spectrum matches with the continuum Hamiltonian (2.9) of the RR theory (with $\nu_L = \nu_R = 0$), up to an overall constant. More explicitly, we identify the lattice and continuum modes as

$$\begin{aligned} \chi_{R,k} &= d_k , & |k| &\leq L_0 \ll L , \\ \chi_{L,k} &= d_{N-k} , & |k| &\leq L_0 \ll L . \end{aligned} \tag{4.7}$$

To conclude, the low-lying lattice momentum modes created by the $d_k^\dagger$'s near $k = 0$ give rise to the continuum right-moving fermion modes $\chi_R$, while those near $k = N$ give rise to the left-moving fermion modes $\chi_L$.

We now turn to the continuum limit of the lattice symmetries. We use the generators with the preferred phase choice $T_{RR}$ and $(-1)^F$ of (3.69), $\mathcal{P}$ of (3.71), and $\mathcal{T}$. (Recall that we did not

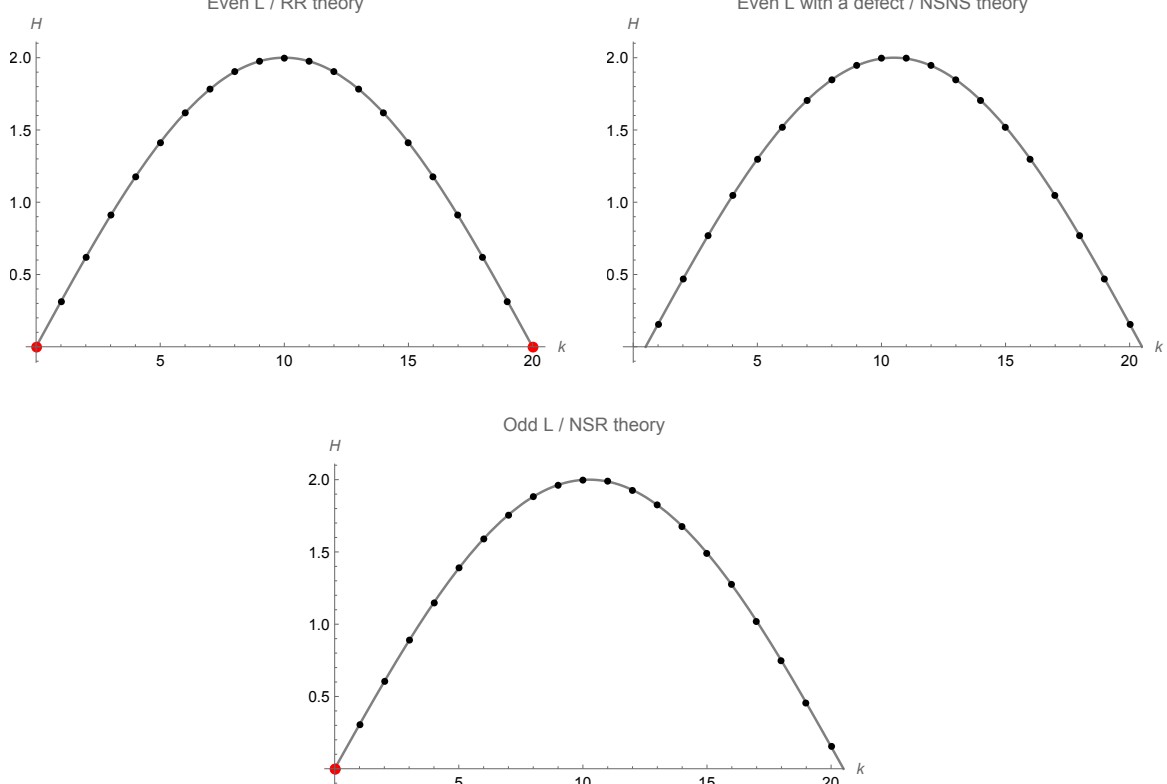

Figure 1: The spectrum of the free fermion Hamiltonian (4.1) for even $L = 2N$ and odd $L = 2N + 1$, and that for the twisted Hamiltonian (4.18) for even $L = 2N$. (The figures are for $N = 20$.) The Hamiltonians in momentum space are given in (4.5), (4.36), and (4.22). In each case, the low-lying modes near $k = 0$ and $k = N$ correspond to the right- and left-moving modes in the continuum Majorana CFT. More specifically, the three cases correspond to the RR, NSR, and NSNS theories in the continuum, respectively. The black dots represent the nonzero modes, while the red dots represent the zero modes. The gray line plots the energy function by treating the momentum $k$ as a continuous variable.

rescale the time-reversal operator $\mathcal{T}$.) They act on momentum modes as

$$
\begin{aligned}
T_{\text{RR}}\, d_k\, T_{\text{RR}}^{-1} &= \exp\left(\frac{\pi i k}{N}\right) d_k\,, \\
(-1)^F\, d_k\, (-1)^F &= -d_k \\
\mathcal{P} d_k \mathcal{P}^{-1} &= d_{N-k}\,, \\
\mathcal{T} d_k \mathcal{T}^{-1} &= -d_{N-k}\,.
\end{aligned}
\tag{4.8}
$$

In particular, $T_{\text{RR}} d_0 T_{\text{RR}}^{-1} = d_0$, $T_{\text{RR}} d_N T_{\text{RR}}^{-1} = -d_N$.

Using the map to the low-energy fields (4.7), we find the action of the translation operator on

the left and right-moving continuum modes:

$$T_{\text{RR}} \chi_{\text{R},k} T_{\text{RR}}^{-1} = \exp\left(\frac{2\pi i k}{L}\right) \chi_{\text{R},k}\,,$$
$$T_{\text{RR}} \chi_{\text{L},k} T_{\text{RR}}^{-1} = -\exp\left(-\frac{2\pi i k}{L}\right) \chi_{\text{L},k}\,. \tag{4.9}$$

Interestingly, $T_{\text{RR}}$ acts on $\chi_{\text{L},k}$ with an additional minus sign relative to the expected phase of continuum translation. This indicates that the lattice translation $T_{\text{RR}}$ does not flow to the continuum translation $e^{2\pi i P/L}$; rather, an internal global symmetry emanates from the lattice translation. It is the chiral fermion parity $(-1)^{F_{\text{L}}}$.

In order to find the precise relation between the lattice generator $T_{\text{RR}}$ and the continuum symmetries $P$ and $(-1)^{F_{\text{L}}}$, we need to know not only how they act on the various operators, but also how they act on the Hilbert space. On the lattice, we can choose an orthonormal basis $\{|0\rangle, |1\rangle\}$ for the ground states, which by definition obey

$$d_k |0\rangle = d_k |1\rangle = 0\,, \quad k = 1, \cdots, N - 1\,. \tag{4.10}$$

Using these and the expression for $T_{\text{RR}}$ in terms of the fundamental fields and our phase choice, we can choose the basis $\{|0\rangle, |1\rangle\}$ such that

$$T_{\text{RR}} |0\rangle = + |0\rangle\,,$$
$$T_{\text{RR}} |1\rangle = - |1\rangle \tag{4.11}$$

and $(-1)^F$ exchanges them.

In the continuum RR theory, we also have two ground states $\{|0\rangle, |1\rangle\}$ generated by the zero modes $\chi_{\text{L},0}, \chi_{\text{R},0}$. They are exchanged by $(-1)^F$ (4.11) and have vanishing continuum momentum $P$. We can choose $(-1)^{F_{\text{L}}}$ in the continuum to act on the ground states as

$$(-1)^{F_{\text{L}}} |0\rangle = + |0\rangle\,,$$
$$(-1)^{F_{\text{L}}} |1\rangle = - |1\rangle\,, \tag{4.12}$$

as in (2.18). Hence, we have a match between the lattice and the continuum ground states.

We thus obtain the following relation between the lattice translation $T_{\text{RR}}$ and the continuum operators $(-1)^{F_{\text{L}}}, P$ for the low-lying modes:

$$T_{\text{RR}} = (-1)^{F_{\text{L}}} e^{\frac{2\pi i P}{L}}\,. \tag{4.13}$$

We also have the lattice relations

$$T_{\text{RR}}^L = 1\,,$$
$$\left[(-1)^F\right]^2 = 1\,, \tag{4.14}$$

which are compatible with the continuum relations in the RR Hilbert space in (2.19), (2.25)

$$e^{2\pi i P} = 1\,,$$
$$\left[(-1)^{F_{\text{L}}}\right]^2 = 1\,,$$
$$\left[(-1)^F\right]^2 = 1\,. \tag{4.15}$$

As with other emanant symmetries [32], on the lattice, only $T_{\mathrm{RR}}$ is meaningful. The separation in the right-hand side between $(-1)^{F_{\mathrm{L}}}$ and $e^{\frac{2\pi i P}{L}}$ is meaningful only at low energies, where we take the eigenvalues of $P$ to be of order one and much smaller than $L$. Also, as in [32], the relation (4.13) between the lattice and the continuum operators is exact and does not have $\mathcal{O}(1/L)$ corrections.

Under the dictionary (4.13), we see that the symmetry algebra on the lattice in (3.72) agrees with that in the continuum RR theory in (2.19)

$$
\begin{aligned}
(-1)^F(-1)^{F_{\mathrm{L}}} &= -(-1)^{F_{\mathrm{L}}}(-1)^F\,, \\
(-1)^F \mathcal{P} &= -\mathcal{P}(-1)^F\,, \\
\mathcal{P}\,(-1)^{F_{\mathrm{L}}} &= -i(-1)^F\,(-1)^{F_{\mathrm{L}}}\,\mathcal{P}\,.
\end{aligned}
\tag{4.16}
$$

The projective phases in (3.72), and in particular those in (3.73):

$$
\begin{aligned}
(-1)^F T_{\mathrm{RR}} &= -T_{\mathrm{RR}}(-1)^F\,, \\
(-1)^F \mathcal{P} &= -\mathcal{P}(-1)^F\,, \\
\mathcal{P}\,T_{\mathrm{RR}} &= -i(-1)^F T_{\mathrm{RR}}^{-1}\,\mathcal{P}\,,
\end{aligned}
\tag{4.17}
$$

were interpreted as anomalies on the lattice in Section 3.3 and they match with the continuum phases in (4.16).

These phases signal the following anomalies in the continuum:

- the mod 8 anomaly of $(-1)^F, (-1)^{F_{\mathrm{L}}}, \mathcal{P}$ classified by $\mathrm{Hom}(\mathrm{Tors}\,\Omega_3^{\mathrm{DPin}}(pt), U(1)) = \mathbb{Z}_8$,

- the mod 8 anomaly of $(-1)^F, (-1)^{F_{\mathrm{L}}}$ classified by $\mathrm{Hom}(\mathrm{Tors}\,\Omega_3^{\mathrm{Spin}}(B\mathbb{Z}_2), U(1)) = \mathbb{Z}_8$,

- the mod 2 anomaly of $(-1)^F, \mathcal{P}$ classified by $\mathrm{Hom}(\mathrm{Tors}\,\Omega_3^{\mathrm{Pin}^+}(pt), U(1)) = \mathbb{Z}_2$.

Finally, we comment that the chiral fermion parity $(-1)^{F_{\mathrm{R}}}$ for the right-movers emanates from the lattice symmetry $\hat{T} = (-1)^N T\mathsf{G}$ defined in (3.19), up a phase. This is consistent with the continuum relation $(-1)^{F_{\mathrm{R}}} = i(-1)^{F_{\mathrm{L}}}(-1)^F$ in Section 2.2. Indeed, $\hat{T} = \mathcal{T}T\mathcal{T}^{-1}$ is related to $T$ by conjugation of the time-reversal symmetry.

## 4.2  Even $L$ with a defect and the NSNS theory

Next, we insert a fermion parity defect at the $(2N, 1)$-link. There are two equivalent ways to proceed. In the first approach, we extend the fermion fields periodically by $\chi_{\ell+2N} = \chi_\ell$ but represent the defect as a modification of the Hamiltonian at one link as in

$$
H_{\mathsf{G}} = \frac{i}{2}\sum_{\ell=1}^{2N-1}\chi_{\ell+1}\chi_\ell - \frac{i}{2}\chi_1\chi_{2N}\,, \quad \chi_{\ell+2N} = \chi_\ell\,.
\tag{4.18}
$$

Alternatively, we can extend the one-dimensional periodic chain to an infinite chain, and impose the twisted boundary condition on the fermion fields as $\chi_{\ell+2N} = -\chi_\ell$, but leave the Hamiltonian as the untwisted one (4.1). The fermion fields are not functions, but sections, over the periodic

chains. In the second approach, it is clear that the twisted problem corresponds to the NSNS theory in the continuum where the fermions obey the anti-periodic boundary condition. Below we will proceed using the first approach and restrict the range of $\ell$ to $1, 2, \cdots, 2N$.

For the twisted problem, we expand the Majorana field using half-integer momentum modes:

$$\chi_\ell = \frac{1}{\sqrt{N}} \sum_k \exp\left(i\pi \frac{\ell(k - \frac{1}{2})}{N}\right) b_{k-\frac{1}{2}}, \quad \ell = 1, 2, \cdots, 2N, \tag{4.19}$$

where the sum over $k$ is over integers modulo $2N$. It is important that the range of $\ell$ is restricted to $1, \cdots, 2N$ in the expression above. We define $b_{k-\frac{1}{2}+2N} = b_{k-\frac{1}{2}}$. Since $\chi_\ell$ is hermitian, we have

$$b_{k-\frac{1}{2}} = b^\dagger_{-k+\frac{1}{2}}. \tag{4.20}$$

They obey the anticommutation relation

$$\{b_{k-\frac{1}{2}}, b_{-k'+\frac{1}{2}}\} = \delta_{k,k'}. \tag{4.21}$$

The Hamiltonian in momentum space is

$$H_{\mathsf{G}} = 2 \sum_{k=1}^N \sin\left(\frac{\pi(k - \frac{1}{2})}{N}\right) b^\dagger_{k-\frac{1}{2}} b_{k-\frac{1}{2}} + \text{const}. \tag{4.22}$$

See Figure 1 for the spectrum.

Since there is no zero mode, there is a unique ground state $|\Omega\rangle_{\mathrm{NSNS}}$ satisfying

$$b_{k-\frac{1}{2}}|\Omega\rangle_{\mathrm{NSNS}} = 0, \quad k = 1, \cdots, N. \tag{4.23}$$

Next, we analyze the large-$L$, low-energy limit, as we did in the RR case above. We focus on the low-lying modes $b_{k-\frac{1}{2}}|\Omega\rangle_{\mathrm{NSNS}}$ near $k = 0$ and $k = N$. The Hamiltonian for these modes is

$$H_{\mathsf{G}} \sim \frac{2\pi}{N} \sum_{k=1}^{L_0} \left(k - \frac{1}{2}\right) b^\dagger_{k-\frac{1}{2}} b_{k-\frac{1}{2}} + \frac{2\pi}{N} \sum_{k=1}^{L_0} \left(k - \frac{1}{2}\right) b^\dagger_{N-k+\frac{1}{2}} b_{N-k+\frac{1}{2}} + \text{const}, \tag{4.24}$$
$$1 \ll L_0 \ll L.$$

We find that the low-lying spectrum matches with the Hamiltonian (2.9) of the NSNS theory (with $\nu_{\mathrm{L}} = \nu_{\mathrm{R}} = \frac{1}{2}$), up to an overall constant. More explicitly, the lattice and continuum modes are matched as follows

$$\chi_{\mathrm{R},k-\frac{1}{2}} = b_{k-\frac{1}{2}}, \quad |k| \leq L_0 \ll L$$
$$\chi_{\mathrm{L},k-\frac{1}{2}} = b_{N-k+\frac{1}{2}}, \quad |k| \leq L_0 \ll L. \tag{4.25}$$

For the symmetry operators, we consider the rescaled operators $T_{\mathrm{NSNS}}$ of (3.74), $(-1)^F$ of (3.69), and $\mathcal{P}_{\mathsf{G}}$ of (3.76).

It is straightforward to check that $(-1)^F = i^N \mathsf{G}$ acts trivially on the ground state:

$$(-1)^F |\Omega\rangle_{\text{NSNS}} = i^N \chi_1 \chi_2 \cdots \chi_{2N} |\Omega\rangle_{\text{NSNS}} = |\Omega\rangle_{\text{NSNS}} . \tag{4.26}$$

This is consistent with our definition of $(-1)^F$ and the identification of the state $|\Omega\rangle_{\text{NSNS}}$ on the lattice as the NSNS ground state in the continuum (2.11) (and hence the same symbol).

Similarly, the rescaled parity operator $\mathcal{P}_\mathsf{G}$ in (3.76) acts on the ground state as $\mathcal{P}_\mathsf{G}|\Omega\rangle_{\text{NSNS}} = |\Omega\rangle_{\text{NSNS}}$, and we identify it with the continuum parity operator $\mathcal{P}$ in the NSNS theory.

The translation symmetry operator that preserves $H_\mathsf{G}$ is $T_{\text{NSNS}}$ acts on the fermion fields as in (3.22). It follows that $T_{\text{NSNS}}$ acts on the momentum modes as

$$T_{\text{NSNS}} b_{k-\frac{1}{2}} T_{\text{NSNS}}^{-1} = \exp\left(\frac{\pi i (k - \frac{1}{2})}{N}\right) b_{k-\frac{1}{2}} . \tag{4.27}$$

Using the map (4.25), its action on the low-energy modes is the same as in the continuum:

$$
\begin{aligned}
T_{\text{NSNS}} \chi_{\text{R},k-\frac{1}{2}} T_{\text{NSNS}}^{-1} &= \exp\left(\frac{2\pi i (k - \frac{1}{2})}{L}\right) \chi_{\text{R},k-\frac{1}{2}} , \\
T_{\text{NSNS}} \chi_{\text{L},k-\frac{1}{2}} T_{\text{NSNS}}^{-1} &= -\exp\left(-\frac{2\pi i (k - \frac{1}{2})}{L}\right) \chi_{\text{L},k-\frac{1}{2}} .
\end{aligned}
\tag{4.28}
$$

Again, we see that the twisted translation operator acts on the left-moving fermion field with an extra minus sign. This means that the continuum chiral fermion parity $(-1)^{F_\text{L}}$ emanates from the lattice translation $T_{\text{NSNS}}$.

Let us derive the relation between $T_{\text{NSNS}}$ and the continuum chiral fermion parity for the low-lying modes. It is straightforward to compute the eigenvalues of $T_{\text{NSNS}}$ on the low-lying states to find

$$T_{\text{NSNS}} = (-1)^{F_\text{L}} e^{\frac{2\pi i P}{L}} . \tag{4.29}$$

We also have the lattice relations

$$
\begin{aligned}
T_{\text{NSNS}}^L &= (-1)^F , \\
\left[(-1)^F\right]^2 &= 1 ,
\end{aligned}
\tag{4.30}
$$

which are consistent with the continuum relations (2.14)

$$
\begin{aligned}
e^{2\pi i P} &= (-1)^F , \\
\left[(-1)^{F_\text{L}}\right]^2 &= 1 .
\end{aligned}
\tag{4.31}
$$

Again, the relation between the continuum and the lattice quantities in (4.29) is exact.

Under the dictionary (4.29), we see that the symmetry algebra on the lattice (3.77) agrees with that in the continuum NSNS theory in (2.12). And in particular, there are no anomalous phases.

## 4.3 Odd $L$, the NSR theory, and the mod 8 anomaly

Let the number of sites be odd $L = 2N + 1$ and focus on the free Hamiltonian:

$$H = \frac{i}{2} \sum_{\ell=1}^{2N+1} \chi_{\ell+1} \chi_\ell \,, \tag{4.32}$$

$$\chi_\ell = \chi_{\ell+2N+1} \,.$$

We define the momentum modes as

$$\chi_\ell = \sqrt{\frac{2}{2N+1}} \sum_k \exp\left( 2\pi i \frac{\ell k}{2N+1} \right) d_k \tag{4.33}$$

where the sum in $k$ is over integers modulo $2N + 1$, i.e., $d_{k+2N+1} = d_k$. Since $\chi_\ell$ is hermitian, we have

$$d_k = d_{-k}^\dagger \,. \tag{4.34}$$

In contrast to the even $L$ case, now we have only one hermitian zero mode rather than two:

$$d_0 = d_0^\dagger \,. \tag{4.35}$$

The Hamiltonian in momentum space is

$$H = 2 \sum_{k=1}^{N} \sin\left( \frac{2\pi k}{2N+1} \right) d_k^\dagger d_k + \text{const} \tag{4.36}$$

and

$$\{ d_k, d_{k'} \} = \delta_{k,-k'} \,. \tag{4.37}$$

See Figure 1 for the spectrum.

There is a unique ground state $|\Omega\rangle_{\text{NSR}}$ satisfying

$$d_k |\Omega\rangle_{\text{NSR}} = 0 \,, \quad k = 1, \cdots, N \,. \tag{4.38}$$

It is a one-dimensional representation of the one-dimensional Clifford algebra $\{d_0, d_0\} = 1$ generated by the hermitian zero mode $d_0$, i.e., $d_0 = +1/\sqrt{2}$ or $d_0 = -1/\sqrt{2}$. For concreteness, we choose the former quantization. We denote this state as $|\Omega\rangle_{\text{NSR}}$ because it is to be identified with the ground state of the NSR continuum theory in (2.27).

Next we examine the low-energy theory for large $L$. The modes created by $d_k^\dagger$ near $k = 0$ correspond to the right-movers, while the modes created by $d_k^\dagger$ near $k = N$ correspond to the left-movers. The Hamiltonian for these low-lying modes is

$$H \sim \frac{4\pi}{2N+1} \sum_{k=1}^{L_0} k \, d_k^\dagger d_k + \frac{4\pi}{2N+1} \sum_{k'=1}^{L_0} \left( k' - \frac{1}{2} \right) d_{N-k'+1}^\dagger d_{N-k'+1} + \text{const.} \tag{4.39}$$

$$1 \ll L_0 \ll L \,.$$

We find that the low-lying spectrum matches with the continuum Hamiltonian (2.9) of the NSR theory (with $\nu_{\mathsf{L}} = \frac{1}{2}, \nu_{\mathsf{R}} = 0$), up to an overall constant. More explicitly, we identify the lattice and the continuum modes as follows

$$\begin{aligned} \chi_{\mathsf{R},k} &= d_k \,, \quad |k| \le L_0 \ll L \,, \\ \chi_{\mathsf{L},k-\frac{1}{2}} &= d_{N-k+1} \,, \quad |k| \le L_0 \ll L \,. \end{aligned} \tag{4.40}$$

We will consider the phase of the translation operator $T_{\text{odd}}$ as in (3.78) with $x = y = 0$. It acts on the momentum modes as

$$T_{\text{odd}} \, d_k \, T_{\text{odd}}^{-1} = \exp\left(\frac{2\pi i k}{2N+1}\right) d_k \,. \tag{4.41}$$

It follows that

$$\begin{aligned} T_{\text{odd}} \chi_{\mathsf{R},k} T_{\text{odd}}^{-1} &= \exp\left(\frac{2\pi i k}{L}\right) \chi_{\mathsf{R},k} \,, \\ T_{\text{odd}} \chi_{\mathsf{L},k-\frac{1}{2}} T_{\text{odd}}^{-1} &= -\exp\left(-\frac{2\pi i}{L}\left(k - \frac{1}{2}\right)\right) \chi_{\mathsf{L},k-\frac{1}{2}} \end{aligned} \tag{4.42}$$

As in the previous cases, this means that lattice translation leads to an emanant $(-1)^{F_{\mathsf{L}}}$ symmetry. It is straightforward to work out the action of the symmetry on the low-lying states to find the exact expression

$$T_{\text{odd}} = (-1)^{F_{\mathsf{L}}} e^{\frac{2\pi i P}{L}} \,, \tag{4.43}$$

which is consistent with the lattice relation

$$T_{\text{odd}}^L = e^{-\frac{2\pi i}{16}} \tag{4.44}$$

and the continuum relations (2.28)

$$\begin{aligned} e^{2\pi i P} &= (-1)^{F_{\mathsf{L}}} e^{-\frac{2\pi i}{16}} \,, \\ \left((-1)^{F_{\mathsf{L}}}\right)^2 &= 1 \,. \end{aligned} \tag{4.45}$$

Finally, we can conjugate the Hamiltonian $H$ (4.32) by either the parity P or the time-reversal operator $\mathcal{T}$ (which are not symmetries of $H$) to obtain a twisted Hamiltonian $H_{\mathsf{G}}$ as in (3.57) and (3.62), which describes the RNS theory. The twisted Hamiltonian $H_{\mathsf{G}}$ is equivalent to $-H$ by a field redefinition as in (3.58), and the chiral fermion parity $(-1)^{F_{\mathsf{R}}}$ of the RNS theory emanates from the operator $\hat{T} = \mathcal{T} T \mathcal{T}^{-1} = \mathsf{P} T^{-1} \mathsf{P}^{-1}$ defined in (3.63) on the lattice.

*Precursor of the mod 8 anomaly*

The relation (4.43) is a precursor of an anomaly in the continuum Majorana CFT. As discussed in Section 2.3, in the continuum, the mod 8 anomaly of the chiral fermion parity $(-1)^{F_{\mathsf{L}}}$ can be detected by the momentum eigenvalue $P$ in the continuum NSR Hilbert space, i.e., $P \in \frac{1}{16} + \frac{\mathbb{Z}}{2}$ (see

(2.29) with $\nu = 1$). On the lattice, the exact normalization of the translation operator is subject to the ambiguity from the local phase redefinition $T \to e^{i\alpha L + i\beta} T$ as in (3.68). Nonetheless, we can normalize it such that for large $L$, we have (4.43) with $(-1)^{F_L} = \pm 1$ and all the low-lying states have finite $P$ (as opposed to $P$ of order $L$). Once we have set this normalization, we no longer have the freedom to redefine $T_{\text{odd}}$ by a phase. Consequently, in the low-energy theory, the phase in

$$T_{\text{odd}}^L = e^{-\frac{2\pi i}{16}} \tag{4.46}$$

is also meaningful. It leads to the modulo 8 anomaly of the continuum theory.

We see that by looking at the low-lying states at large but finite $L$, we can detect a precursor of the continuum modulo 8 anomaly.

We should stress that, as we discussed around (3.80), the complete lattice model exhibits only a modulo 2 anomaly as the phase in (4.46) can be redefined to be an odd power of the phase there. However, by looking at the low-lying states, $T_{\text{odd}}$ has a natural phase, such that (4.43) is satisfied with $P$ of order one. And then, the full modulo 8 anomaly is visible.[36]

In order to demonstrate the distinction between the modulo 2 anomaly on the lattice and the more refined modulo 8 anomaly seen in the low-lying states, we can do the following. The modulo 2 anomaly is independent of the choice of Hamiltonian. In particular, it is present both with the free Hamiltonian $H$ (3.2), and also with $-H$ in (3.59). The low-energy spectra of these two Hamiltonians lead to the NSR and the RNS theories of a free Majorana CFT. Let us compare these two systems. The chiral fermion parities $(-1)^{F_L}$ and $(-1)^{F_R}$ emanate from the corresponding translation operators of the two problems (which we denote as $T$ in (3.32) and $\hat{T}$ in (3.63), respectively). As discussed in Section 2.3, $(-1)^{F_L}$ and $(-1)^{F_R}$ carry opposite continuum anomalies, i.e., $\nu = 1$ versus $\nu = -1$ mod 8. This is consistent with the claim that the exact anomaly of the lattice model is determined by $\nu$ mod 2. However, if we consider a particular Hamiltonian and focus on its low-energy states, we can see a lattice precursor of the more subtle modulo 8 continuum anomaly.[37]

# 5 Local Hilbert spaces and bosonization

In the introduction, we mentioned some subtleties of fermionic theories: locality when the fundamental degrees of freedom anticommute at separated points, the relation to a tensor product Hilbert space, and the dependence on a choice of spin structure. We now discuss these issues in more detail and will demonstrate them in the Majorana chain.

---

[36]In the bosonic problem analyzed in [32], a similar anomaly was seen. There, lattice translation also leads to an emanant $\mathbb{Z}_2$ symmetry. And by examining the action of lattice translations on the low-lying states, an anomaly involving lattice translation can be seen. It is a lattice precursor of the pure $\mathbb{Z}_2$ anomaly of the continuum theory. Just like here, that anomaly cannot be seen by looking at all the states in the lattice Hilbert space and it is visible only in the low-energy spectrum. However, unlike the case here, in that bosonic problem, there is no analog of the modulo 2 translation anomaly (3.80), which exists for every Hamiltonian.

[37]In this example of free fermions, the lattice anomaly appears to only capture $\nu$ mod 2, which is classified by the "Arf layer" $H^1(\mathbb{Z}_2, \mathbb{Z}_2) \simeq \mathbb{Z}_2$ [42]. It is a quotient of the $\mathbb{Z}_8$ classification of the full anomaly.

## 5.1 Locality and tensor product Hilbert space

It is completely standard to express fermions in terms of bosons, using the famous Jordan-Wigner transformation. More precisely, this transformation expresses fermions like $\chi_\ell$ that anticommute for different values of $\ell$ in terms of bosonic variables $\mathbf{b}_j$ that commute for different values of $j$. Furthermore, the total Hilbert space can be taken as a tensor product of local Hilbert spaces $\mathcal{H}_j$ (1.9)

$$\mathcal{H} = \bigotimes_{j=1}^{N} \mathcal{H}_j \tag{5.1}$$

such that $\mathbf{b}_j$ acts only on $\mathcal{H}_j$. In this form, the theory seems to be a bosonic theory with a standard tensor product Hilbert space. However, as we will soon review, the situation is not so simple.

Let us do it more concretely and start with the case of even $L = 2N$. (We will turn to the odd $L$ case in Section 5.4.) The Hilbert space is $2^N$ dimensional. We represent it as a tensor product of $N$ factors of a two-dimensional Hilbert space (5.1).

The local bosonic operators $\mathbf{b}_j$ that act only on $\mathcal{H}_j$ can be expressed in terms of the Pauli matrices. For each $\mathcal{H}_j$, we will use the basis

$$|{\uparrow}\rangle = \begin{pmatrix} 1 \\ 0 \end{pmatrix}, \quad |{\downarrow}\rangle = \begin{pmatrix} 0 \\ 1 \end{pmatrix} \tag{5.2}$$

so that $\sigma^z|{\uparrow}\rangle = |{\uparrow}\rangle$, $\sigma^z|{\downarrow}\rangle = -|{\downarrow}\rangle$. Then, a basis for the full Hilbert space $\mathcal{H}$ is given by $|m_1, m_2, \cdots, m_N\rangle$ with $m_j = {\uparrow}, {\downarrow}$. And $\sigma_j^a$ with $a = x, y, z$ and $j = 1, 2, \cdots, N$ denotes the operator acting as the identity on all $\mathcal{H}_k$ with $k \neq j$ and as $\sigma^a$ on $\mathcal{H}_j$. Crucially, $\sigma_j^a$ commutes with $\sigma_{j'}^b$ for $j \neq j'$.

In terms of these bosonic degrees of freedom, the Jordan-Wigner transformation on a periodic chain is[38]

$$\chi_1 = -\sigma_1^x$$
$$\chi_2 = \sigma_1^y$$
$$\chi_3 = -\sigma_1^z \sigma_2^x$$
$$\chi_4 = \sigma_1^z \sigma_2^y \tag{5.3}$$
$$\vdots$$
$$\chi_{2N-1} = -\sigma_1^z \sigma_2^z \cdots \sigma_{N-1}^z \sigma_N^x$$
$$\chi_{2N} = \sigma_1^z \sigma_2^z \cdots \sigma_{N-1}^z \sigma_N^y \, .$$

It is easy to check that these operators satisfy the correct anticommutation relations $\{\chi_\ell, \chi_{\ell'}\} =$

---

[38]Here the signs are conventional, and we chose them so that the final Ising Hamiltonian in (5.20) is ferromagnetic, rather than antiferromagnetic. One can redefine $\sigma_j^x \to (-1)^j \sigma_j^x, \sigma_j^y \to (-1)^{j+1}\sigma_j^y, \sigma_j^z \to -\sigma_j^z$ to remove these signs. Alternatively, we could have redefined $\chi_\ell \to (-1)^\ell \chi_\ell$ and flipped the overall signs of the fermionic Hamiltonians (3.2) and (3.21).

$2\delta_{\ell,\ell'}$. The inverse transformation is

$$
\begin{aligned}
\sigma_1^x &= -\chi_1\,, & \sigma_1^y &= \chi_2 \\
\sigma_2^x &= -i\chi_1\chi_2\chi_3\,, & \sigma_2^y &= i\chi_1\chi_2\chi_4 \\
&\ \vdots & &\ \vdots \\
\sigma_N^x &= i^{N+1}\chi_1\chi_2\cdots\chi_{2N-2}\chi_{2N-1}\,, & \sigma_N^y &= i^{N-1}\chi_1\chi_2\cdots\chi_{2N-2}\chi_{2N}\,.
\end{aligned}
\tag{5.4}
$$

The expressions (5.3) and (5.4) demonstrate the first subtlety above. If we view the operators $\sigma_j^a$ as local operators, the fermions $\chi_\ell$ are not local. $\chi_\ell$ is a line operator stretching from $\ell$ to the first site $\ell = 1$ (5.3). Conversely, if we view the fermions $\chi_\ell$ as local operators, then the bosons $\sigma_j^x$ and $\sigma_j^y$ and are nonlocal (5.4). However, some operators, e.g.,

$$
\begin{aligned}
\chi_{2j-1}\chi_{2j} &= -i\sigma_j^z\,, & j &= 1,2,\cdots,N\,, \\
\chi_{2j}\chi_{2j+1} &= -i\sigma_j^x\sigma_{j+1}^x\,, & j &= 1,2,\cdots,N-1\,.
\end{aligned}
\tag{5.5}
$$

are local both in terms of the fermions and the bosons. Note that in the last equation we excluded the case $j = N$ because

$$
\chi_{2N}\chi_1 = -\sigma_1^z\sigma_2^z\cdots\sigma_{N-1}^z\sigma_N^y\sigma_1^x = i(\sigma_1^z\sigma_2^z\cdots\sigma_N^z)\sigma_N^x\sigma_1^x\,.
\tag{5.6}
$$

It is similar to the expression in (5.5) except that it is multiplied by the nonlocal operator $\sigma_1^z\sigma_2^z\cdots\sigma_N^z$.

Let us express our symmetry operators in terms of the bosonic degrees of freedom. The internal symmetry operator is

$$
(-1)^F = i^N\chi_1\cdots\chi_{2N} = \sigma_1^z\sigma_2^z\cdots\sigma_N^z\,.
\tag{5.7}
$$

Here we expressed it as a product of the local fermion parity operators in $\mathcal{H}_j$, which are given by $i\chi_{2j-1}\chi_{2j} = \sigma_j^z$.

The time-reversal transformation is particularly simple in this bosonic description. It is independent of the $\sigma$ matrices $\mathcal{T} = \mathcal{K}$, where $\mathcal{K}$ denotes complex conjugation. Clearly, this expression satisfies the defining property $\mathcal{T}\chi_\ell\mathcal{T}^{-1} = (-1)^{\ell+1}\chi_\ell$ in (3.39) and $\mathcal{T}^2 = 1$.

## 5.2 Majorana translation operators

Unlike the action of $(-1)^F$, the action of the translation operator $T_{\text{RR}}$ in (3.69) (or equivalently $T$) is more complicated. First, since $T_{\text{RR}}$ maps $\chi_\ell \to \chi_{\ell+1}$, it cannot lead to a simple action on $\mathcal{H}_j$. Instead, one can consider $T_{\text{RR}}^2$ and hope that it maps $\mathcal{H}_j \to \mathcal{H}_{j+1}$ (with $j \sim j + N$). This is indeed the case for $\sigma_j^z$

$$
T_{\text{RR}}^2\,\sigma_j^z\,T_{\text{RR}}^{-2} = \sigma_{j+1}^z\,.
\tag{5.8}
$$

But this is not true for $\sigma_j^x$ and $\sigma_j^y$. We have

$$
T_{\text{RR}}^2\,\sigma_j^a\,T_{\text{RR}}^{-2} =
\begin{cases}
\sigma_1^z\sigma_{j+1}^a\,, & j = 1,2,\cdots,N-1 \\
\sigma_1^a\sigma_2^z\sigma_3^z\cdots\sigma_N^z\,, & j = N
\end{cases}
\,,\quad a = x,y\,.
\tag{5.9}
$$

Similarly, for the twisted translation $T_{\text{NSNS}}$ in (3.74) (or equivalently $T_{\text{G}}$),

$$T_{\text{NSNS}}^2 \, \sigma_j^z \, T_{\text{NSNS}}^{-2} = \sigma_{j+1}^z \,,$$

$$T_{\text{NSNS}}^2 \, \sigma_j^a \, T_{\text{NSNS}}^{-2} = \begin{cases} \sigma_1^z \sigma_{j+1}^a \,, & j = 1, 2, \cdots, N-1 \\ -\sigma_1^a \sigma_2^z \sigma_3^z \cdots \sigma_N^z \,, & j = N \end{cases} \,, \qquad a = x, y \,. \tag{5.10}$$

We see that while $T_{\text{RR}}$ and $T_{\text{NSNS}}$ act locally on the fermions, they do not act locally on the bosonic degrees of freedom $\sigma_j^a$. And even $T_{\text{RR}}^2$ and $T_{\text{NSNS}}^2$ do not act in the naive way on the bosonic variables.

This action on the bosonic variables has an obvious intuitive explanation. As we emphasized, the Jordan-Wigner transformation (5.3), (5.4) is nonlocal. A local fermion is expressed in terms of a line operator constructed out of the bosons. And a local boson is expressed in terms of a line operator constructed out of the fermions. Then, when the fermionic translation operators $T_{\text{RR}}$ and $T_{\text{NSNS}}$ act on the bosons, they should also translate the line operator, and hence the nonlocal expressions in (5.9) and (5.10).

The fact that $T_{\text{RR}}$ and $T_{\text{NSNS}}$ or even their squares $T_{\text{RR}}^2$ and $T_{\text{NSNS}}^2$ do not simply translate $\sigma_j^a$ means that they do not act in the naive way by mapping the local factors $\mathcal{H}_j \to \mathcal{H}_{j+1}$. However, in the basis with diagonal $\sigma_j^z$, $T_{\text{RR}}^2 |m_1, m_2, \cdots, m_N\rangle$ differs from the naive result $|m_N, m_1, \cdots, m_{N-1}\rangle$ by a phase. For example,

$$\begin{aligned} T_{\text{RR}}^2 \, |\uparrow\uparrow \cdots \uparrow\rangle &= |\uparrow\uparrow \cdots \uparrow\rangle \,, & T_{\text{RR}}^2 \, |\downarrow\downarrow \cdots \downarrow\rangle &= (-1)^{N-1} |\downarrow\downarrow \cdots \downarrow\rangle \,, \\ T_{\text{NSNS}}^2 \, |\uparrow\uparrow \cdots \uparrow\rangle &= |\uparrow\uparrow \cdots \uparrow\rangle \,, & T_{\text{NSNS}}^2 \, |\downarrow\downarrow \cdots \downarrow\rangle &= (-1)^N |\downarrow\downarrow \cdots \downarrow\rangle \,. \end{aligned} \tag{5.11}$$

The normalization of the translation operators in (3.69) and (3.74) is natural also because the phases in (5.11) are only signs. However, we see that even with these choices, there is a nontrivial sign when these operators act on $|\downarrow\downarrow \cdots \downarrow\rangle$.

## 5.3 Summing over the spin structures on the lattice

The Hilbert space and the bosonic variables $\sigma_j^a$ are the same as in the Ising model. However, as we stressed above, the fundamental degrees of freedom $\sigma_j^x$ and $\sigma_j^y$ are nonlocal relative to the fermions $\chi_\ell$. (Note that $\sigma_j^z$ is local (5.5).) In the continuum, the relation between the fermionic theory and the bosonic theory is well understood and will be reviewed in Appendix A.3. The bosonic theory is given by summing the fermionic theory over its spin structures, a procedure that can be thought of as gauging $(-1)^F$ and is known in string theory as the GSO projection.

In order to see that on the lattice, we first express the untwisted Hamiltonian (1.7) and the

twisted Hamiltonian (1.8), with the mass terms (3.41) and (3.54) using the bosonic variables

$$
\begin{aligned}
H &= \frac{i}{2} \sum_{\ell=1}^{2N} (1 + (-1)^\ell m) \chi_{\ell+1} \chi_\ell \\
&= -\frac{1-m}{2} \sum_{j=1}^{N} \sigma_j^z - \frac{1+m}{2} \sum_{j=1}^{N-1} \sigma_{j+1}^x \sigma_j^x + \frac{1+m}{2} (-1)^F \sigma_1^x \sigma_N^x , \\
H_{\mathsf{G}} &= \frac{i}{2} \sum_{\ell=1}^{2N-1} (1 + (-1)^\ell m) \chi_{\ell+1} \chi_\ell - \frac{i(1+m)}{2} \chi_1 \chi_{2N} \\
&= -\frac{1-m}{2} \sum_{j=1}^{N} \sigma_j^z - \frac{1+m}{2} \sum_{j=1}^{N-1} \sigma_{j+1}^x \sigma_j^x - \frac{1+m}{2} (-1)^F \sigma_1^x \sigma_N^x .
\end{aligned}
\tag{5.12}
$$

If not for the factor of $(-1)^F$ in the last term, these two Hamiltonians are the same as the Hamiltonians for the Ising model with or without a $\mathbb{Z}_2$ defect at the link $(N, 1)$. However, the factor of $(-1)^F$ is very important for two reasons. First, $(-1)^F$ is an operator rather than a c-number. And furthermore, it is a nonlocal operator, affecting all the local factors $\mathcal{H}_j$. Second, as we stressed, the bosonic degrees of freedom are nonlocal.

To address these two issues, we imitate the sum over the spin structures of the continuum theory. See [41] for related discussions on the connection between the continuum bosonization/fermionization and the Jordan-Wigner transformation on the lattice. The authors of [92–95] have also considered the two fermionic Hamiltonians $H, H_G$ in (5.12) (as well as their translation symmetries), and the two other bosonic Hamiltonians $H_{\text{Ising}}, H_{\text{tw}}$ that we discuss below. Here we focus more on the global symmetries of the bosonic and fermionic models.

First, we double the system by taking a direct sum of two copies of the Hilbert space $\mathcal{H}$ and denote them $\mathcal{H}_{\text{NSNS}}$ and $\mathcal{H}_{\text{RR}}$

$$
\widetilde{\mathcal{H}} = \mathcal{H}_{\text{NSNS}} \oplus \mathcal{H}_{\text{RR}} .
\tag{5.13}
$$

In the continuum limit, they become the NSNS and the RR Hilbert spaces, as discussed in Section 4.2 and 4.1. We also define an operator $\widetilde{\mathcal{S}} = \begin{pmatrix} 0 & 1 \\ 1 & 0 \end{pmatrix}$ that implements the natural isomorphism $\mathcal{H}_{\text{RR}} \cong \mathcal{H}_{\text{NSNS}}$. (Here and below, the notation with such $2 \times 2$ matrices that act on the Hilbert space $\widetilde{H}$ should be thought of as having $N \times N$ blocks.) We take the Hamiltonian and the translation operator on this bigger Hilbert space $\widetilde{\mathcal{H}}$ to be

$$
\begin{aligned}
\widetilde{H} &= \begin{pmatrix} H_{\mathsf{G}} & 0 \\ 0 & H \end{pmatrix} , \\
\widetilde{T} &= \begin{pmatrix} T_{\text{NSNS}}^2 & 0 \\ 0 & T_{\text{RR}}^2 \end{pmatrix} .
\end{aligned}
\tag{5.14}
$$

Recall that while $T_{\text{RR}}, T_{\text{NSNS}}$ are symmetries of $H$ and $H_{\mathsf{G}}$ at the massless point $m = 0$, only $T_{\text{RR}}^2, T_{\text{NSNS}}^2$ are symmetries of these Hamiltonians for a generic $m$. Thus, $\widetilde{T}$ is a symmetry of $\widetilde{H}$

for any $m$.[39]

In this bigger Hilbert space $\widetilde{\mathcal{H}}$, in addition to the fermion parity operator[40]

$$(-1)^F = \begin{pmatrix} (-1)^F & 0 \\ 0 & (-1)^F \end{pmatrix} \tag{5.15}$$

of the original fermion problem, we gain a new unitary $\mathbb{Z}_2$ symmetry

$$\widetilde{\eta} = \begin{pmatrix} 1 & 0 \\ 0 & -1 \end{pmatrix}. \tag{5.16}$$

Together, $(-1)^F$ and $\widetilde{\eta}$ generate a $\mathbb{Z}_2^F \times \mathbb{Z}_2^\eta$ symmetry of $\widetilde{H}$ in $\widetilde{\mathcal{H}}$.

In terms of the bosonic variables $\sigma_j^a$, the Hamiltonian $\widetilde{H}$ on $\widetilde{\mathcal{H}}$ is not a local Hamiltonian. To obtain a local, bosonic model, we can perform two different projections of the $2^{N+1}$-dimensional Hilbert space $\widetilde{\mathcal{H}}$ to a $2^N$-dimensional subspace. In one case we find the Ising model, and in the other case we find the Ising model with a $\mathbb{Z}_2$ defect. This implements the GSO projection, or equivalently, summing over the spin structures, on the lattice.

### 5.3.1 Ising model

To obtain the Ising model, we project $\mathcal{H}_{\text{NSNS}}$ to the subspace $\mathcal{H}_{\text{NSNS}}^+$ with $(-1)^F = +1$, and project $\mathcal{H}_{\text{RR}}$ to the subspace $\mathcal{H}_{\text{RR}}^-$ with $(-1)^F = -1$. In other words, we project to the $\widetilde{\eta}(-1)^F = 1$ subspace of $\widetilde{\mathcal{H}}$. We denote this $2^N$-dimensional subspace of $\widetilde{\mathcal{H}}$ as

$$\mathcal{H}_{\text{Ising}} = \mathcal{H}_{\text{NSNS}}^+ \oplus \mathcal{H}_{\text{RR}}^-. \tag{5.17}$$

This corresponds to summing over the spin structures in the continuum.

What are the operators that act in $\mathcal{H}_{\text{Ising}}$? The operators $\begin{pmatrix} \sigma_j^a & 0 \\ 0 & \sigma_j^a \end{pmatrix}$ with $a = x, y, z$ act within $\mathcal{H}_{\text{RR}}$ and within $\mathcal{H}_{\text{NSNS}}$. For $a = z$, they commute with the projection, but for $a = x, y$, they take us out of the projected Hilbert space $\mathcal{H}_{\text{Ising}}$. However,

$$X_j = \begin{pmatrix} 0 & \sigma_j^x \\ \sigma_j^x & 0 \end{pmatrix}, \qquad Y_j = \begin{pmatrix} 0 & \sigma_j^y \\ \sigma_j^y & 0 \end{pmatrix} \tag{5.18}$$

---

[39]We can think of the difference between the untwisted problem with $H$ and the twisted problem with $H_G$ as coupling the system to a background $\mathbb{Z}_2$ gauge field. Unlike standard lattice gauge fields, here we place it only on the last link $(L, 1)$. This can be viewed as a classical $\mathbb{Z}_2$ gauge field with vanishing field strength and therefore only its holonomy around the lattice is meaningful and it is represented by $\pm 1$ in the last link. The procedure of doubling the Hilbert space and the later projection that we will perform amounts to making this gauge field dynamical, while keeping its field strength zero. This will be discussed in more detail in [47].

[40]We slightly abuse the notation where we use the same symbol $(-1)^F$ both for the operators acting in the smaller Hilbert spaces $\mathcal{H}_{\text{NSNS}}$ and $\mathcal{H}_{\text{RR}}$ and also in the larger Hilbert space $\widetilde{\mathcal{H}}$. Also, unlike our other operators acting in $\widetilde{\mathcal{H}}$, we do not write $\widetilde{(-1)^F}$ in the left-hand side.

map $\mathcal{H}_{\mathrm{RR}} \leftrightarrow \mathcal{H}_{\mathrm{NSNS}}$ and commute with the projectors. We also define

$$Z_j = \begin{pmatrix} \sigma_j^z & 0 \\ 0 & \sigma_j^z \end{pmatrix}. \tag{5.19}$$

$X_j, Y_j, Z_j$ satisfy the standard commutation relations of Paul matrices.

Now, we can write the Hamiltonian $\widetilde{H}$ in the projected subspace $\mathcal{H}_{\mathrm{Ising}}$ in terms of operators that commute with the projections:[41]

$$H_{\mathrm{Ising}} = \widetilde{H}\Big|_{\mathcal{H}_{\mathrm{Ising}}} = -\frac{1-m}{2} \sum_{j=1}^{N} Z_j - \frac{1+m}{2} \sum_{j=1}^{N} X_{j+1} X_j. \tag{5.20}$$

We recognize it as the Ising model Hamiltonian. Here $\big|_{\mathcal{H}_{\mathrm{Ising}}}$ means the restriction of an operator to the subspace $\mathcal{H}_{\mathrm{Ising}} \subset \widetilde{\mathcal{H}}$.[42]

This analysis also reveals a lattice version of the standard quantum symmetry that arises after gauging (or equivalently, orbifolding). In the continuum, this was first done for bosonic theories in [96]. For a recent discussion of the continuum fermionic theory, see e.g., [37]. Since we projected on $\widetilde{\eta}(-1)^F = +1$, $\widetilde{\eta}$ and $(-1)^F$ are not independent in the projected Hilbert space $\mathcal{H}_{\mathrm{Ising}}$. In other words, we do not have a $\mathbb{Z}_2^F \times \mathbb{Z}_2^\eta$ symmetry in $\mathcal{H}_{\mathrm{Ising}}$, but only a single $\mathbb{Z}_2$

$$\eta = (-1)^F\Big|_{\mathcal{H}_{\mathrm{Ising}}} = \widetilde{\eta}\Big|_{\mathcal{H}_{\mathrm{Ising}}} = \prod_{j=1}^{N} Z_j\Big|_{\mathcal{H}_{\mathrm{Ising}}}. \tag{5.21}$$

It acts as on the new bosonic local operators as

$$\eta Z_j \eta^{-1} = Z_j, \quad \eta X_j \eta^{-1} = -X_j. \tag{5.22}$$

We identify this symmetry $\eta$ as the $\mathbb{Z}_2$ symmetry of the Ising model $H_{\mathrm{Ising}}$.

The bosonic Hilbert space $\mathcal{H}_{\mathrm{Ising}}$ can now be written as $\mathcal{H}_{\mathrm{Ising}}^+ \oplus \mathcal{H}_{\mathrm{Ising}}^-$ with $\pm$ denoting the grading with respect to the $\mathbb{Z}_2$ symmetry $\eta$. We hence obtain

$$\mathcal{H}_{\mathrm{Ising}}^+ = \mathcal{H}_{\mathrm{NSNS}}^+, \quad \mathcal{H}_{\mathrm{Ising}}^- = \mathcal{H}_{\mathrm{RR}}^-. \tag{5.23}$$

This matches with the bosonization relation (A.24) in the continuum.[43]

---

[41]Following on the discussion in footnote 38, suppose we had redefined the Pauli matrices in the Jordan-Wigner transformation (5.3) by $\sigma_j^x \to (-1)^j \sigma_j^x, \sigma_j^y \to (-1)^{j+1}\sigma_j^y, \sigma_j^z \to -\sigma_j^z$. This would have flipped the signs of $\sum_{j=1}^{N-1} \sigma_{j+1}^x \sigma_j^x$ in (5.12). As a result, the final Ising Hamiltonian (5.20) would be in the anti-ferromagnetic phase, rather than the ferromagnetic phase. It is known that the anti-ferromagnetic Ising model flows to the untwisted Ising field theory in the continuum when $N$ is even, and to the twisted version when $N$ is odd. Correspondingly, in this alternative sign convention, our projections below would have depended on the parity of $N$.

[42]The Pauli matrices $X_j, Z_j$ in (5.20) should be understood as the restriction to the projected Hilbert space $\mathcal{H}_{\mathrm{Ising}}$. Here and below we suppress the restriction symbol $\Big|_{\mathcal{H}_{\mathrm{Ising}}}$ for $X_j, Z_j$ to avoid cluttering when there is no potential confusion. Similar comments apply to $\mathcal{H}_{\mathrm{tw}}$ below.

[43]The discussion in Appendix A.3 is more general than the case of the Majorana and Ising CFTs, and hence we use a slightly different notation. The $\mathcal{H}_{\mathrm{NSNS}}^\pm, \mathcal{H}_{\mathrm{RR}}^\pm$ here correspond to $\mathcal{H}_{\mathrm{NS}}^\pm, \mathcal{H}_{\mathrm{R}}^\pm$ in (A.24), and $\mathcal{H}_{\mathrm{Ising}}^\pm$ corresponds to $\mathcal{H}_{\mathrm{untw}}^\pm$ there.

Finally, we discuss the action of the translation operator $\widetilde{T}$ in (5.14), which comes from the square of the Majorana translation operators. We first work in the bigger Hilbert space $\widetilde{\mathcal{H}}$. Using

$$\widetilde{T}\widetilde{\mathcal{S}}\widetilde{T}^{-1} = \begin{pmatrix} 0 & T_{\text{NSNS}}^2 T_{\text{RR}}^{-2} \\ T_{\text{RR}}^2 T_{\text{NSNS}}^{-2} & 0 \end{pmatrix} = \begin{pmatrix} 0 & \sigma_1^z \\ \sigma_1^z & 0 \end{pmatrix}, \tag{5.24}$$

then (5.9) and (5.10) become

$$\widetilde{T} Z_j \widetilde{T}^{-1} = Z_{j+1},$$

$$\widetilde{T} X_j \widetilde{T}^{-1} = \begin{cases} X_{j+1}, & j = 1, 2, \cdots, N-1 \\ X_1 \begin{pmatrix} (-1)^F & 0 \\ 0 & (-1)^{F+1} \end{pmatrix}, & j = N \end{cases} \tag{5.25}$$

Next, we define

$$T_{\text{Ising}} = \widetilde{T}\Big|_{\mathcal{H}_{\text{Ising}}} \tag{5.26}$$

as the restriction of the operator $\widetilde{T}$ to the subspace $\mathcal{H}_{\text{Ising}} \subset \widetilde{\mathcal{H}}$. In the projected Hilbert space $\mathcal{H}_{\text{Ising}}$, (5.25) simplifies to

$$T_{\text{Ising}} Z_j T_{\text{Ising}}^{-1} = Z_{j+1}, \quad T_{\text{Ising}} X_j T_{\text{Ising}}^{-1} = X_{j+1}, \tag{5.27}$$

with $j \sim j + N$. Therefore, in the projected Hilbert space $\mathcal{H}_{\text{Ising}}$, $T_{\text{Ising}}$ generates a standard $\mathbb{Z}_N$ lattice translation symmetry of the Ising model on a closed chain with $N$ sites.

The Ising translation operator can be written in terms of the bosonic variables as [93]

$$T_{\text{Ising}} = t_1^{\text{Ising}} t_2^{\text{Ising}} \cdots t_{N-1}^{\text{Ising}},$$

$$t_j^{\text{Ising}} = \frac{1}{2} \left( X_j X_{j+1} + Y_j Y_{j+1} + Z_j Z_{j+1} + 1 \right), \tag{5.28}$$

where $t_j^{\text{Ising}}$ is known as the SWAP gate acting on the $j$-th and $j+1$-th sites. We can pick a basis for $\mathcal{H}_{\text{Ising}}$ that diagonalizes $Z_j$, and denote it as $|m_1, m_2, \cdots, m_N\rangle_{\text{Ising}}$.[44] Then, it is straightforward to check that $t_j^{\text{Ising}}|\cdots m_j, m_{j+1}\cdots\rangle_{\text{Ising}} = |\cdots m_{j+1}, m_j\cdots\rangle_{\text{Ising}}$, and hence

$$T_{\text{Ising}}|m_1, m_2, \cdots, m_N\rangle_{\text{Ising}} = |m_N, m_1, \cdots, m_{N-1}\rangle_{\text{Ising}}. \tag{5.29}$$

This provides another justification for our phase choices in (3.69) and (3.74).

### 5.3.2 Ising model with a twist

Starting from the $2^{N+1}$-dimensional Hilbert space $\widetilde{\mathcal{H}}$, there is an alternative projection to obtain a local bosonic model, which is the Ising model with a $\mathbb{Z}_2$ defect. Instead of (5.17), we now

---

[44]We note that despite the similarity, the basis we use here differs from the one in (5.11), as they are bases of different Hilbert spaces.

project $\mathcal{H}_{\text{NSNS}}$ to the subspace $\mathcal{H}_{\text{NSNS}}^-$ with $(-1)^F = -1$, and project $\mathcal{H}_{\text{RR}}$ to the subspace $\mathcal{H}_{\text{RR}}^+$ with $(-1)^F = +1$. In other words, we project on the $\widetilde{\eta}(-1)^F = -1$ subspace of $\widetilde{\mathcal{H}}$. We denote this $2^N$-dimensional subspace of $\widetilde{\mathcal{H}}$ as

$$\mathcal{H}_{\text{tw}} = \mathcal{H}_{\text{NSNS}}^- \oplus \mathcal{H}_{\text{RR}}^+ \,. \tag{5.30}$$

The operators $Z_j, X_j$ also act within the Hilbert space $\mathcal{H}_{\text{tw}}$. The Hamiltonian $\widetilde{H}$ restricted to this other projected Hilbert space is

$$H_{\text{tw}} = \widetilde{H}\Big|_{\mathcal{H}_{\text{tw}}} = -\frac{1-m}{2}\sum_{j=1}^{N} Z_j - \frac{1+m}{2}\sum_{j=1}^{N-1} X_{j+1}X_j + \frac{1+m}{2}X_1 X_N \,. \tag{5.31}$$

We recognize this as the Ising Hamiltonian with a twist at the $(N, 1)$-link by the $\mathbb{Z}_2$ symmetry.

Again, in the projected Hilbert space $\mathcal{H}_{\text{tw}}$, $\widetilde{\eta}$ and $(-1)^F$ are not independent, and we do not have a $\mathbb{Z}_2^F \times \mathbb{Z}_2^\eta$ symmetry. Rather, we have a single $\mathbb{Z}_2$ symmetry generated by[45]

$$\eta_{\text{tw}} = (-1)^F\Big|_{\mathcal{H}_{\text{tw}}} = -\widetilde{\eta}\Big|_{\mathcal{H}_{\text{tw}}} = \prod_{j=1}^{N} Z_j\Big|_{\mathcal{H}_{\text{tw}}} \,. \tag{5.32}$$

It acts on $X_j, Z_j$ in the same way as in (5.22). We identify $\eta_{\text{tw}}$ as the standard $\mathbb{Z}_2$ symmetry of the twisted Ising model. We can grade the Hilbert space $\mathcal{H}_{\text{tw}}$ using this $\eta_{\text{tw}}$ symmetry, i.e., $\mathcal{H}_{\text{tw}} = \mathcal{H}_{\text{tw}}^- \oplus \mathcal{H}_{\text{tw}}^+$, and find

$$\mathcal{H}_{\text{tw}}^- = \mathcal{H}_{\text{NSNS}}^- \,, \qquad \mathcal{H}_{\text{tw}}^+ = \mathcal{H}_{\text{RR}}^+ \,. \tag{5.33}$$

This again matches with the bosonization (A.24) in the continuum reviewed in Section A.3.

Next, we define the translation operator in the twisted Ising model as the restriction of the operator $\widetilde{T}$ to the subspace $\mathcal{H}_{\text{tw}} \subset \widetilde{\mathcal{H}}$

$$T_{\text{tw}} = \widetilde{T}\Big|_{\mathcal{H}_{\text{tw}}} \,. \tag{5.34}$$

It is a symmetry of the Hamiltonian $H_{\text{tw}}$ (5.31). Using (5.25), we find that it acts on $Z_j, X_j$ as

$$\begin{aligned} T_{\text{tw}}\, Z_j\, T_{\text{tw}}^{-1} &= Z_{j+1} \,, \\ T_{\text{tw}}\, X_j\, T_{\text{tw}}^{-1} &= \begin{cases} X_{j+1} \,, & j = 1, 2, \cdots, N-1 \\ -X_1 \,, & j = N \end{cases} \,, \end{aligned} \tag{5.35}$$

which is the expected twisted translation that leaves the twisted Ising Hamiltonian (5.31) invariant.

---

[45]In this presentation, the Hilbert space of the Ising model and the Hilbert space of the twisted Ising model are different subspaces of $\widetilde{\mathcal{H}}$. If we consider these two models as different Hamiltonians acting on a smaller, $2^N$-dimensional, Hilbert space, then $\eta$ and $\eta_{\text{tw}}$ are the same operator.

## 5.4 Jordan-Wigner transformation for odd $L$ and the Kramers-Wannier duality defect

So far, we have focused on bosonization of a Majorana chain with even $L$ sites. We now turn to the odd $L$ case, which is more subtle. The resulting bosonic lattice model we obtain is the critical transverse-field Ising model twisted by the Kramers-Wannier duality defect [97, 93, 98, 99, 68].

For odd $L = 2N + 1$, we perform the Jordan-Wigner transformation as

$$
\begin{aligned}
\chi_1 &= -\sigma_1^x \\
\chi_2 &= \sigma_1^y \\
\chi_3 &= -\sigma_1^z \sigma_2^x \\
\chi_4 &= \sigma_1^z \sigma_2^y \\
&\vdots \\
\chi_{2N-1} &= -\sigma_1^z \sigma_2^z \cdots \sigma_{N-1}^z \sigma_N^x \\
\chi_{2N} &= \sigma_1^z \sigma_2^z \cdots \sigma_{N-1}^z \sigma_N^y, \\
\chi_{2N+1} &= \sigma_1^z \sigma_2^z \cdots \sigma_{N-1}^z \sigma_N^z,
\end{aligned}
\tag{5.36}
$$

and extend it periodically for other values of $\ell$.[46] So far, the Hilbert space is $2^N$-dimensional, and the expressions for $\chi_j$ with $j = 1, \cdots, 2N$ are the same as in the even $L$ case (5.3). We still have (5.5), and

$$
\begin{aligned}
\chi_{2N}\chi_{2N+1} &= i\sigma_N^x, \\
\chi_{2N+1}\chi_1 &= -(\sigma_1^z \cdots \sigma_N^z)\sigma_1^x.
\end{aligned}
\tag{5.37}
$$

The Hamiltonians of the system without and with a defect are

$$
\begin{aligned}
H &= \frac{i}{2} \sum_{\ell=1}^{2N+1} \chi_{\ell+1}\chi_\ell \\
&= -\frac{1}{2} \sum_{j=1}^{N} \sigma_j^z - \frac{1}{2} \sum_{j=1}^{N-1} \sigma_j^x \sigma_{j+1}^x + \frac{i}{2}(\sigma_1^z \cdots \sigma_N^z)\sigma_1^x + \frac{1}{2}\sigma_N^x, \\
H_{\mathsf{G}} &= \mathsf{P} H \mathsf{P}^{-1} = \frac{i}{2} \sum_{\ell=1}^{2N} \chi_{\ell+1}\chi_\ell - \frac{i}{2}\chi_1\chi_{2N+1} \\
&= -\frac{1}{2} \sum_{j=1}^{N} \sigma_j^z - \frac{1}{2} \sum_{j=1}^{N-1} \sigma_j^x \sigma_{j+1}^x - \frac{i}{2}(\sigma_1^z \cdots \sigma_N^z)\sigma_1^x + \frac{1}{2}\sigma_N^x.
\end{aligned}
\tag{5.38}
$$

Neither $H$ or $H_{\mathsf{G}}$ is local with respect to the bosonic variables of the Pauli matrices. (Note that there is no mass term (3.41) for odd $L$.)

---

[46]With this choice, $\mathcal{C} = (-i)^N$. We can change its sign by flipping the signs of all the fermions. Again, we take $\mathcal{T} = \mathcal{K}$ without any modifications.

To proceed, as in Section 5.3,[47] we double the system by taking a direct sum of two copies of the Hilbert space $\mathcal{H}$ for odd $L$ and denote them $\mathcal{H}_{\mathrm{RNS}}$ and $\mathcal{H}_{\mathrm{NSR}}$. We denote the resulting $2^{N+1}$-dimensional Hilbert space by $\mathcal{H}_{\mathrm{D}}$:

$$\mathcal{H}_{\mathrm{D}} = \mathcal{H}_{\mathrm{RNS}} \oplus \mathcal{H}_{\mathrm{NSR}} \, . \tag{5.39}$$

(The subscript D will be justified soon.) We consider the Hamiltonian

$$H_{\mathrm{D}} = \begin{pmatrix} H_{\mathrm{G}} & 0 \\ 0 & H \end{pmatrix} , \tag{5.40}$$

in this $2^{N+1}$-dimensional Hilbert space $\mathcal{H}_{\mathrm{D}}$. One way to think about this system is by adding another spin, labeled by $j = N + 1$, to the original ones and then

$$H_{\mathrm{D}} = -\frac{1}{2} \sum_{j=1}^{N} \sigma_j^z - \frac{1}{2} \sum_{j=1}^{N-1} \sigma_j^x \sigma_{j+1}^x - \frac{i}{2}(\sigma_1^z \cdots \sigma_N^z)\sigma_1^x \sigma_{N+1}^z + \frac{1}{2}\sigma_N^x \tag{5.41}$$

As in (5.18), (5.19) for the even $L$ case, we redefine the bosonic variables to

$$\begin{aligned}
X_j &= \sigma_j^x \sigma_{N+1}^x , & Y_j &= \sigma_j^y \sigma_{N+1}^x , & Z_j &= \sigma_j^z , & j &= 1, \cdots, N , \\
X_{N+1} &= -\sigma_{N+1}^x , & Y_{N+1} &= -(\sigma_1^z \sigma_2^z \cdots \sigma_N^z)\sigma_{N+1}^y , & Z_{N+1} &= \sigma_1^z \sigma_2^z \cdots \sigma_N^z \sigma_{N+1}^z .
\end{aligned} \tag{5.42}$$

It is easy to check that these variables satisfy the standard relations of $N + 1$ decoupled Pauli matrices. In terms of these bosonic variables in the bigger Hilbert space, the Hamiltonian is

$$H_{\mathrm{D}} = -\frac{1}{2} \sum_{j=1}^{N} Z_j - \frac{1}{2} \sum_{j=1}^{N} X_j X_{j+1} - \frac{1}{2} X_1 Y_{N+1} \, . \tag{5.43}$$

Locally, away from the link $(N + 1, 1)$, this is the Hamiltonian for the critical transverse-field Ising model on $N + 1$ sites. At the $(N + 1, 1)$ link, there is a modification, which represents a local defect. In fact, it is precisely the Kramers-Wannier duality defect D in [97,93–95,98,99,68].

In the continuum limit, this lattice system becomes the Ising model with a topological line defect $\mathcal{D}$ associated with the Kramers-Wannier duality of the continuum Ising CFT. The latter has been studied extensively by various authors, including [76–79]. From the modern perspective on generalized global symmetries in relativistic quantum field theory [100, 81], this topological defect and its corresponding operator implement a non-invertible global symmetry in the Ising CFT.

---

[47]As in footnote 39, we can think of the two systems with or without the twist as differing by coupling to a background $\mathbb{Z}_2$ gauge field. However, unlike the situation of even $L$, for odd $L$, the system does not have such a $\mathbb{Z}_2$ global symmetry. Therefore, we can place a spatial background $\mathbb{Z}_2$ gauge field by flipping the sign of the fermion term on the last link $(L, 1)$, but since there is no operator G for that symmetry, we cannot introduce similar gauge fields in the time direction. Related to that, below, we will not perform a projection on the large Hilbert space $\mathcal{H}_{\mathrm{D}}$. As a result, the combined system should be thought of as having a defect of a non-invertible symmetry. This will be discussed in more detail in [47].

One way to see that, is to examine the spectrum. The spectrum of the Hamiltonian (5.43) can be found exactly using (5.39). We did it using the fermionic description in Section 4.3. In the continuum limit, it consists of Virasoro primaries with conformal weights $(h_\mathrm{L}, h_\mathrm{R})$:

$$\mathcal{H}_\mathrm{D} \to \mathcal{H}_\mathcal{D} = \left(\frac{1}{16}, 0\right) \oplus \left(\frac{1}{16}, \frac{1}{2}\right) \oplus \left(0, \frac{1}{16}\right) \oplus \left(\frac{1}{2}, \frac{1}{16}\right), \tag{5.44}$$

where $\mathcal{H}_\mathcal{D}$ is the defect Hilbert space associated with the duality defect $\mathcal{D}$ in the continuum Ising CFT. This is also consistent with the results of [98, 99]. Note that the momentum operators $P$ take values in $\frac{\mathbb{Z}}{2} \pm \frac{1}{16}$, consistent with the spin selection rule in [68, 79, 88].

The fact that this lattice bosonic system flows in the continuum limit to the Ising CFT with a duality defect fits our general picture here. As we discussed in Section 3.1.3, we can view the odd $L = 2N + 1$ Majorana chain as its even $L = 2N$ counterpart with a translation symmetry defect inserted. Furthermore we have argued that the translation symmetry operator leads to an emanant $(-1)^{F_\mathrm{L}}$ internal symmetry. Indeed, in the continuum, the NSNS or the RR theory with a $(-1)^{F_\mathrm{L}}$ defect is the same as the RNS or the NSR theory. In the construction above, we took a direct some of the NSR and the RNS theory. This can be thought of as a direct sum of the RR and the NSNS theories, each with a translation defect. Therefore, it should flow in the continuum limit to the direct some of these theories with a $(-1)^{F_\mathrm{L}}$ defect.

This picture is consistent with the known facts about the continuum theory. First, as shown in [36, 39, 40, 101], in the Ising CFT, the non-invertible duality defect $\mathcal{D}$ arises via bosonization from the chiral fermion parity defect $(-1)^{F_\mathrm{L}}$. Also, unlike the discussion in Section 5.3 for the even $L$ case, here we do not need to perform any projection for odd $L$. This is consistent with the expectation in the continuum, where the defect Hilbert space $\mathcal{H}_\mathcal{D}$ associated with the non-invertible duality defect of the bosonic Ising CFT is related to the fermionic Hilbert spaces of the Majorana CFT as $\mathcal{H}_\mathcal{D} = \mathcal{H}_\mathrm{RNS} \oplus \mathcal{H}_\mathrm{NSR}$.

We will discuss more about this non-invertible symmetry of the lattice Ising model in the following Section and in [47].

## 5.5 Summary of bosonization

The continuum Ising CFT is known to have three topological defects: the trivial one, the $\mathbb{Z}_2$ defect, and the Kramers-Wannier duality defect [76–79]. On the other hand, the free, massless Majorana CFT has a $\mathbb{Z}_2 \times \mathbb{Z}_2^f$ global symmetry, associated with four topological defects leading to the NSNS, RR, NSR, and RNS Hilbert spaces. The relations between these two continuum theories and their defects are discussed in [36–39].

The lattice transverse-field Ising model is also known to have three topological defects: [97, 93, 98, 99, 68]. (The duality defect is only topological at the critical point.) In Sections 5.3 and 5.4 we showed the relation of these lattice defects in the bosonic Ising model to the fermionic Majorana chain. More specifically, starting with the Majorana chain with even number $L$ of sites, the lattice bosonization leads to the Ising model without or with the $\mathbb{Z}_2$ defect, while for odd $L$ we find the

Ising model with the Kramers-Wannier duality defect. Below we summarize the discussion in this section.

### 5.5.1 Even $L$

We start with the Hamiltonians $H$ and $H_G$ without and with a defect in (5.12) for even $L = 2N$ sites. They act on the $2^N$-dimensional Hilbert spaces denoted by $\mathcal{H}_{RR}, \mathcal{H}_{NSNS}$, respectively. We then take the direct sum $\widetilde{\mathcal{H}} = \mathcal{H}_{NSNS} \oplus \mathcal{H}_{RR}$ of these two Hilbert spaces, and perform the Jordan-Wigner transformation.

The $2^{N+1}$-dimensional Hilbert space $\widetilde{\mathcal{H}}$ decomposes into four sectors, each is $2^{N-1}$-dimensional. These four sectors can be written in two equivalent ways using (5.23) and (5.33):

$$
\begin{aligned}
\widetilde{\mathcal{H}} &= \mathcal{H}_{NSNS}^+ && \oplus \, \mathcal{H}_{NSNS}^- && \oplus \mathcal{H}_{RR}^+ && \oplus \mathcal{H}_{RR}^- \\
&= \mathcal{H}_{Ising}^+ && \oplus \, \mathcal{H}_{tw}^- && \oplus \mathcal{H}_{tw}^+ && \oplus \mathcal{H}_{Ising}^- \, .
\end{aligned}
\tag{5.45}
$$

In the continuum limit, the theory flows to the Ising CFT and then the left-moving and the right-moving Virasoro representations of these Hilbert spaces are

$$
\begin{aligned}
\mathcal{H}_{NSNS}^+ = \mathcal{H}_{Ising}^+ && \rightarrow && \left(0, 0\right) \oplus \left(\frac{1}{2}, \frac{1}{2}\right) \\
\mathcal{H}_{NSNS}^- = \mathcal{H}_{tw}^- && \rightarrow && \left(0, \frac{1}{2}\right) \oplus \left(\frac{1}{2}, 0\right) \\
\mathcal{H}_{RR}^+ = \mathcal{H}_{tw}^+ && \rightarrow && \left(\frac{1}{16}, \frac{1}{16}\right) \\
\mathcal{H}_{RR}^- = \mathcal{H}_{Ising}^- && \rightarrow && \left(\frac{1}{16}, \frac{1}{16}\right)
\end{aligned}
\tag{5.46}
$$

where the arrows denote the continuum limits. Even though $\mathcal{H}_{RR}^+$ and $\mathcal{H}_{RR}^-$ transform the same under the left-moving and the right-moving Virasoro algebras, they are viewed as different Hilbert spaces, as they have different $(-1)^F$ eigenvalues.

The Hamiltonian $\widetilde{H}$ on $\widetilde{\mathcal{H}}$ enjoys a $\mathbb{Z}_2^F \times \mathbb{Z}_2^\eta$ symmetry generated by $(-1)^F$ and $\widetilde{\eta}$. (This symmetry was referred to as the "categorical symmetry" in [102].) However, as emphasized above, $\widetilde{H}$ on this big Hilbert space $\widetilde{\mathcal{H}}$ is not a local 1+1d Hamiltonian in terms of the bosonic spin variables $\sigma_j^a$. To obtain a local 1+1d system, we performed projections to $\widetilde{\eta}(-1)^F = \pm 1$, leading to the untwisted and twisted Ising models on the $2^N$-dimensional Hilbert spaces $\mathcal{H}_{Ising}$ and $\mathcal{H}_{tw}$, respectively. After the projection on the Hilbert spaces $\mathcal{H}_{Ising}$ or $\mathcal{H}_{tw}$, there is only a single $\mathbb{Z}_2$ symmetry generated by either $\eta$ or $\eta_{tw}$.

To conclude, we either have a non-local Hamiltonian $\widetilde{H}$ with a $\mathbb{Z}_2^F \times \mathbb{Z}_2^\eta$ symmetry in a $2^{N+1}$-dimensional Hilbert space $\widetilde{\mathcal{H}}$, or a local 1+1d Hamiltonian ($H_{Ising}$ or $H_{tw}$) with only a single $\mathbb{Z}_2$ symmetry in a smaller $2^N$-dimensional Hilbert space. And of course, we can also have local fermionic theories (with or without a $(-1)^F$ twist), again, with a single $\mathbb{Z}_2$ symmetry $(-1)^F$.

Alternatively, each of the four sectors can be realized as the Hilbert space of a 2+1d TQFT with a choice of a boundary condition and an anyon line inserted in the bulk [103–105]. In our case, of the Ising model, the bulk TQFT can be a $\mathbb{Z}_2$ gauge theory. From a more modern perspective, it

means that the theory on the big Hilbert space is a relative theory at the boundary of a 2+1d bulk system, rather than an absolute 1+1d theory. See [98, 106, 86, 39, 102, 107–112, 88, 113] for this bulk perspective for the different sectors of the continuum Ising CFT.

### 5.5.2 Odd $L$

For odd $L = 2N + 1$, we start with the Hamiltonians $H$ and $H_\mathsf{G}$ in (5.38). They act on the $2^N$-dimensional Hilbert spaces denoted by $\mathcal{H}_\mathrm{NSR}, \mathcal{H}_\mathrm{RNS}$. In the continuum limit, they flow to the Majorana CFT in the NSR and RNS Hilbert spaces with the following left- and right-moving Virasoro primaries:

$$
\begin{aligned}
\mathcal{H}_\mathrm{NSR} \quad &\rightarrow \quad \left(0, \frac{1}{16}\right) \oplus \left(\frac{1}{2}, \frac{1}{16}\right) \\
\mathcal{H}_\mathrm{RNS} \quad &\rightarrow \quad \left(\frac{1}{16}, 0\right) \oplus \left(\frac{1}{16}, \frac{1}{2}\right)
\end{aligned}
\tag{5.47}
$$

where again, the arrows denote the continuum limits.

We then take the direct sum of these two Hilbert spaces $\mathcal{H}_\mathsf{D} = \mathcal{H}_\mathrm{NSR} \oplus \mathcal{H}_\mathrm{RNS}$, and perform a Jordan-Wigner transformation to obtain a local Hamiltonian $H_\mathsf{D}$ in terms of the bosonic variables. We do not perform a projection for odd $L$, and the final Hilbert space $\mathcal{H}_\mathsf{D}$ is $2^{N+1}$-dimensional. The new Hamiltonian $H_\mathsf{D}$ describes the transverse-field Ising model on $N + 1$ sites with a duality defect. In the continuum limit, it flows to the Hilbert space of the Ising CFT with a duality defect $\mathcal{D}$ whose Virasoro representations are

$$
\mathcal{H}_\mathsf{D} \quad \rightarrow \quad \left(0, \frac{1}{16}\right) \oplus \left(\frac{1}{2}, \frac{1}{16}\right) \oplus \left(\frac{1}{16}, 0\right) \oplus \left(\frac{1}{16}, \frac{1}{2}\right) .
\tag{5.48}
$$

# 6 Non-invertible lattice translation of the transverse-field Ising model

In Section 5.4, we found a duality *defect* D in the transverse-field Ising model, which becomes the non-invertible Kramers-Wannier duality defect $\mathcal{D}$ in the continuum limit. Can we find the corresponding conserved *operator* on the lattice?

To proceed, we return to the case of the Majorana chain with even $L = 2N$ in Section 5.3. We will see that a byproduct of the lattice version of the GSO projection is that it leads to a conserved operator that mixes with the lattice translation of the critical transverse field Ising model.

Under the Jordan-Wigner transformation, each pair of Majorana fermions $\chi_\ell$ (with $\ell = 1, 2, \cdots, 2N$) gives rise to a local Hilbert space $\mathcal{H}_j$ (with $j = 1, 2, \cdots, N$). We saw that the square of the fermion lattice translation operator, $\widetilde{T} = \begin{pmatrix} T^2_\mathrm{NSNS} & 0 \\ 0 & T^2_\mathrm{RR} \end{pmatrix}$ (defined in (5.14)) acts as the ordinary translation operator $T_\mathrm{Ising}$ in the Ising model when restricted to $\mathcal{H}_\mathrm{Ising}$. Similar comments apply to the twisted Ising model.

Let us consider the Majorana translation operator by one site

$$\widetilde{T}_{\text{Maj}} = \begin{pmatrix} T_{\text{NSNS}} & 0 \\ 0 & T_{\text{RR}} \end{pmatrix} : \qquad \chi_\ell \to \chi_{\ell+1} \,. \tag{6.1}$$

(It is denoted as $\widetilde{T}_{\text{Maj}}$ to remind us that it acts in the large Hilbert space $\widetilde{\mathcal{H}}$.) Since

$$\widetilde{T} = \widetilde{T}_{\text{Maj}}^2 \,, \tag{6.2}$$

we can think of $\widetilde{T}_{\text{Maj}}$ as roughly the square root of the translation operator $\widetilde{T}$. Clearly, $\widetilde{T}_{\text{Maj}}$ is a symmetry of the Hamiltonian $\widetilde{H}$ only at the massless point $m = 0$. Also, because of (3.72) and (3.77):

$$\begin{aligned} (-1)^F \, T_{\text{NSNS}} &= T_{\text{NSNS}} \, (-1)^F \,, \\ (-1)^F \, T_{\text{RR}} &= -T_{\text{RR}}(-1)^F \,, \end{aligned} \tag{6.3}$$

(where the minus sign in the second line was interpreted as an anomaly in (3.73)), $\widetilde{T}_{\text{Maj}}$ does not commute with $\widetilde{\eta}(-1)^F$. Therefore, the operator $\widetilde{T}_{\text{Maj}}$ does not act within the projected Hilbert spaces $\mathcal{H}_{\text{Ising}}$ or $\mathcal{H}_{\text{tw}}$.

Instead, we introduce the operator[48]

$$\widetilde{\mathsf{D}} = \frac{1}{2}(1 + \widetilde{\eta})\widetilde{T}_{\text{Maj}} = \begin{pmatrix} T_{\text{NSNS}} & 0 \\ 0 & 0 \end{pmatrix} \,, \tag{6.4}$$

in the big Hilbert space $\widetilde{\mathcal{H}}$. This new operator commutes with $\widetilde{H}$ when $m = 0$, and obeys $\widetilde{\mathsf{D}}^2 = \frac{1}{2}(1 + \widetilde{\eta})\widetilde{T}$.

Since $\widetilde{\mathsf{D}}$ commutes with $\widetilde{\eta}(-1)^F$, it acts within the projected Hilbert spaces $\mathcal{H}_{\text{Ising}}$ or $\mathcal{H}_{\text{tw}}$. This motivates us to define

$$\mathsf{D} = \widetilde{\mathsf{D}}\Big|_{\mathcal{H}_{\text{Ising}}} = \begin{pmatrix} T_{\text{NSNS}} & 0 \\ 0 & 0 \end{pmatrix}\Big|_{\mathcal{H}_{\text{Ising}}} \tag{6.5}$$

as the restriction of $\mathsf{D}$ to the projected Hilbert space $\mathcal{H}_{\text{Ising}}$. We can similarly define the symmetry operator $\widetilde{\mathsf{D}}\Big|_{\mathcal{H}_{\text{tw}}}$ in the twisted Ising theory.

The operator $\mathsf{D}$ of (6.5) has a kernel – the states with $\eta = -1$. In the orthogonal complement of the kernel, i.e., the states with $\eta = +1$, it acts as a unitary symmetry operator. Such a transformation is known as a partial isometry.

Although this construction was motivated by the fermion theory, it can be discussed entirely in the bosonic transverse-field Ising model. We start with the Ising Hamiltonian

$$H_{\text{Ising}} = -\frac{1-m}{2}\sum_{j=1}^{N} Z_j - \frac{1+m}{2}\sum_{j=1}^{N} X_{j+1}X_j \,. \tag{6.6}$$

---

[48]We can write (6.4) as $\widetilde{\mathsf{D}} = \frac{1}{2}(1 + \widetilde{\eta})\sqrt{\widetilde{T}}$ to make it clear that it is related to "half-translation."

with Pauli operators $Z_j, X_j$ acting in a $2^N$-dimensional tensor product Hilbert space, and define the operator D as[49]

$$\mathsf{D} = e^{-\frac{2\pi i N}{8}} (\mathsf{d}_1^z \mathsf{d}_1^x)(\mathsf{d}_2^z \mathsf{d}_2^x) \cdots (\mathsf{d}_{N-1}^z \mathsf{d}_{N-1}^x) \mathsf{d}_N^z \times \frac{1+\eta}{2},$$

$$\mathsf{d}_j^z = e^{\frac{i\pi}{4} Z_j} = \frac{1+iZ_j}{\sqrt{2}} \quad, \quad \mathsf{d}_j^x = e^{\frac{i\pi}{4} X_j X_{j+1}} = \frac{1+iX_j X_{j+1}}{\sqrt{2}} \quad, \quad \eta = \prod_{j=1}^{N} Z_j. \tag{6.7}$$

Note that $\frac{1+\eta}{2}$ is a projection operator on the states with $\eta = +1$. This operator obeys the following algebra[50]

$$\mathsf{D} Z_j = X_j X_{j+1} \mathsf{D}, \qquad \mathsf{D} X_j X_{j+1} = Z_{j+1} \mathsf{D}, \quad j = 1, 2, \cdots, N. \tag{6.8}$$

Hence, it implements the Kramers-Wannier duality on a closed periodic chain in a translationally invariant way. Indeed, this expression holds true for all $j = 1, \cdots, N$ with $X_{N+1} = X_1, Z_{N+1} = Z_1$.

To summarize, at the critical point $m = 0$, D commutes with the Ising Hamiltonian:[51]

$$\mathsf{D}\, H_{\text{Ising}} = H_{\text{Ising}}\, \mathsf{D}, \quad \text{for} \quad m = 0. \tag{6.9}$$

One can add to $H_{\text{Ising}}$ the deformation

$$-h \sum_{j=1}^{N} Z_{j+1} Z_j - h \sum_{j=1}^{N} X_{j+2} X_j \tag{6.10}$$

which preserves the non-invertible symmetry D for any $h$. This term is locally related to the four-Fermi term $\sum_\ell \chi_\ell \chi_{\ell+1} \chi_{\ell+2} \chi_{\ell+3}$ via the Jordan-Wigner transformation. See [57] for discussions of this deformed bosonic lattice model.

We see that D is a symmetry of the transverse-field Ising model (6.6) at the critical point. However, D is not an ordinary symmetry. It is not invertible (i.e., $\mathsf{D}^{-1}$ does not exist) because it

---

[49]Under the Jordan-Wigner transformation, these local factors are related to those in (3.17) as $\mathsf{d}_j^z = t_{2j-1}^{-1}$ for $j = 1, 2, \cdots, N$ and $\mathsf{d}_j^x = t_{2j}^{-1}$ for $j = 1, 2, \cdots, N-1$.

[50]On the $2^N$-dimensional tensor product Hilbert space for a closed, periodic Ising chain in 1+1d dimensions, the Kramers-Wannier duality cannot be implemented by a unitary transformation. To see that, suppose there is a unitary operator $U_{\text{KW}}$ such that $U_{\text{KW}} Z_j U_{\text{KW}}^{-1} = X_j X_{j+1}$ for all $j = 1, \cdots, N$, but then $U_{\text{KW}} \eta U_{\text{KW}}^{-1} = U_{\text{KW}}(\prod_{j=1}^{N} X_j X_{j+1}) U_{\text{KW}}^{-1} = 1$, which is a contradiction since the $\mathbb{Z}_2$ operator $\eta$ is nontrivial. (See [114] for an alternative realization of the Kramers-Wannier duality in terms of a unitary operator for the 1+1d edge mode of a 2+1d bulk system.) In contrast, as we will soon see, our D is not invertible, and in particular not unitary.

[51]When we discuss the symmetries, we should distinguish between symmetries of either of the fermionic models with Hamiltonians $H$ and $H_{\text{G}}$ and symmetries of the Ising or twisted Ising model. At the critical point, the two fermionic models have new symmetries $T_{\text{RR}}$ and $T_{\text{NSNS}}$ respectively. These symmetries are standard invertible symmetries. (This was discussed in a context similar to ours in [93–95].) However, it is important that these are not symmetries of the two bosonic models, Ising and twisted Ising. In contrast, our operator D of the Ising model is non-invertible and is a symmetry of $H_{\text{Ising}}$ at the critical point. (Similar comments apply to the twisted Ising model discussed below.) These operators are different and are related as in (6.4) and (6.5).

has a nontrivial kernel. It is a non-invertible translation symmetry of the critical Ising model.[52] In contrast, the operator $(\mathsf{d}_1^z \mathsf{d}_1^x) \cdots (\mathsf{d}_{N-1}^z \mathsf{d}_{N-1}^x) \mathsf{d}_N^z$ (which has been discussed in, for example, [67,69,115,74]) is unitary, but it does not commute with the Ising translation operator $T_{\text{Ising}}$ or the Ising Hamiltonian $H_{\text{Ising}}$ for any $m$.

In the projected Hilbert space $\mathcal{H}_{\text{Ising}}$, the non-invertible translation operator D and the $\mathbb{Z}_2$ symmetry $\eta$ obey the following algebra

$$\mathsf{D}^2 = \frac{1}{2}(1 + \eta)\, T_{\text{Ising}}\,,$$

$$\eta^2 = 1\,, \quad \mathsf{D}\,\eta = \eta\,\mathsf{D} = \mathsf{D}\,, \tag{6.11}$$

$$T_{\text{Ising}}^N = 1\,, \quad T_{\text{Ising}}\,\mathsf{D} = \mathsf{D}\,T_{\text{Ising}}\,, \quad T_{\text{Ising}}\,\eta = \eta\,T_{\text{Ising}}\,.$$

As a result, we also have $\mathsf{D}^{2N} = \frac{1}{2}(1+\eta)$. One can similarly consider $\mathsf{D}^\dagger = \mathsf{D}T_{\text{Ising}}^{-1}$, which satisfies $\mathsf{D}\mathsf{D}^\dagger = \mathsf{D}^\dagger\mathsf{D} = \frac{1}{2}(1+\eta)$.

The algebra (6.11) is reminiscent of the fusion algebra

$$\mathcal{D}^2 = 1 + \eta \qquad , \qquad \mathcal{D}\eta = \eta\mathcal{D} = \mathcal{D} \qquad , \qquad \eta^2 = 1 \tag{6.12}$$

of the non-invertible symmetry operator $\mathcal{D}$ of the continuum Ising CFT at the critical point [76–79], which is associated with the Kramers-Wannier duality. (Recall that we use D for the lattice operator and $\mathcal{D}$ for the continuum operator.) Soon, we will explain the relation between them.

The main difference between the lattice D and the continuum $\mathcal{D}$ is that D mixes with the lattice translation operator $T_{\text{Ising}}$ in (5.28). For the low-lying states in the large $N$ limit, $T_{\text{Ising}} \sim 1$, and therefore, the lattice algebra coincides with the continuum one up to rescaling.

Let us compare our algebra with related lattice algebras in the literature. In [116], the authors realize the fusion algebra (6.12) in a lattice model with a non-tensor-factorized Hilbert space. More specifically, their Hilbert space is a *direct sum* $\mathcal{H}_{\text{site}} \oplus \mathcal{H}_{\text{link}}$ of the Hilbert space of spins on the sites $\mathcal{H}_{\text{site}}$ and another one $\mathcal{H}_{\text{link}}$ on the links . This is to be contrasted with our setup, where $\mathcal{H}_{\text{Ising}}$ is a tensor product of local Hilbert spaces. So one either has an internal non-invertible algebra (6.12) on a non-tensor-factorized Hilbert space, or a non-invertible algebra mixing with the lattice translation (6.11) on a tensor product Hilbert space. Our non-invertible operator D is also different from the one in [68,70–73,117]. The authors of these papers considered a map $\mathcal{N}$ from one Hilbert space of spins on the sites, to another one with spins on the links. Similarly, there is another map $\mathcal{N}^\dagger$ that maps the Hilbert space on the links to that on the sites.

We claim that just as $(-1)^{F_{\text{L}}}$ emanates from lattice translation in the massless fermion problem, the non-invertible global symmetry of the continuum Ising CFT $\mathcal{D}$ emanates from the non-invertible translation symmetry D of the transverse field Ising model.[53] Consequently, the contin-

---

[52]Instead of D, one might be tempted to consider the conserved operator $\begin{pmatrix} T_{\text{NSNS}} & 0 \\ 0 & 1 \end{pmatrix} = \mathsf{D} + \frac{1-\eta}{2}$, which is unitary, and in particular, invertible. However, this is not a valid invertible symmetry, as it does not map local operators to local operators.

[53]There are other microscopic realizations of the Kramers-Wannier topological line in the Ising CFT. See [68,116] for such a realization in the statistical Ising model. The generalization of the anyonic chain [118] gives a Hamiltonian lattice model with a non-invertible conserved operator.

uum non-invertible symmetry $\mathcal{D}$ is not an emergent symmetry, but an emanant symmetry. It arises from the exact non-invertible translation symmetry of the critical transverse field Ising model D. And as all emanant symmetries [32], it is exact in the continuum limit and is not violated even by irrelevant operators.

Let us give a supporting argument for this claim. In the continuum, it is known that the chiral fermion parity $(-1)^{F_L}$ of the massless Majorana CFT becomes the non-invertible global symmetry $\mathcal{D}$ of the bosonic Ising CFT after summing over the spin structures [39, 40, 101].[54] As discussed in Section 4, the chiral fermion parity $(-1)^{F_L}$ of the continuum Majorana CFT emanates from the Majorana translation operators $T_{RR}, T_{NSNS}$ (see (4.13) and (4.29)). The new operator D is the Majorana translation $T_{NSNS}$ after we sum over the spin structures on the lattice (see Section 5.3). Therefore, the non-invertible global symmetry $\mathcal{D}$ of the continuum Ising CFT emanates from the non-invertible translation symmetry D on the lattice. More precisely, on the low-lying states, we have

$$D = \frac{1}{\sqrt{2}} \mathcal{D} e^{\frac{2\pi i P}{2N}} , \qquad (6.13)$$

where $P = h_L - h_R$ is the continuum translation operator of the Ising CFT.[55]

We stress that the relation (6.13) between the lattice operator D and the continuum operators $\mathcal{D}$ and $P$ is exact in the low-energy spectrum. It does not have any $\mathcal{O}(1/N)$ finite-size corrections.

We leave a thorough investigation of this non-invertible translation symmetry operator D and its defect (5.43) in the transverse-field Ising model for an upcoming work [47].

# 7   Conclusions

In this paper, we studied the 1+1d closed Majorana chain and focused on its symmetries and anomalies. Our analysis was divided into three cases:

- even $L$, which we refer to as RR,

- even $L$ with a fermion parity defect, which we refer to as NSNS, and

- odd $L$, which we refer to as NSR (it is closely related to odd $L$ with a fermion parity defect, which we refer to as RNS).

The symmetries we discussed are lattice translation, parity, time reversal, and fermion parity $(-1)^F$.

The anomalies of these lattice symmetries appear as projective phases in the algebras realized on these Hilbert spaces. Some of these phases can be moved around by phase redefinitions of the operators, but some others cannot. In order to respect spatial locality, we restrict the allowed

---

[54]Alternatively, the non-invertible Kramers-Wannier symmetry in the Ising CFT can also be constructed by gauging the $\mathbb{Z}_2$ global symmetry in half of the spacetime and imposing a topological Dirichlet boundary condition, a construction known as half gauging [119].

[55]Even though the lattice D and the continuum $\mathcal{D}$ are conserved operators, since they are non-invertible, without further input, they do not have a natural normalization. Correspondingly, we could simplify (6.13) by rescaling either D or $\mathcal{D}$ by a factor of $\sqrt{2}$.

phase redefinitions of operators that are products of local factors, such as the translation operator, to be of the form (3.68)

$$\mathcal{S} \to e^{i\alpha L + i\beta} \mathcal{S} \,. \tag{7.1}$$

A phase redefinition satisfying this restriction can be thought of as adding a local counterterm on the symmetry line operator.

For even $L$, the algebras of the rescaled translation operators $T_{\text{RR}}$ and $T_{\text{NSNS}}$ with other symmetries are presented in (3.72) and (3.77). The projective phases of the former are interpreted as the anomalies on the lattice.

For odd $L$, we find that there is no local phase redefinition of the lattice translation operator such that for all $L$, $T^L$ is the identity operator. The best one can achieve is $T^L_{\text{odd}} = e^{\frac{2\pi i n}{16}}$ as in (3.80), with $n$ an odd integer that depends on the choice of the counterterm. This is a more subtle anomaly.

We then focused on the specific case of the free fermion Hamiltonians (1.7), (1.8)

$$
\begin{aligned}
H &= \frac{i}{2} \sum_{\ell=1}^{L} \chi_{\ell+1} \chi_\ell = \frac{i}{2} \sum_{\ell=1}^{L-1} \chi_{\ell+1} \chi_\ell + \frac{i}{2} \chi_1 \chi_L \,, \\
H_{\mathsf{G}} &= \frac{i}{2} \sum_{\ell=1}^{L-1} \chi_{\ell+1} \chi_\ell - \frac{i}{2} \chi_1 \chi_L \,.
\end{aligned}
\tag{7.2}
$$

The continuum limits of $H$ with even $L$, $H_{\mathsf{G}}$ with even $L$, $H$ with odd $L$, and $H_{\mathsf{G}}$ with odd $L$ correspond to the Majorana CFT with RR, NSNS, NSR, and RNS boundary conditions, respectively (and hence, the terminology mentioned above). That is, they correspond to fermions with periodic, anti-periodic, and mixed boundary conditions.

While the non-chiral fermion parity $(-1)^F$ is a manifest internal symmetry of the Majorana chain, the origin of the chiral fermion parity $(-1)^{F_L}$ is more subtle. It is not an emergent symmetry; rather it emanates from the lattice translation symmetry. The precise relations between the chiral fermion parity and the lattice translation are in (4.13), (4.29), and (4.43), and summarized in Table 1.

Under this dictionary between the lattice and the continuum symmetries, the lattice algebras (3.72), (3.77), (3.80) agree with the those in the continuum RR, NSNS, and NSR theories in (2.19), (2.12), and (2.28). In particular, the projective phases of the lattice algebra (3.72) match with those in the continuum in (2.19), which are consequences of the following anomalies (see discussions in Section 2.2):

- the mod 8 anomaly of $(-1)^F, (-1)^{F_L}, \mathcal{P}$ classified by $\text{Hom}(\text{Tors } \Omega_3^{\text{DPin}}(pt), U(1)) = \mathbb{Z}_8$,

- the mod 8 anomaly of $(-1)^F, (-1)^{F_L}$ classified by $\text{Hom}(\text{Tors } \Omega_3^{\text{Spin}}(B\mathbb{Z}_2), U(1)) = \mathbb{Z}_8$,

- the mod 2 anomaly of $(-1)^F, \mathcal{P}$ classified by $\text{Hom}(\text{Tors } \Omega_3^{\text{Pin}^+}(pt), U(1)) = \mathbb{Z}_2$.

Even though the projective phases of the lattice system match with the continuum projective phases, the lattice model does not capture all the mod 8 anomalies of the continuum theory. All its anomalies are mod 2 anomalies. (As emphasized around (3.80) even this phase reflects only a

mod 2 anomaly.[56]) However, as in [32], if we restrict ourselves to the low-lying states of a particular lattice Hamiltonian (or small deformations of it), the lattice model can capture more subtle behavior, which is the precursor of anomalies in emanant symmetries of the continuum theory. Indeed, the discussion around (4.46) shows that for large but finite $L$, the low-lying spectrum of the lattice theory captures the mod 8 anomaly of the continuum theory.

In Section 5, we performed the Jordan-Wigner transformation of the Majorana chain to rewrite the fermion fields in terms of bosonic fields. The Jordan-Wigner transformation pairs up two Majorana sites into a single bosonic site. We carefully summed over the spin structures on the lattice to obtain local Hamiltonians in terms of the bosonic variables:

- For even $L$, we found either the transverse-field Ising Hamiltonian $H_{\text{Ising}}$ in (5.20) or its $\mathbb{Z}_2$-twisted version $H_{\text{tw}}$ in (5.31).

- For odd $L$, we found the Ising Hamilton $H_{\text{D}}$ twisted by the duality defect (5.43).

We have thus found all three topological defects (the trivial, $\mathbb{Z}_2$, and duality defects) of the critical transverse-field Ising model from the Majorana chain via bosonization.

As an interesting byproduct, we found that the Majorana translation operator leads to a non-invertible translation symmetry D of the transverse-field Ising model, and we gave an explicit expression for this operator in terms of the Ising spins in (6.7). It squares to the translation operator of the Ising spins by one site $T_{\text{Ising}}$, times a projection operator (6.11)

$$\mathsf{D}^2 = \frac{1}{2}(1 + \eta)\, T_{\text{Ising}}\,. \tag{7.3}$$

At the critical point, $m = 0$, the operator D commutes with the Ising Hamiltonian. However, it does not have an inverse. Hence, it is a non-invertible lattice symmetry. In the continuum limit, it flows to the non-invertible global symmetry $\mathcal{D}$, associated with the Kramers-Wannier duality of the continuum Ising CFT. We learn that the continuum $\mathcal{D}$ is a non-invertible emanant symmetry. See Table 1. A more detailed analysis of this emanant non-invertible symmetry will appear in [47].

Throughout the paper we used the phrase "on-site" symmetry action and discussed its various meanings. Let us summarize this discussion. We distinguish between four different cases:

- The simplest case is when the degrees of freedom reside on the sites (or more generally in the unit cells) labeled by $j$, the Hilbert space factorizes as in (1.9)

$$\mathcal{H} = \bigotimes_{j=1}^{N} \mathcal{H}_j\,, \tag{7.4}$$

the translation operator $T$ maps (1.10)

$$T\,:\quad \mathcal{H}_j \to \mathcal{H}_{j+1} \quad,\qquad j \sim j + N\,, \tag{7.5}$$

---

[56]This is similar to an anomaly in a Euclidean 4-dimensional lattice fermion model, as discussed in [46]. There, only a mod 2 reduction of the full $\Omega_5^{\text{Spin}^{\mathbb{Z}_4}}(pt) = \mathbb{Z}_{16}$ continuum 't Hooft anomaly is visible on the lattice.

| | number of sites/ boundary conditions | lattice translations | emanant symmetries |
|---|---|---|---|
| Majorana chain | even $L$ periodic – RR | $T_{\mathrm{RR}}^L = 1$ | $T_{\mathrm{RR}} = (-1)^{F_{\mathrm{L}}} e^{\frac{2\pi i P}{L}}$ |
| | even $L$ with defect antiperiodic – NSNS | $T_{\mathrm{NSNS}}^L = (-1)^F$ | $T_{\mathrm{NSNS}} = (-1)^{F_{\mathrm{L}}} e^{\frac{2\pi i P}{L}}$ |
| | odd $L$ mixed – NSR | $T_{\mathrm{odd}}^L = e^{-\frac{2\pi i}{16}}$ | $T_{\mathrm{odd}} = (-1)^{F_{\mathrm{L}}} e^{\frac{2\pi i P}{L}}$ |
| Ising model | general $N$ periodic | $\mathsf{D}^2 = \frac{1+\eta}{2} T_{\mathrm{Ising}}$  $T_{\mathrm{Ising}}^N = 1$ | $\mathsf{D} = \frac{1}{\sqrt{2}} \mathcal{D} e^{\frac{2\pi i P}{2N}}$ |

Table 1: The lattice translation operators of the Majorana chains and the Ising model, and their emanant symmetries in the continuum Majorana and Ising CFTs. We restrict to the specific fermionic Hamiltonians in Section 4 and the transverse-field Ising Hamiltonian (5.20). Here NS and R stand for the anti-periodic and periodic boundary conditions for the fermions, respectively. In the third column, we show the algebras of the lattice translation operators and in the fourth column, we show the relations between the lattice operators and the emanant chiral fermion parity $(-1)^{F_{\mathrm{L}}}$ of the Majorana CFT, and the non-invertible symmetry $\mathcal{D}$ of the Ising CFT. Here $(-1)^F$ is the (non-chiral) fermion parity of the Majorana chain and $\eta$ is the $\mathbb{Z}_2$ symmetry of the Ising model. We emphasize that restricting to the low-lying modes, the relations between the lattice and the continuum operators in the fourth column is exact and does not suffer from finite-size corrections.

and the internal symmetry operators are given by products of local factors (1.11)

$$\mathbf{S} = \mathsf{s}_1 \mathsf{s}_2 \cdots \mathsf{s}_N \,, \tag{7.6}$$

with the local factor $\mathsf{s}_j$ acting linearly on $\mathcal{H}_j$ . A typical example is the Ising chain (5.20). This example is referred to as on-site symmetry action.

- A more subtle situation occurs when the Hilbert space factorizes as in (7.4), the translation operator still maps as in (7.5), the internal symmetry factorizes in terms of local factors as in (7.6), but the local factors $\mathsf{s}_j$ act projectively on the local Hilbert spaces $\mathcal{H}_j$. In order to avoid this projective action, we can enlarge the unit cell to include several factors such that the local factors act linearly on them. For example, in the Heisenberg chain, we can group pairs of local Hilbert spaces, $\mathcal{H}'_k = \mathcal{H}_{2k-1} \otimes \mathcal{H}_{2k}$ such that $\mathsf{s}_{2k-1}\mathsf{s}_{2k}$ acts linearly on them. However, in that case $T$ does not act simply on the local factors $\mathcal{H}'_k$, and only $T^2$ acts simply

$\mathcal{H}'_k \to \mathcal{H}'_{k+1}$. This behavior is characteristic of the LSM anomaly [3] and was used in the various papers analyzing it. We will show similar anomalous behavior in a fermionic theory in [48]. Some authors refer to this projective action as being on-site, while other authors refer to it as not-on-site. (See a detailed discussion of this terminology in [32].)

- An even more subtle situation occurs when the Hilbert space factorizes as in (7.4), the translation operator still maps as in (7.5), but the internal symmetry does not have a factorization as in (7.6) in terms of $\mathsf{s}_j$ acting only on $\mathcal{H}_j$. This anomalous situation arises in various examples including the Levin-Gu model [120] and the modified Villain model [32]. This behavior is always being referred to as not-on-site. (See also [121] for a related discussion about on-site symmetry action in Euclidean lattice models.)

- In the critical Majorana chain, discussed in this paper, we encountered a more subtle situation. The unit cell includes a single site with a single Majorana fermion $\chi_\ell$. There is no local Hilbert space labeled by $\ell$. We can have local Hilbert spaces $\mathcal{H}_j$ as in (7.4), but they are associated with two unit cells. Here, the internal $\mathbb{Z}_2$ symmetry generated by G factorizes as in (7.6) with local factors acting linearly on $\mathcal{H}_j$ (see the discussion after equation (3.13).) The situation with $T$ is more interesting. $T$ acts simply on the fundamental fermions $\chi_\ell \to \chi_{\ell+1}$, but since $\mathcal{H}_j$ is associated with two fermions, $\chi_{2j-1}$ and $\chi_{2j}$, $T$ does not act simply on $\mathcal{H}_j$. However, unlike the second case above, where $T^2$ acts simply on $\mathcal{H}_j$, this is not the case here. For this reason, the anomaly in the Majorana chain differs from that of the LSM anomaly.

Finally, this work can be extended in many directions. In particular, in [47, 48], we will present a more detailed discussion of this system, its symmetries, defects, anomalies, and their consequences. We will also repeat this analysis for other related systems. It would be nice to apply these techniques to all the minimal models and to relate the picture of the defects in [122] to ours.[57] The special case of the 3-state Potts model was recently studied in [123].

# Acknowledgements

We thank T. Banks for collaboration in the initial stage of a related project [48]. We are grateful to M. Cheng, Y. Choi, D. Delmastro, D. Gaiotto, A. Jaffe, J. McGreevy, S. Pufu, L. Rastelli, B. Rayhaun, S. Ryu, S. Seifnashri, N. Tantivasadakarn, J. Wang, X.-G. Wen, Y. Zheng, W. Zhang and S. Zhong for useful discussions. We thank P. Fendley, S. Seifnashri, Y. Tachikawa, and Y. Zheng for comments on the manuscript. The work of NS was supported in part by DOE grant DE−SC0009988 and by the Simons Collaboration on Ultra-Quantum Matter, which is a grant from the Simons Foundation (651440, NS). The work of SHS was supported in part by NSF grant PHY-2210182. SHS thanks Harvard University for its hospitality during the course of this work. The authors of this paper were ordered alphabetically. Opinions and conclusions expressed here are those of the authors and do not necessarily reflect the views of funding agencies.

---

[57]We thank the referee for making this interesting suggestion.

# A  Review of the continuum Majorana and Ising CFTs

## A.1  Partition functions

In this section, we will review the standard computations of the partition functions of a free Majorana CFT in the continuum. We will find perfect agreement between the relations of these continuum partition functions and those found on the lattice in Section 3.4.

NSNS

Define

$$\mathcal{Z}_{\text{NSNS}}(\tau, \bar{\tau}; m, m_{\text{L}}) = \text{Tr}_{\mathcal{H}_{\text{NSNS}}}[q^{h_{\text{L}} - \frac{1}{48}} \bar{q}^{h_{\text{R}} - \frac{1}{48}} \left((-1)^F\right)^m \left((-1)^{F_{\text{L}}}\right)^{m_{\text{L}}}], \tag{A.1}$$

where $q = e^{2\pi i \tau}$, $\bar{q} = e^{-2\pi i \bar{\tau}}$, and $m, m_{\text{L}} = 0, 1$. They are

$$\mathcal{Z}_{\text{NSNS}}(\tau, \bar{\tau}; m = 0, m_{\text{L}} = 0) = q^{-\frac{1}{48}} \bar{q}^{-\frac{1}{48}} \prod_{k=1}^{\infty} |1 + q^{k - \frac{1}{2}}|^2 = \left| \frac{\theta_3(\tau)}{\eta(\tau)} \right|,$$

$$\mathcal{Z}_{\text{NSNS}}(\tau, \bar{\tau}; m = 1, m_{\text{L}} = 0) = q^{-\frac{1}{48}} \bar{q}^{-\frac{1}{48}} \prod_{k=1}^{\infty} |1 - q^{k - \frac{1}{2}}|^2 = \left| \frac{\theta_4(\tau)}{\eta(\tau)} \right|,$$

$$\mathcal{Z}_{\text{NSNS}}(\tau, \bar{\tau}; m = 0, m_{\text{L}} = 1) = q^{-\frac{1}{48}} \bar{q}^{-\frac{1}{48}} \prod_{k=1}^{\infty} (1 - q^{k - \frac{1}{2}})(1 + \bar{q}^{k - \frac{1}{2}}) = \sqrt{\frac{\theta_4(\tau)}{\eta(\tau)}} \sqrt{\frac{\theta_3(\bar{\tau})}{\eta(\bar{\tau})}}, \tag{A.2}$$

$$\mathcal{Z}_{\text{NSNS}}(\tau, \bar{\tau}; m = 1, m_{\text{L}} = 1) = q^{-\frac{1}{48}} \bar{q}^{-\frac{1}{48}} \prod_{k=1}^{\infty} (1 + q^{k - \frac{1}{2}})(1 - \bar{q}^{k - \frac{1}{2}}) = \sqrt{\frac{\theta_3(\tau)}{\eta(\tau)}} \sqrt{\frac{\theta_4(\bar{\tau})}{\eta(\bar{\tau})}}.$$

The fact that $\mathcal{Z}_{\text{NSNS}}(\tau, \bar{\tau}; m, m_{\text{L}} = 0)$ is invariant under $\text{Re}(\tau) \to -\text{Re}(\tau)$ agrees with the first line of (3.88) with even $n$ on the lattice. Similarly, $\mathcal{Z}_{\text{NSNS}}(\tau; \bar{\tau}; m, m_{\text{L}} = 1)$ with $m = 0, 1$ are related by $\text{Re}(\tau) \to -\text{Re}(\tau)$ corresponds to the first line of (3.88) with odd $n$.

Next, we insert the parity operator and define

$$\mathcal{Z}_{\text{NSNS}}^{\mathcal{P}}(\mathsf{t}; m, m_{\text{L}}) = \text{Tr}_{\mathcal{H}_{\text{NSNS}}}[e^{-2\pi \mathsf{t}(\Delta - \frac{1}{24})} \mathcal{P} \left((-1)^F\right)^m \left((-1)^{F_{\text{L}}}\right)^{m_{\text{L}}}], \tag{A.3}$$

where $\Delta = h_{\text{L}} + h_{\text{R}}$. In the presence of the parity operator, the partition function is independent of $m$, and we have

$$\mathcal{Z}_{\text{NSNS}}^{\mathcal{P}}(\mathsf{t}; m, m_{\text{L}} = 0) = \mathsf{q}^{-\frac{1}{24}} \prod_{k=1}^{\infty} (1 + \mathsf{q}^{2k-1}) = \sqrt{\frac{\theta_3(2i\mathsf{t})}{\eta(2i\mathsf{t})}},$$

$$\mathcal{Z}_{\text{NSNS}}^{\mathcal{P}}(\mathsf{t}; m, m_{\text{L}} = 1) = \mathsf{q}^{-\frac{1}{24}} \prod_{k=1}^{\infty} (1 - \mathsf{q}^{2k-1}) = \sqrt{\frac{\theta_4(2i\mathsf{t})}{\eta(2i\mathsf{t})}}, \tag{A.4}$$

where $\mathsf{q} = e^{-2\pi\mathsf{t}}$ with real $\mathsf{t}$. Note that these partition functions with a parity operator insertion depend only on one real modulus $\mathsf{t}$. This is consistent with the fact that on the lattice, $n$ can be set to be $\pm 1$ by the second line of the lattice (3.88), which further implies that these partition functions are independent of $m$.

## RR

In the RR Hilbert space, we define

$$\mathcal{Z}_{\mathrm{RR}}(\tau, \bar{\tau}; m, m_{\mathrm{L}}) = \mathrm{Tr}_{\mathcal{H}_{\mathrm{RR}}}[q^{h_{\mathrm{L}}-\frac{1}{48}} \bar{q}^{h_{\mathrm{R}}-\frac{1}{48}} \left((-1)^F\right)^m \left((-1)^{F_{\mathrm{L}}}\right)^{m_{\mathrm{L}}}]. \tag{A.5}$$

They are

$$\mathcal{Z}_{\mathrm{RR}}(\tau, \bar{\tau}; m = 0, m_{\mathrm{L}} = 0) = 2q^{\frac{1}{24}} \bar{q}^{\frac{1}{24}} \prod_{k=1}^{\infty} |1 + q^k|^2 = \left|\frac{\theta_2(\tau)}{\eta(\tau)}\right|, \tag{A.6}$$

$$\mathcal{Z}_{\mathrm{RR}}(\tau, \bar{\tau}; m, m_{\mathrm{L}}) = 0, \qquad \text{otherwise}.$$

This is consistent with the lattice relation in (3.83). Furthermore, the fact that $\mathcal{Z}_{\mathrm{RR}}(\tau, \bar{\tau}; m = 0, m_{\mathrm{L}} = 0)$ is invariant under $\mathrm{Re}(\tau) \to -\mathrm{Re}(\tau)$ is consistent with the lattice relation (3.84).

Next, we insert the parity operator and define

$$\mathcal{Z}_{\mathrm{RR}}^{\mathcal{P}}(\mathsf{t}; m, m_{\mathrm{L}}) = \mathrm{Tr}_{\mathcal{H}_{\mathrm{RR}}}[e^{-2\pi\mathsf{t}(\Delta-\frac{1}{24})}\mathcal{P} \left((-1)^F\right)^m \left((-1)^{F_{\mathrm{L}}}\right)^{m_{\mathrm{L}}}]. \tag{A.7}$$

Again, it depends only on one real modulus $\mathsf{t}$. This is consistent with the fact that on the lattice, $n$ can be set to be $\pm 1$ when there is a parity operator insertion $m_{\mathcal{P}} = 1$ in (3.86).

The two nonzero partition functions are

$$\mathcal{Z}_{\mathrm{RR}}^{\mathcal{P}}(\mathsf{t}; m = 0, m_{\mathrm{L}} = 1) = \mathrm{Tr}_{\mathcal{H}_{\mathrm{RR}}}[e^{-2\pi\mathsf{t}(\Delta-\frac{1}{24})}\mathcal{P} (-1)^{F_{\mathrm{L}}}] = \sqrt{2}\mathsf{q}^{\frac{1}{12}} \prod_{k=1}^{\infty}(1 + \mathsf{q}^{2k}) = \sqrt{\frac{\theta_2(2i\mathsf{t})}{\eta(2i\mathsf{t})}},$$

$$\mathcal{Z}_{\mathrm{RR}}^{\mathcal{P}}(\mathsf{t}; m = 1, m_{\mathrm{L}} = 1) = \mathrm{Tr}_{\mathcal{H}_{\mathrm{RR}}}[e^{-2\pi\mathsf{t}(\Delta-\frac{1}{24})}\mathcal{P} (-1)^{F} (-1)^{F_{\mathrm{L}}}] = \sqrt{2}i\mathsf{q}^{\frac{1}{12}} \prod_{k=1}^{\infty}(1 + \mathsf{q}^{2k}) = i\sqrt{\frac{\theta_2(2i\mathsf{t})}{\eta(2i\mathsf{t})}},$$
$$\tag{A.8}$$

while the other two partition functions vanish

$$\mathcal{Z}_{\mathrm{RR}}^{\mathcal{P}}(\mathsf{t}; m, m_{\mathrm{L}} = 0) = 0. \tag{A.9}$$

This is consistent with the lattice relation (3.85).

Using (2.19), we derive the following general relation between the partition functions:

$$\mathcal{Z}_{\mathrm{RR}}^{\mathcal{P}}(\mathsf{t}; m = 0, m_{\mathrm{L}} = 1) = -i\mathcal{Z}_{\mathrm{RR}}^{\mathcal{P}}(\mathsf{t}; m = 1, m_{\mathrm{L}} = 1). \tag{A.10}$$

This is consistent with (3.86) with $n = \pm 1$ and $m = 0$.

Define the NSR partition functions as

$$\mathcal{Z}_{\mathrm{NSR}}(\tau, \bar{\tau}; m_{\mathrm{L}}) = \mathrm{Tr}_{\mathcal{H}_{\mathrm{NSR}}}[q^{h_{\mathrm{L}} - \frac{1}{48}} \bar{q}^{h_{\mathrm{R}} - \frac{1}{48}} \left((-1)^{F_{\mathrm{L}}}\right)^{m_{\mathrm{L}}}]. \tag{A.11}$$

We have

$$
\begin{aligned}
\mathcal{Z}_{\mathrm{NSR}}(\tau, \bar{\tau}; m_{\mathrm{L}} = 0) &= q^{-\frac{1}{48}} \bar{q}^{\frac{1}{24}} \prod_{k=1}^{\infty} (1 + q^{k-\frac{1}{2}})(1 + \bar{q}^{k}) = \frac{1}{\sqrt{2}} \sqrt{\frac{\theta_3(\tau)}{\eta(\tau)}} \sqrt{\frac{\theta_2(\bar{\tau})}{\eta(\bar{\tau})}}, \\
\mathcal{Z}_{\mathrm{NSR}}(\tau, \bar{\tau}; m_{\mathrm{L}} = 1) &= q^{-\frac{1}{48}} \bar{q}^{\frac{1}{24}} \prod_{k=1}^{\infty} (1 - q^{k-\frac{1}{2}})(1 + \bar{q}^{k}) = \frac{1}{\sqrt{2}} \sqrt{\frac{\theta_4(\tau)}{\eta(\tau)}} \sqrt{\frac{\theta_2(\bar{\tau})}{\eta(\bar{\tau})}}.
\end{aligned}
\tag{A.12}
$$

Here we choose to quantize the fermion zero mode using a one-dimensional irreducible representation as in Section 2.3.

A tensor product of two copies of the Hilbert space of our NSR theory has a single ground state. However, the canonical quantization of that system as two Majorana fermions that anticommute with each other doubles the Hilbert space and leads to

$$\mathcal{Z}_{\mathrm{NSR}^2}(\tau, \bar{\tau}; m_{\mathrm{L}}) = 2\mathcal{Z}_{\mathrm{NSR}}(\tau, \bar{\tau}; m_{\mathrm{L}})^2. \tag{A.13}$$

Therefore, it is common to define $\mathcal{Z}_{\mathrm{NSR}}$ as the square root of $\mathcal{Z}_{\mathrm{NSR}^2}$, which differs from our definition by a factor of $\sqrt{2}$, but does not admit a Hilbert space interpretation of the NSR problem. See the related discussion in footnote 1.

The RNS partition functions are similarly obtained from the NSR partition function by exchanging L with R and $q$ with $\bar{q}$.

As in the discussion around (A.13), we face a question how to quantize the tensor product of the NSR and RNS theories. The tensor product of their Hilbert spaces differs from the tensor product of the NSNS and RR Hilbert spaces, as each state in the latter problem appears twice. Hence,

$$
\begin{aligned}
2\mathcal{Z}_{\mathrm{NSR}}(\tau, \bar{\tau}; m_{\mathrm{L}} = 0)&\mathcal{Z}_{\mathrm{RNS}}(\tau, \bar{\tau}; m_{\mathrm{R}} = 0) \\
&= \mathcal{Z}_{\mathrm{NSNS}}(\tau, \bar{\tau}; m = 0, m_{\mathrm{L}} = 0)\mathcal{Z}_{\mathrm{RR}}(\tau, \bar{\tau}; m = 0, m_{\mathrm{L}} = 0).
\end{aligned}
\tag{A.14}
$$

This might motivate us to absorb a factor of $\sqrt{2}$ in $\mathcal{Z}_{\mathrm{NSR}}$, but then the NSR problem does not have a Hilbert space interpretation. Again, compare with footnote 1.

## A.2 Modular transformation and the interval Hilbert spaces

We study the modular S transformations of the various partition functions. The following identities will be useful:

$$
\begin{aligned}
\theta_3(-1/\tau) &= \sqrt{-i\tau}\, \theta_3(\tau), \\
\theta_2(-1/\tau) &= \sqrt{-i\tau}\theta_4(\tau), \\
\theta_4(-1/\tau) &= \sqrt{-i\tau}\theta_2(\tau), \\
\eta(-1/\tau) &= \sqrt{-i\tau}\eta(\tau).
\end{aligned}
\tag{A.15}
$$

The modular S transformation leaves the NSNS partition function $\mathcal{Z}_{\mathrm{NSNS}}(\tau, \bar{\tau}; m = 0, m_{\mathrm{L}} = 0)$ without operator insertion invariant. On the other hand, the modular S transformation of $\mathcal{Z}_{\mathrm{NSNS}}(\tau, \bar{\tau}; m = 1, m_{\mathrm{L}} = 0)$ with a $(-1)^F$ operator insertion turns the latter into a defect that modifies the NSNS Hilbert space into the RR one:

$$\mathcal{Z}_{\mathrm{NSNS}}(\tau, \bar{\tau}; m = 1, m_{\mathrm{L}} = 0) = \left| \frac{\theta_2(-\frac{1}{\tau})}{\eta(-\frac{1}{\tau})} \right| = \mathcal{Z}_{\mathrm{RR}}(-\frac{1}{\tau}, -\frac{1}{\bar{\tau}}; m = 0, m_{\mathrm{L}} = 0) \qquad (A.16)$$

Next, the modular S transformation of the NSNS partition function $\mathcal{Z}_{\mathrm{NSNS}}(\tau, \bar{\tau}; m = 1, m_{\mathrm{L}} = 1)$ with a $(-1)^{F_{\mathrm{R}}}$ operator insertion turns the latter into a defect that modifies the Hilbert space to the NSR theory:

$$\mathcal{Z}_{\mathrm{NSNS}}(\tau, \bar{\tau}; m = 1, m_{\mathrm{L}} = 1) = \sqrt{\frac{\theta_3(-\frac{1}{\tau})}{\eta(-\frac{1}{\tau})}} \sqrt{\frac{\theta_2(-\frac{1}{\bar{\tau}})}{\eta(-\frac{1}{\bar{\tau}})}} = \sqrt{2} \mathcal{Z}_{\mathrm{NSR}}(-\frac{1}{\tau}, -\frac{1}{\bar{\tau}}; m_{\mathrm{L}} = 0) \quad (A.17)$$

As in footnote 1 and in the discussion in around (A.13) and (A.14), the extra factor of $\sqrt{2}$ shows the confusing relation between the canonical quantization of the system and its path integral description. The trace over the Hilbert space $\mathcal{H}_{\mathrm{NSR}}$ (A.12) lacks the factor of $\sqrt{2}$, while the functional integral description of the system leads to the partition function (A.17). This issue arises whenever we have an odd number of quantum mechanical real fermions.

Alternatively, as discussed in footnote 1 and in Section 2.3, one can quantize the zero mode in the NSR Hilbert space using a two-dimensional irreducible representation. The resulting NSR partition function is $\mathcal{Z}'_{\mathrm{NSR}} = 2\mathcal{Z}_{\mathrm{NSR}}$. It is still incompatible with the modular transformation (A.17). We conclude that neither of the two canonical quantizations of the NSR theory is compatible with the functional path integral.

Similarly, the modular S transformation of $\mathcal{Z}_{\mathrm{NSNS}}(\tau, \bar{\tau}; m = 0, m_{\mathrm{L}} = 1)$ gives the RNS partition function with the same relative factor of $\sqrt{2}$.

The modular transformation of $\mathcal{Z}^{\mathcal{P}}_{\mathrm{NSNS}}(\mathsf{t}; m, m_{\mathrm{L}} = 0)$ is

$$\mathcal{Z}^{\mathcal{P}}_{\mathrm{NSNS}}(\mathsf{t}; m, m_{\mathrm{L}} = 0) = e^{\frac{2\pi\ell}{48}} \prod_{k=1}^{\infty} (1 + e^{-2\pi\ell(k-\frac{1}{2})}) = \mathrm{Tr}_{\mathcal{H}^{\mathrm{o}}_{\mathrm{NS}}}[e^{-2\pi\ell(\Delta - \frac{1}{48})}] \qquad (A.18)$$

where

$$\ell = \frac{1}{2\mathsf{t}}. \qquad (A.19)$$

This is the partition function of a free Majorana fermion on an interval with the NS boundary conditions

$$\mathrm{NS}: \quad \chi_{\mathrm{L}}\Big|_{x=0} = \chi_{\mathrm{R}}\Big|_{x=0}, \quad \chi_{\mathrm{L}}\Big|_{x=\pi} = -\chi_{\mathrm{R}}\Big|_{x=\pi}. \qquad (A.20)$$

$\mathcal{H}^{\mathrm{o}}_{\mathrm{NS}}$ stands for the interval Hilbert space with the above boundary conditions. The superscript "o" stands for the open string channel. Before the modular transformation, the parity twist is in the time direction. After the transformation, it becomes a parity twist in space, which can be interpreted as an interval. The factor of 1/2 in (A.19) is due to the parity twist.

The modular transform of $\mathcal{Z}^{\mathcal{P}}_{\mathrm{NSNS}}(\mathsf{t}; m, m_{\mathrm{L}} = 1)$ is

$$\mathcal{Z}^{\mathcal{P}}_{\mathrm{NSNS}}(\mathsf{t}; m, m_{\mathrm{L}} = 1) = \sqrt{\frac{\theta_2(1/2\mathsf{t})}{\eta(1/2\mathsf{t})}} = \sqrt{2}\, e^{\frac{-2\pi\ell}{24}} \prod_{k=1}^{\infty} \left(1 + e^{-2\pi\ell k}\right), \qquad \text{(A.21)}$$

One might want to interpret it as the trace $\mathrm{Tr}_{\mathcal{H}^{\mathrm{o}}_{\mathrm{R}}}[e^{-2\pi\ell(\Delta - \frac{1}{48})}]$ over the Hilbert space of a system on an interval with R boundary conditions

$$\mathrm{R}: \quad \chi_{\mathrm{L}}\Big|_{x=0} = \chi_{\mathrm{R}}\Big|_{x=0}, \quad \chi_{\mathrm{L}}\Big|_{x=\pi} = \chi_{\mathrm{R}}\Big|_{x=\pi}. \qquad \text{(A.22)}$$

But the factor of $\sqrt{2}$ shows that this cannot be right. Again we see that the path integral description, which leads to (A.21), is incompatible with the canonical quantization over $\mathcal{H}^{\mathrm{o}}_{\mathrm{R}}$. The source of this factor of $\sqrt{2}$, both in $\mathcal{H}^{\mathrm{o}}_{\mathrm{R}}$ and in $\mathcal{H}_{\mathrm{NSR}}$ or $\mathcal{H}_{\mathrm{RNS}}$, is due to an odd number of fermion zero modes. See [53, 54] for more discussions of a free 1+1d Majorana CFT on an interval, and the relation to the factor of $\sqrt{2}$ in the quantum mechanical model of odd number of real fermions mentioned in footnote 1.

The modular transformation of $\mathcal{Z}^{\mathcal{P}}_{\mathrm{RR}}(\mathsf{t}; m, m_{\mathrm{L}} = 1)$ is

$$\mathcal{Z}^{\mathcal{P}}_{\mathrm{RR}}(\mathsf{t}; m = 0, m_{\mathrm{L}} = 1) = e^{\frac{2\pi\ell}{48}} \prod_{k=1}^{\infty} \left(1 - e^{-2\pi\ell(k-\frac{1}{2})}\right) = \mathrm{Tr}_{\mathcal{H}^{\mathrm{o}}_{\mathrm{NS}}}\left[(-1)^{F_{\mathrm{o}}} e^{-2\pi\ell(\Delta - \frac{1}{48})}\right], \qquad \text{(A.23)}$$

which is interpreted in the dual channel as the partition function of a Majorana fermion on an interval of length $\ell = 1/2\mathsf{t}$ with NS boundary condition (A.20). The symmetry operator $(-1)^{F_{\mathrm{o}}}$ is the fermion parity operator on the interval.

## A.3    Bosonization and fermionization

Here we review the standard bosonization and fermionization in 1+1d CFT. We assume $c_{\mathrm{L}} - c_{\mathrm{R}} = 0$ mod 8 throughout, which is a necessary condition for every bosonic CFT. Our review follows the recent discussions in [124, 36–41, 125–129], which builds on the classic paper [130].

Let $\mathcal{F}$ be a 1+1d fermionic CFT with a fermion parity symmetry $(-1)^F$. We consider two Hilbert spaces associated with $\mathcal{F}$. The first one is the NS Hilbert space $\mathcal{H}_{\mathrm{NS}}$, which is in one-to-one correspondence with the local operators of $\mathcal{F}$ by the operator-state correspondence. The second one is obtained from $\mathcal{H}_{\mathrm{NS}}$ by a $(-1)^F$ twist, which we call the R Hilbert space $\mathcal{H}_{\mathrm{R}}$.[58] We can further grade each Hilbert space by $(-1)^F$, i.e., $\mathcal{H}_{\mathrm{NS}} = \mathcal{H}^+_{\mathrm{NS}} \oplus \mathcal{H}^-_{\mathrm{NS}}$ and $\mathcal{H}_{\mathrm{R}} = \mathcal{H}^+_{\mathrm{R}} \oplus \mathcal{H}^-_{\mathrm{R}}$.

In 1+1d, there is an invertible fermionic topological field theory, denoted as $(-1)^{\mathrm{Arf}}$. On a closed spin spacetime manifold, $(-1)^{\mathrm{Arf}} = +1$ if the spin structure is even and $(-1)^{\mathrm{Arf}} = -1$ if the spin structure is odd. It can be viewed as a fermionic local counterterm.

---

[58] In general, there are other global symmetries of $\mathcal{F}$ that one can use to define a twisted Hilbert space. For instance, for the massless Majorana CFT, we also have a chiral fermion parity $(-1)^{F_{\mathrm{L}}}$, which together with $(-1)^F$, lead to four Hilbert spaces, $\mathcal{H}_{\mathrm{NSNS}}, \mathcal{H}_{\mathrm{RR}}, \mathcal{H}_{\mathrm{NSR}}, \mathcal{H}_{\mathrm{RNS}}$. In this appendix we do not assume any other symmetry than $(-1)^F$. In this setting there are only two relevant Hilbert spaces of interest to us. When we apply this general discussion to the massless Majorana CFT, we should replace $\mathcal{H}_{\mathrm{NS}}, \mathcal{H}_{\mathrm{R}}$ by $\mathcal{H}_{\mathrm{NSNS}}, \mathcal{H}_{\mathrm{RR}}$, respectively.

Given any fermionic CFT $\mathcal{F}$, we can stack it with $(-1)^{\mathrm{Arf}}$. The stacking flips the $(-1)^F$ eigenvalue in $\mathcal{H}_\mathrm{R}$ and thus exchanges $\mathcal{H}_\mathrm{R}^+$ with $\mathcal{H}_\mathrm{R}^-$.

Starting with a fermionic CFT $\mathcal{F}$, we can sum over the spin structures to obtain a bosonic CFT $\mathcal{B}$. This is also known as bosonization, which is equivalent to gauging $(-1)^F$. Since $(-1)^F$ is gauged, it is no longer a global symmetry in $\mathcal{B}$. Rather, we obtain a quantum $\mathbb{Z}_2$ symmetry $\eta$ (which is free of 't Hooft anomaly) in the bosonic CFT $\mathcal{B}$, which is implemented by the Wilson line of the gauged $(-1)^F$ symmetry. Let $\mathcal{H}_\mathrm{untw}$ be the (untwised) Hilbert space of $\mathcal{B}$, which is in one-to-one correspondence with the local operators of $\mathcal{B}$. We can also twist the Hilbert space by $\eta$ to obtain a twisted Hilbert space $\mathcal{H}_\mathrm{tw}$. We can grade $\mathcal{H}_\mathrm{untw}$ and $\mathcal{H}_\mathrm{tw}$ by the $\mathbb{Z}_2$ symmetry $\eta$, i.e., $\mathcal{H}_\mathrm{untw} = \mathcal{H}_\mathrm{untw}^+ \oplus \mathcal{H}_\mathrm{untw}^-$ and $\mathcal{H}_\mathrm{tw} = \mathcal{H}_\mathrm{tw}^+ \oplus \mathcal{H}_\mathrm{tw}^-$. The $\mathbb{Z}_2$ symmetry $\eta$ is free of 't Hooft anomaly.

In both $\mathcal{F}$ and $\mathcal{B}$, there are four sectors of Hilbert spaces, and they are mapped to each under the bosonization as follows:

$$\begin{aligned} \mathcal{H}_\mathrm{untw}^+ = \mathcal{H}_\mathrm{NS}^+ \,, && \mathcal{H}_\mathrm{untw}^- = \mathcal{H}_\mathrm{R}^- \,, \\ \mathcal{H}_\mathrm{tw}^+ = \mathcal{H}_\mathrm{R}^+ \,, && \mathcal{H}_\mathrm{tw}^- = \mathcal{H}_\mathrm{NS}^- \,. \end{aligned} \tag{A.24}$$

Conversely, one can start with $\mathcal{B}$ and couple it to a fermionic topological field theory to retrieve $\mathcal{F}$.

Starting from $\mathcal{F}$, we can first stack it with $(-1)^{\mathrm{Arf}}$, and then gauge the diagonal $(-1)^F$. This gives another bosonic CFT $\mathcal{B}'$, whose four sectors are related to the fermionic ones as in (A.24) but with $\mathcal{H}_\mathrm{R}^+$ and $\mathcal{H}_\mathrm{R}^-$ exchanged. The new bosonic CFT $\mathcal{B}'$ is related to $\mathcal{B}$ by gauging the $\mathbb{Z}_2$ symmetry $\eta$, i.e., $\mathcal{B}' = \mathcal{B}/\mathbb{Z}_2$.[59]

Let us illustrate this bosonization procedure in the case of the massless Majorana CFT $\mathcal{F}$. The states in $\mathcal{H}_\mathrm{NS}$ correspond to the following Virasoro primary operators

$$\mathcal{H}_\mathrm{NS}^+ : \ 1, \chi_\mathrm{L}\chi_\mathrm{R} \,, , \quad \mathcal{H}_\mathrm{NS}^- : \ \chi_\mathrm{L}, \chi_\mathrm{R} \,, \tag{A.25}$$

where $\chi_\mathrm{L}$ and $\chi_\mathrm{R}$ have conformal weights $(h_\mathrm{L}, h_\mathrm{R}) = (\frac{1}{2}, 0)$ and $(0, \frac{1}{2})$, respectively. The states in $\mathcal{H}_\mathrm{R}$ correspond to the following Virsoro primary operators

$$\mathcal{H}_\mathrm{R}^+ : \ \mu \,, \quad \mathcal{H}_\mathrm{R}^- : \ \sigma \,, \tag{A.26}$$

where $\mu, \sigma$ are the spin fields with conformal weights $(h_\mathrm{L}, h_\mathrm{R}) = (\frac{1}{16}, \frac{1}{16})$.

The bosonization of the Majorana CFT gives the bosonic Ising CFT $\mathcal{B}$, a.k.a., the (3,4) Virasoro minimal model. The states in the untwisted Hilbert space correspond to the following local Virasoro primary operators

$$\mathcal{H}_\mathrm{untw}^+ : \ 1, \varepsilon \,, \quad \mathcal{H}_\mathrm{untw}^- : \ \sigma \,, \tag{A.27}$$

where $\sigma$ is the order operator with $(h_\mathrm{L}, h_\mathrm{R}) = (\frac{1}{16}, \frac{1}{16})$, and $\varepsilon$ is the thermal operator with $(h_\mathrm{L}, h_\mathrm{R}) = (\frac{1}{2}, \frac{1}{2})$. The states in the $\mathbb{Z}_2$ twisted Hilbert space $\mathcal{H}_\mathrm{tw}$ correspond to non-local operators attached to a $\mathbb{Z}_2$ topological line. The Virasoro primaries in this twisted Hilbert space are

$$\mathcal{H}_\mathrm{tw}^+ : \ \mu \,, \quad \mathcal{H}_\mathrm{tw}^- : \ \chi_\mathrm{L}, \chi_\mathrm{R} \,, \tag{A.28}$$

---

[59]In the string theory literature, $\mathcal{B}'$ is sometimes called the $\mathbb{Z}_2$ orbifold of $\mathcal{B}$. It is also common to refer to the two theories $\mathcal{B}$ and $\mathcal{B}'$ as type 0A and type 0B.

where $\mu$ is the disorder operator with $(h_{\mathrm{L}}, h_{\mathrm{R}}) = (\frac{1}{16}, \frac{1}{16})$. Indeed, we see that these Hilbert spaces are related to each other as (A.24), where we have identified $\varepsilon$ in $\mathcal{B}$ as $\chi_{\mathrm{L}}\chi_{\mathrm{R}}$ in $\mathcal{F}$ under bosonization.

Importantly, while the fermion operators $\chi_{\mathrm{L}}$ and $\chi_{\mathrm{R}}$ are local operators in $\mathcal{F}$, they are non-local operators attached to a $\mathbb{Z}_2$ line in $\mathcal{B}$. Conversely, while the order operator $\sigma$ is a local operator in $\mathcal{B}$, it becomes a non-local operator attached to a $(-1)^F$ line in $\mathcal{F}$. This is the continuum counterpart of the discussion in Section 5.1 for the Jordan-Wigner transformation.

If we stack $(-1)^{\mathrm{Arf}}$ in the Majorana CFT before bosonization, then we would obtain $\mathcal{B}'$ which has $\mu$ in $\mathcal{H}_{\mathrm{untw}}^-$ and $\sigma$ in $\mathcal{H}_{\mathrm{tw}}^+$. In this case, $\mathcal{B}' = \mathcal{B}/\mathbb{Z}_2$ happens to be isomorphic to $\mathcal{B}$ because the critical Ising CFT is invariant under gauging the $\mathbb{Z}_2$ symmetry, which exchanges the order operator $\sigma$ with the disorder operator $\mu$. This is related to the fact that the Majorana CFT $\mathcal{F}$ is invariant under stacking it with $(-1)^{\mathrm{Arf}}$.

We summarize the bosonization/fermionization and $\mathbb{Z}_2$ orbifold below for the case of the Majorana and the Ising CFTs. In the entries we record the Virasoro primary operators in each sector.

| $\mathcal{F}$ | $(-1)^F$-even | $(-1)^F$-odd |
|---|---|---|
| NS | $1, \chi_{\mathrm{L}}\chi_{\mathrm{R}}$ | $\chi_{\mathrm{L}}, \chi_{\mathrm{R}}$ |
| R | $\mu$ | $\sigma$ |

$\xleftrightarrow{\;\otimes\,(-1)^{\mathrm{Arf}}\;}$

| $\mathcal{F} \otimes (-1)^{\mathrm{Arf}}$ | $(-1)^F$-even | $(-1)^F$-odd |
|---|---|---|
| NS | $1, \chi_{\mathrm{L}}\chi_{\mathrm{R}}$ | $\chi_{\mathrm{L}}, \chi_{\mathrm{R}}$ |
| R | $\sigma$ | $\mu$ |

fermionization $\updownarrow$ bosonization $\qquad\qquad$ fermionization $\updownarrow$ bosonization

| $\mathcal{B}$ | $\mathbb{Z}_2$-even | $\mathbb{Z}_2$-odd |
|---|---|---|
| untwisted | $1, \varepsilon$ | $\sigma$ |
| twisted | $\mu$ | $\chi_{\mathrm{L}}, \chi_{\mathrm{R}}$ |

$\xleftrightarrow{\;\mathbb{Z}_2 \text{ orbifold}\;}$

| $\mathcal{B}'$ | $\mathbb{Z}_2$-even | $\mathbb{Z}_2$-odd |
|---|---|---|
| untwisted | $1, \varepsilon$ | $\mu$ |
| twisted | $\sigma$ | $\chi_{\mathrm{L}}, \chi_{\mathrm{R}}$ |

$$\text{(A.29)}$$

We can couple the bosonic CFT $\mathcal{B}$ to a 2+1d $\mathbb{Z}_2$ gauge theory by gauging the $\mathbb{Z}_2$ symmetry. In this new system, $\mathcal{B}$ becomes a gapless boundary of a 2+1d theory. The four sectors now become the Hilbert spaces of the 2+1d $\mathbb{Z}_2$ gauge theory on a disk with a topological line insertion in the middle, and the above gapless boundary condition at the boundary of the disk. (These topological lines arise from the anyon lines of the microscopic toric code.) More explicitly, $\mathcal{H}_{\mathrm{untw}}^+, \mathcal{H}_{\mathrm{untw}}^-, \mathcal{H}_{\mathrm{tw}}^+, \mathcal{H}_{\mathrm{tw}}^-$ correspond to the trivial line $1$, the electric boson line $e$, the magnetic boson line $m$, and the fermion line $\psi$, respectively [98, 86].

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
