# Peer review of "Majorana chain and Ising model -- (non-invertible) translations, anomalies, and emanant symmetries"

_SciPost Physics_

## Round 2 · Referee Report · Anonymous (Referee 1) · 2023-11-30

Strengths

see report

Weaknesses

see report

Report

This is an excellent paper that should be published.
It describes constraints from symmetry and anomalies on 1d local models made from majorana modes. Although there are exactly solvable (free fermion) models that satisfy these demands, most of the results are independent of a particular choice of Hamiltonian and also apply to interacting systems about which it is generally difficult to make precise statements.

A great virtue of the paper is that it gives a very clear analysis of the otherwise very confusing situation of an odd number of majorana modes. Such a situation is often discarded by condensed matter physicists as unphysical, because they imagine that the majorana modes arise from electron creation operators and therefore always come in pairs. The present paper describes (via Jordan-Wigner transform) at least one reasonable physical situation with a microscopic Hilbert space made just from spins, where one is confronted with an odd number of majorana modes.

The paper is long, partly because it reviews many things that are well-known, but it does so very clearly, in order to bring out all possible subtleties.

Some nice points were the proposal for which phase redefinitions of operators respect locality, the expressions for the (nonlocal) action of fermion translations on the Jordan-Wigner-dual bosons, and the completely explicit expression for the lattice Kramers-Wannier operator.

The paper also contains some valuable appendices summarizing the continuum field theory story.

The paper includes a careful comparison between lattice models and continuum field theory. One way to look at the main results is as a very concrete lattice understanding of fancy fermionic field theory anomalies (involving torsion parts of spin bordism groups). The analysis indeed confirms the expectation that these fancy topological field theory results do indeed account for the anomalies of these lattice models (at least after including some information about the Hamiltonian*). Perhaps this point is worth emphasizing more.

  • This point is one place where the paper could conceivably be strengthened. The authors show that the continuum mod 8 anomaly does emerge in the low-energy Hilbert space of the free theory, i.e. for a particular lattice Hamiltonian. It would be nice if it were possible to identify what properties of the lattice Hamiltonian suffice for this to be the case. Even showing that it is still the case for weak perturbations of the free theory would be valuable.

-- After equation (1.8) I was a little confused by the sentences "This defect is topological; it can be moved by conjugating HG by a unitary transformation. For example, conjugating it by χ1 moves the defect to the link (1, 2)."
Is $\chi_1$ the unitary transformation referred to in the first sentence? I guess it is indeed unitary in the sense that $\chi_1^2 = 1$. But perhaps some comment on its fermionic nature is warranted.

-- Before equation (1.16) "phrase redefinitions"-> "phase redefinitions"

-- page 44 "it cannot lead to a simply action on ..."

Requested changes

see report

---

## Round 2 · Referee Report · Anonymous (Referee 2) · 2024-2-11

Strengths

  1. Excellent review of the literature, both from a lattice and continuum perspective
  2. Gives a general framework for dealing with subtleties of lattice topological defects and their continuum counterparts that will be very valuable in the ongoing search for a dictionary between lattice and the continuum

Report

The authors present an in depth treatment of the interplay between translation, fermion parity, reflection and time-reversal symmetry in closed Majorana chains. On the fermionic Hilbert space, they find that these operators are realized projectively, which corresponds to a lattice counterpart to the anomalies of the corresponding continuum theory. Via a Jordan-Wigner transformation, the model is bosonized to the Ising model, and the Majorana translation operator is mapped to the well known non-invertible Kramers-Wannier duality operator of the critical Ising model.

The study of topological defects, both on the lattice and in the continuum, is a very active area of research and has received a lot of attention over the past years. Nevertheless, this paper presents a novelty by tackling a number of subtleties associated to topological defects on the lattice in a clear and systematic framework, thereby clarifying how one should think of lattice regularisations of continuum topological defects in the first place. I am convinced this paper is an invaluable addition to the existing literature and I am happy to recommend it for publication in SciPost Physics.

I have one remark that the authors might find interesting; there is quite a bit of literature on lattice regularisations of 2d rational CFTS via the use of integrable lattice models. For the minimal model CFTs, lattice expressions of the topological defects have been derived by essentially brute forcing the (boundary) Yang-Baxter equations (see e.g. https://doi.org/10.1016/S0370-2693(01)00982-0 and references therein). For the case of minimal models $M(p,p+1)$, these topological defects are labeled by a tuple $(r,s)$, the same labeling for the primaries. It has been observed that the lattice defects of type $(1,s)$ satisfy the expected fusion algebra, but that the defects of type $(r,1)$ only do so in the continuum limit. One finds that the defect $(p,1)$ corresponds to a half-site shift; in particular, for the Ising model $M(2,3)$ this implies that the defect $(2,2)$ satsifies the relations (6.11), and therefore most likely corresponds to the non-invertible operator $D$ proposed in this work. It would be interesting to apply the framework of emanant symmetries to these more general examples.

---

## Editorial Decision

resubmitted